# Exact Generalization Guarantees for (Regularized) Wasserstein Distributionally Robust Models

**Waïss Azizian,  Franck Iutzeler,  Jérôme Malick**
Univ. Grenoble Alpes, CNRS, Grenoble INP
Grenoble, 38000, France
`firstname.lastname@univ-grenoble-alpes.fr`

## Abstract

Wasserstein distributionally robust estimators have emerged as powerful models for prediction and decision-making under uncertainty. These estimators provide attractive generalization guarantees: the robust objective obtained from the training distribution is an exact upper bound on the true risk with high probability. However, existing guarantees either suffer from the curse of dimensionality, are restricted to specific settings, or lead to spurious error terms. In this paper, we show that these generalization guarantees actually hold on general classes of models, do not suffer from the curse of dimensionality, and can even cover distribution shifts at testing. We also prove that these results carry over to the newly-introduced regularized versions of Wasserstein distributionally robust problems.

## 1 Introduction

### 1.1 Generalization and (Wasserstein) Distributionally Robust Models

We consider the fundamental question of generalization of machine learning models. Let us denote by $f_\theta$ the loss induced by a model parametrized by $\theta$ for some uncertain variable $\xi$ (typically a data point). When $\xi$ follows some distribution P, seeking the best parameter $\theta$ writes as minimizing the expected loss

$$\min_{\theta \in \Theta} \ \mathbb{E}_{\xi \sim \mathrm{P}}[f_\theta(\xi)] \,.$$

We usually do not have a direct knowledge of P but rather we have access to samples $(\xi_i)_{i=1}^n$ independently drawn from P. The empirical risk minimization approach then consists in minimizing the expected loss over the associated empirical distribution $\widehat{\mathrm{P}}_\mathrm{n} = \frac{1}{n} \sum_{i=1}^n \delta_{\xi_i}$ (as a proxy for the expected loss over P), i.e.,

$$\min_{\theta \in \Theta} \ \mathbb{E}_{\xi \sim \widehat{\mathrm{P}}_\mathrm{n}}[f_\theta(\xi)] \ \left( = \frac{1}{n} \sum_{i=1}^n f_\theta(\xi_i) \right) \,.$$

Classical statistical learning theory ensures that, with high probability, $\mathbb{E}_\mathrm{P}[f_\theta]$ is close to $\mathbb{E}_{\widehat{\mathrm{P}}_\mathrm{n}}[f_\theta]$ up to $\mathcal{O}(1/\sqrt{n})$ error terms, see e.g., the monographs Boucheron et al. (2013); Wainwright (2019).

A practical drawback of empirical risk minimization is that it can lead to over-confident decisions (when $\mathbb{E}_{\widehat{\mathrm{P}}_\mathrm{n}}[f_\theta] < \mathbb{E}_\mathrm{P}[f_\theta]$, the real loss can be higher that the empirical one (Esfahani and Kuhn, 2018)). In addition, this approach is also sensitive to distribution shifts between training and application. To overcome these drawbacks, an approach gaining momentum in machine learning is *distributionally robust* optimization, which consists in minimizing the *worst expectation* of the loss when the distribution lives in a neighborhood of $\widehat{\mathrm{P}}_\mathrm{n}$:

$$\min_{\theta \in \Theta} \ \sup_{\mathrm{Q} \in \mathcal{U}(\widehat{\mathrm{P}}_\mathrm{n})} \ \mathbb{E}_{\xi \sim \mathrm{Q}}[f_\theta(\xi)] \,, \tag{1}$$

37th Conference on Neural Information Processing Systems (NeurIPS 2023).

where the inner sup is thus taken over Q in the neighborhood $\mathcal{U}(\widehat{P}_n)$ of $\widehat{P}_n$ in the space of probability distributions. Popular choices of distribution neighborhoods are based on the Kullback-Leibler (KL) divergence (Laguel et al., 2020; Levy et al., 2020), kernel tools (Zhu et al., 2021a; Staib and Jegelka, 2019; Zhu et al., 2021b), moments (Delage and Ye, 2010; Goh and Sim, 2010), or Wasserstein distance (Shafieezadeh Abadeh et al., 2015; Esfahani and Kuhn, 2018). If $P \in \mathcal{U}(\widehat{P}_n)$, distributionally robust models can benefit from direct generalization guarantees as,

$$\sup_{Q \in \mathcal{U}(\widehat{P}_n)} \mathbb{E}_{\xi \sim Q}[f_\theta(\xi)] \;\geq\; \mathbb{E}_{\xi \sim P}[f_\theta(\xi)]. \tag{2}$$

Thus, for well-chosen neighborhoods $\mathcal{U}(\widehat{P}_n)$, distributionally robust objectives are able to provide *exact upper-bounds* on the expected loss over distribution P, i.e., the true risk.

Wasserstein distributionally robust optimization (WDRO) problems correspond to (1) with

$$\mathcal{U}(\widehat{P}_n) = \left\{ Q \in \mathcal{P}(\Xi) : W(\widehat{P}_n, Q) \leq \rho \right\},$$

where $W(\widehat{P}_n, Q)$ denotes the Wasserstein distance between $\widehat{P}_n$ and Q and $\rho > 0$ controls the required level of robustness around $\widehat{P}_n$. As a natural metric to compare discrete and absolutely continuous probability distributions, the Wasserstein distance has attracted a lot of interest in both machine learning (Shafieezadeh Abadeh et al., 2015; Sinha et al., 2018; Shafieezadeh-Abadeh et al., 2019; Li et al., 2020; Kwon et al., 2020) and operation research (Zhao and Guan, 2018; Arrigo et al., 2022) communities; see e.g., the review articles Blanchet et al. (2021); Kuhn et al. (2019).

WDRO benefits from out-of-the-box generalization guarantees in the form of (2) since it inherits the concentration properties of the Wasserstein distance (Esfahani and Kuhn, 2018). More precisely, under mild assumptions on P, (Fournier and Guillin, 2015) establishes that $W(\widehat{P}_n, P) \leq \rho$ with high probability as soon as $\rho \sim 1/n^{1/d}$ where $d$ denotes the dimension of the samples space. Thus, a major issue is the prescribed radius $\rho$ suffers from the curse of the dimensionality: when $d$ is large, $\rho$ decreases slowly as the number of samples $n$ increases. This constrasts with other distributionally robust optimization ambiguity sets, such as Maximum Mean Discrepancy (MMD) (Staib and Jegelka, 2019; Zeng and Lam, 2022), where the radius scales as $1/\sqrt{n}$. Moreover, the existing scaling for WDRO is overly conservative for WDRO objectives since recent works (Blanchet et al., 2022a; Blanchet and Shapiro, 2023) prove that a radius behaving as $1/\sqrt{n}$ is asymptotically optimal. The main difference with (Esfahani and Kuhn, 2018) is that they — and us — consider the WDRO objective as a whole, instead of proceeding in two steps: first considering the Wasserstein distance independently and invoking concentration results on the Wasserstein distance and then plugging this result in the WDRO problem.

### 1.2 Contributions and related works

In this paper, we show that WDRO provides exact upper-bounds on the true risk with high probability. More precisely, we prove non-asymptotic generalization bounds of the form of (2), that hold for general classes of functions, and that only require $\rho$ to scale as $1/\sqrt{n}$ and not $1/n^{1/d}$. To do so, we construct an interval for the radius $\rho$ for which it is both sufficiently large so that we can go from the empirical to the true estimator (i.e., at least of the order of $1/\sqrt{n}$) and sufficiently small so that the robust problem does not become degenerate (i.e., smaller than some critical radius, that we introduce as an explicit constant). Our results imply proving concentration results on Wasserstein Distributionally Robust objectives that are of independent interest.

This work is part of a rich and recent line of research about theoretical guarantees on WDRO for machine learning. One of this first results, Lee and Raginsky (2018), provides generalization guarantees, for a general class of models and a fixed $\rho$, that, however, become degenerate as the radius goes to zero. In the particular case of linear models, WDRO models admit an explicit form that allows Shafieezadeh-Abadeh et al. (2019); Chen and Paschalidis (2018) to provide generalization guarantees (2) with the radius scaling as $1/\sqrt{n}$. The case of general classes of models, possibly non-linear, is more intricate. Sinha et al. (2018) showed that a modified version of (2) holds at the price of non-negligible error terms. Gao (2022); An and Gao (2021) made another step towards broad generalization guarantees for WDRO but with error terms that vanish only when $\rho$ goes to zero.

In contrast, our analysis provides exact generalization guarantees in the form (2) without additional error terms, that hold for general classes of functions and allow for a non-vanishing uncertainty radius to cover for distribution shifts at testing. Moreover, our guarantees also carry over to the recently introduced regularized versions of WDRO (Wang et al., 2023; Azizian et al., 2023), whose statistical properties have not been studied yet.

This paper is organized as follows. In Section 2, we introduce notations and our blanket assumptions. In Section 3, we present our main results, an idea of proof, and discussions. The complete proofs are deferred to the appendix.

## 2   Setup and Assumptions

In this section, we formalize our setting and introduce Wasserstein Distributionally Robust risks.

### 2.1   Wasserstein Distributionally Robust risk functions

In this paper, we consider as a samples space $\Xi$ a subset of $\mathbb{R}^d$ equipped with the Euclidean norm $\|\cdot\|$. We rely on Wasserstein distances of order 2, in line with the seminal work Blanchet et al. (2022a) on generalization of WDRO. This distance is defined for two distributions $Q, Q'$ in the set of probability distributions on $\Xi$, denoted by $\mathcal{P}(\Xi)$, as

$$W_2(Q, Q') := \left( \inf_{\pi \in \mathcal{P}(\Xi \times \Xi), \pi_1 = Q, \pi_2 = Q'} \mathbb{E}_{(\xi, \zeta) \sim \pi} \left[ \frac{1}{2} \|\xi - \zeta\|^2 \right] \right)^{1/2},$$

where $\mathcal{P}(\Xi \times \Xi)$ is the set of probability distributions in the product space $\Xi \times \Xi$, and $\pi_1$ (resp. $\pi_2$) denotes the first (resp. second) marginal of $\pi$.

We denote by $f : \Xi \to \mathbb{R}$ the loss function of some model over the sample space. The model may depend on some parameter $\theta$, that we drop for now to lighten the notations; instead, we consider a class of functions $\mathcal{F}$ encompassing our various models and losses of interest (we come back to classes of parametric models of the form $\mathcal{F} = \{f_\theta : \theta \in \Theta\}$ in Section 4).

We define the empirical Wasserstein Distributionally Robust risk $\widehat{\mathcal{R}}_{\rho^2}(f)$ centered on $\widehat{P}_n$ and similarly the true robust risk $\mathcal{R}_{\rho^2}(f)$ centered on P as

$$\widehat{\mathcal{R}}_{\rho^2}(f) := \sup_{\substack{Q \in \mathcal{P}(\Xi) \\ W_2^2(\widehat{P}_n, Q) \leq \rho^2}} \mathbb{E}_{\xi \sim Q}[f(\xi)] \qquad \text{and} \qquad \mathcal{R}_{\rho^2}(f) := \sup_{\substack{Q \in \mathcal{P}(\Xi) \\ W_2^2(P, Q) \leq \rho^2}} \mathbb{E}_{\xi \sim Q}[f(\xi)]. \tag{3}$$

Note that $\widehat{\mathcal{R}}_{\rho^2}(f)$, which is based on the empirical distribution $\widehat{P}_n$, is a computable proxy for the true robust risk $\mathcal{R}_{\rho^2}(f)$. Note also that the true robust risk $\mathcal{R}_{\rho^2}(f)$ immediately upper-bounds the true (non-robust) risk $\mathbb{E}_{\xi \sim P}[f(\xi)]$ and also upper-bounds $\mathbb{E}_{\xi \sim Q}[f(\xi)]$ for neighboring distributions Q that correspond to distributions shifts of magnitude smaller than $\rho$ in Wasserstein distance.

### 2.2   Regularized versions

Entropic regularization of WDRO problems was recently studied in Wang et al. (2023); Blanchet and Kang (2020); Piat et al. (2022); Azizian et al. (2023) and used in Dapogny et al. (2023); Song et al. (2023); Wang and Xie (2022); Wang et al. (2022). Inspired by the entropic regularization in optimal transport (OT) (Peyré and Cuturi, 2019, Chap. 4), the idea is to regularize the objective by adding a KL divergence, that is defined, for any transport plan $\pi \in \mathcal{P}(\Xi \times \Xi)$ and a fixed reference $\overline{\pi} \in \mathcal{P}(\Xi \times \Xi)$, by

$$\text{KL}(\pi \,|\, \overline{\pi}) = \begin{cases} \int \log \frac{\mathrm{d}\pi}{\mathrm{d}\overline{\pi}} \, \mathrm{d}\pi & \text{when } \pi \ll \overline{\pi} \\ +\infty & \text{otherwise.} \end{cases}$$

Unlike in OT though, the choice of the reference measure in WDRO is not neutral and introduces a bias in the robust objective (Azizian et al., 2023). For their theoretical convenience, we take reference measures that have Gaussian conditional distributions

$$\pi_\sigma(\mathrm{d}\zeta|\xi) \propto \mathbb{1}_{\zeta \in \Xi} \, e^{-\frac{\|\xi - \zeta\|^2}{2\sigma^2}} \mathrm{d}\zeta, \qquad \text{for all } \xi \in \Xi, \tag{4}$$

where $\sigma > 0$ controls the spread of the second marginals, following Wang et al. (2023); Azizian et al. (2023). Then, the regularized version of $\widehat{\mathcal{R}}_{\rho^2}(f)$ (WDRO empirical risk) is given by

$$\widehat{\mathcal{R}}^{\varepsilon}_{\rho^2}(f) \coloneqq \sup_{\substack{\pi \in \mathcal{P}(\Xi \times \Xi), \pi_1 = \widehat{\mathrm{P}}_{\mathrm{n}} \\ \mathbb{E}_{(\xi,\zeta) \sim \pi}\left[\frac{1}{2}\|\xi - \zeta\|^2\right] \leq \rho^2}} \mathbb{E}_{\xi \sim \pi_2}\left[f(\xi)\right] - \varepsilon\, \mathrm{KL}\left(\pi \,|\, \pi^n_\sigma\right) \qquad \text{with } \pi^n_\sigma = \widehat{\mathrm{P}}_{\mathrm{n}}(\mathrm{d}\xi)\,\pi_\sigma(\mathrm{d}\zeta|\xi) \quad (5)$$

and similarly, the regularized version of $\mathcal{R}_{\rho^2}(f)$ is given by

$$\mathcal{R}^{\varepsilon}_{\rho^2}(f) \coloneqq \sup_{\substack{\pi \in \mathcal{P}(\Xi \times \Xi), \pi_1 = \mathrm{P} \\ \mathbb{E}_{(\xi,\zeta) \sim \pi}\left[\frac{1}{2}\|\xi - \zeta\|^2\right] \leq \rho^2}} \mathbb{E}_{\xi \sim \pi_2}\left[f(\xi)\right] - \varepsilon\, \mathrm{KL}\left(\pi \,|\, \pi_\sigma\right) \qquad \text{with } \pi_\sigma = \mathrm{P}(\mathrm{d}\xi)\,\pi_\sigma(\mathrm{d}\zeta|\xi). \quad (6)$$

These regularized risks have been studied in terms of computational or approximation properties, but their statistical properties have not been investigated yet. The analysis we develop for WDRO estimators is general enough to carry over to these settings.

In (5) and (6), note finally that the regularization is added as a penalization in the supremum, rather than in the constraint. As in Wang et al. (2023), penalizing in the constraint leads to an ambiguity set defined by the regularized Wasserstein distance, that we introduce in Section 3.2. We refer to Azizian et al. (2023) for a unified presentation of the two penalizations.

### 2.3 Blanket assumptions

Our analysis is carried under the following set of assumptions that will be in place throughout the paper. First, we assume that the sample space $\Xi \subset \mathbb{R}^d$ is convex and compact, which is in line with previous work, e.g., (Lee and Raginsky, 2018; An and Gao, 2021).

**Assumption 1** (On the set $\Xi$). The sample space $\Xi$ is a compact convex subset of $\mathbb{R}^d$.

Second, we require the class of loss functions $\mathcal{F}$ to be sufficiently regular. In particular, we assume that they have Lipschitz continuous gradients.

**Assumption 2** (On the function class). The functions of $\mathcal{F}$ are twice differentiable, uniformly bounded, and their derivatives are uniformly bounded and uniformly Lipschitz.

Finally, we assume that $\widehat{\mathrm{P}}_{\mathrm{n}}$ is made of independent and identically distributed (i.i.d.) samples of $\mathrm{P}$ and that $\mathrm{P}$ is supported on the interior of $\Xi$ (which can be done without loss of generality by slightly enlarging $\Xi$ if needed).

**Assumption 3** (On the distributions). $\widehat{\mathrm{P}}_{\mathrm{n}} = \frac{1}{n}\sum_{i=1}^{n}\delta_{\xi_i}$ where $\xi_1, \ldots, \xi_n$ are i.i.d. samples of $\mathrm{P}$. We further assume that there is some $R > 0$ such that $\mathrm{P}$ satisfies $\operatorname{supp}\mathrm{P} + \overline{\mathbb{B}}(0, R) \subset \Xi$.

## 3 Main results and discussions

The main results of our paper establish that the empirical robust risk provide high probability bounds, of the form of (2), on the true risk. Since the results and assumptions slightly differ between the WDRO models and their regularized counterparts, we present them separately in Section 3.1 and Section 3.2. In Section 3.3, we provide the common outline for the proofs of these results, the proofs themselves being provided in the appendix. Finally, in Section 4, we detail some examples.

### 3.1 Exact generalization guarantees for WDRO models

In this section, we require the two following additional assumptions on the function class. The first assumption is common in the WDRO litterature, see e.g., Blanchet et al. (2022a); Blanchet and Shapiro (2023); Gao (2022); An and Gao (2021).

**Assumption 4.** The quantity $\inf_{f \in \mathcal{F}} \mathbb{E}_{\mathrm{P}}\left[\|\nabla f\|^2\right]$ is positive.

The second assumption we consider in this section makes use of the notation $d(\xi, A)$, for a set $A \subset \Xi$ and a point $\xi \in \Xi$, to denote the distance between $\xi$ and $A$, i.e., $d(\xi, A) = \inf_{\zeta \in A}\|\xi - \zeta\|$.

**Assumption 5.**

1. For any $R > 0$, there exists $\Delta > 0$ such that,

$$\forall f \in \mathcal{F}, \forall \zeta \in \Xi, \quad d(\zeta, \arg\max f) \geq R \implies f(\zeta) - \max f \leq -\Delta\,.$$

2. The following growth condition holds: there exist $\mu > 0$ and $L > 0$ such that, for all $f \in \mathcal{F}, \xi \in \Xi$ and $\xi^*$ a projection of $\xi$ on $\arg\max f$, i.e., $\xi^* \in \arg\min_{\arg\max f}\|\xi - \cdot\|$,

$$f(\xi^*) \geq f(\xi) + \frac{\mu}{2}\|\xi - \xi^*\|^2 - \frac{L}{6}\|\xi - \xi^*\|^3\,.$$

The first item of this assumption has a natural interpretation: we show in Lemma A.7, that it is equivalent to the relative compactness of the function space $\mathcal{F}$ w.r.t. to the distance

$$D(f, g) := \|f - g\|_\infty + D_H(\arg\max f, \arg\max g)\,,$$

where $D_H$ denotes the (Hausdorff) distance between sets and $\|f\|_\infty := \sup_{\xi \in \Xi}|f(\xi)|$ is the infinity norm. The last one is a structural assumption on the functions $\mathcal{F}$ that is new in our context but is actually very close the so-called parametric Morse-Bott condition, introduced in of bilevel optimization (Arbel and Mairal, 2022), see Section A.5.

We now state our main generalization result for WDRO risks.

**Theorem 3.1.** *Under Assumptions 4 and 5, there is an explicit constant $\rho_c$ depending only on $\mathcal{F}$ and P such that for any $\delta \in (0, 1)$ and $n \geq 1$, if*

$$\mathcal{O}\left(\sqrt{\frac{1 + \log 1/\delta}{n}}\right) \leq \rho \leq \frac{\rho_c}{2} - \mathcal{O}\left(\sqrt{\frac{1 + \log 1/\delta}{n}}\right) \tag{7}$$

*then, there is $\rho_n = \mathcal{O}\left(\sqrt{\frac{1 + \log 1/\delta}{n}}\right)$ such that, with probability $1 - \delta$,*

$$\forall f \in \mathcal{F}, \quad \widehat{\mathcal{R}}_{\rho^2}(f) \geq \mathbb{E}_{\xi \sim Q}\left[f(\xi)\right] \qquad \text{for all } Q \text{ such that } W_2^2(P, Q) \leq \rho(\rho - \rho_n)\,. \tag{8}$$

*In particular, with probability $1 - \delta$, we have*

$$\forall f \in \mathcal{F}, \quad \widehat{\mathcal{R}}_{\rho^2}(f) \geq \mathbb{E}_{\xi \sim P}\left[f(\xi)\right]\,. \tag{9}$$

The second part of the result, (9), is an *exact* generalization bound: it is an actual upper-bound on the true risk $\mathbb{E}_{\xi \sim P}[f(\xi)]$, that we cannot access in general, through a quantity that we can actually compute with $\widehat{P}_n$. The first part of the result, (8) gives us insight into the robustness guarantees offered by the WDRO risk. Indeed, it tells us that, when $\rho$ is greater than the minimal radius $\rho_n \propto 1/\sqrt{n}$ by some margin, the empirical robust risk $\widehat{\mathcal{R}}_{\rho^2}(f)$ is an upper-bound on the loss even with some perturbations of the true distribution. Hence, as long as $\rho$ is large enough, the WDRO objective enables us to guarantee the performance of our model even in the event of a distribution shift at testing time. In other words, the empirical robust risk is an *exact* upper-bound on the true robust risk $\mathcal{R}_{\rho(\rho-\rho_n)}(f)$ with a reduced radius.

The range of admissible radiuses is described by (7). The lower-bound, roughly proportional to $1/\sqrt{n}$, is optimal, following the results of Blanchet et al. (2022a). The upper-bound, almost independent of $n$, depends on a constant $\rho_c$, that we call *critical radius* and that has an interesting interpretation, that we formalize in the following remark. Note, finally, that, the big-O notation in this theorem has a slightly stronger meaning[1] than the usual one, being non-asymptotic in $n$ and $\delta$.

**Remark 3.2** (Interpretation of critical radius)**.** *The critical radius $\rho_c$, appearing in (7), is defined by*

$$\rho_c^2 := \inf_{f \in \mathcal{F}} \mathbb{E}_{\xi \sim P}\left[\frac{1}{2}d^2(\xi, \arg\max f)\right]\,.$$

---

[1]Eg., $\rho_n = \mathcal{O}\left(\sqrt{\frac{1 + \log 1/\delta}{n}}\right)$ means that $\exists C > 0$ such that $\rho_n \leq C\sqrt{\frac{1 + \log 1/\delta}{n}}$ for all $\delta \in (0, 1)$ and $n \geq 1$.

It can be interpreted as the threshold at which the WDRO problem w.r.t. P *starts becoming degenerate. Indeed, when* $\rho^2 \geq \mathbb{E}_{\xi \sim P}\left[\frac{1}{2}d^2(\xi, \arg\max f)\right]$ *for some* $f \in \mathcal{F}$ *that we fix, the distribution* Q *given by the second marginal of the transport plan* $\pi$ *defined by,*

$$\pi(d\xi, d\zeta) := P(d\xi)\delta_{\zeta^\star(\xi)}(d\zeta) \quad where \quad \zeta^\star(\xi) \in \underset{\zeta \in \arg\max f}{\arg\min} \; d^2(\xi, \zeta),$$

*satisfies*

$$W_2^2(P, Q) \leq \mathbb{E}_{(\xi,\zeta)\sim\pi}\left[\frac{1}{2}\|\xi - \zeta\|^2\right] = \mathbb{E}_{\xi\sim P}\left[\frac{1}{2}d^2(\xi, \arg\max f)\right] \leq \rho^2.$$

*As a consequence, the robust problem is equal to*

$$\mathcal{R}_{\rho^2}(f) = \sup_{\substack{Q \in \mathcal{P}(\Xi) \\ W_2^2(P,Q) \leq \rho^2}} \mathbb{E}_{\xi\sim Q}\left[f(\xi)\right] = \max_{\xi \in \Xi} f(\xi).$$

*Thus, when the radius exceeds* $\rho_c$, *there is some* $f$ *such that the robust problem becomes degenerate as it does not depend on* P *nor* $\rho$ *anymore.*

Finally, note that we can obtain the same generalization guarantee as Theorem 3.1 without Assumption 5 at the expense of losing the above interpration on the condition on the radius. More precisely, we have the following result.

**Theorem 3.3.** *Let Assumption 4 hold. For any* $\delta \in (0, 1)$ *and* $n \geq 1$, *if* $\rho$ *satisfies (7), and if, in addition, it is smaller than a positive constant which depends only on* P, $\mathcal{F}$ *and* $\Xi$, *then both conclusions of Theorem 3.1 hold.*

This theorem can be compared to existing results, and in particular with Gao (2022); An and Gao (2021). These two papers provide generalization bounds for WDRO under a similar assumption on $\mathcal{F}$ and a weakened version of Assumption 3. However, these generalization bounds involve extra error terms, that require $\rho$ to be vanishing. In comparison, with a similar set of assumptions, Theorem 3.3 improves on these two issues, by allowing $\rho$ not to vanish as $n \to \infty$ and by providing the exact upper-bound (9). Allowing non-vanishing radiuses is an attractive feature of our results that enables us to cover distribution shifts.

### 3.2 Regularized WDRO models

The analysis that we develop for the standard WDRO estimators is general enough to also cover the regularized versions presented in Section 2.2. We thus obtain the following Theorem 3.4 which is the first generalization guarantee for regularized WDRO. This theorem is very similar to Theorem 3.1 with still a couple of differences. First, the regularization leads to ambiguity sets defined in terms of $W_{2,\tau}(P, \cdot)$, the *regularized* Wasserstein distance to the true distribution P, defined, for some regularization parameter $\tau > 0$, as

$$W_{2,\tau}^2(P, Q) := \inf\left\{\mathbb{E}_\pi\left[\frac{1}{2}\|\xi - \zeta\|^2\right] + \tau \, \mathrm{KL}(\pi \,|\, \pi_\sigma) : \pi \in \mathcal{P}(\Xi \times \Xi), \pi_1 = P, \pi_2 = Q\right\},$$

where $\pi_\sigma$ appears in the definition of the regularized robust risk (6). Besides, the regularization allows us to avoid Assumptions 4 and 5 to show our generalization result.

**Theorem 3.4.** *For* $\sigma = \sigma_0\rho$ *with* $\sigma_0 > 0$, $\varepsilon = \varepsilon_0\rho$ *with* $\varepsilon_0 > 0$ *such that* $\varepsilon_0/\sigma_0^2$ *is small enough depending on* $\mathcal{F}$, P, $\Xi$, *there is an explicit constant* $\rho_c$ *depending only on* $\mathcal{F}$, P *and* $\Xi$ *such that for all* $\delta \in (0, 1)$ *and* $n \geq 1$, *if*

$$\mathcal{O}\left(\sqrt{\frac{1 + \log 1/\delta}{n}}\right) \leq \rho \leq \frac{\rho_c}{2} - \mathcal{O}\left(\frac{1}{\sqrt{n}}\right), \quad and \quad \rho_c \geq \mathcal{O}\left(\frac{1}{n^{1/6}} + \left(\frac{1 + \log 1/\delta}{n}\right)^{1/4}\right),$$

*then, there are* $\tau = \mathcal{O}(\varepsilon\rho)$ *and* $\rho_n = \mathcal{O}\left(\sqrt{\frac{1+\log 1/\delta}{n}}\right)$ *such that, with probability at least* $1 - \delta$,

$$\forall f \in \mathcal{F}, \quad \widehat{\mathcal{R}}_{\rho^2}^\varepsilon(f) \geq \mathbb{E}_{\xi\sim Q}\left[f(\xi)\right] \qquad for \; all \; Q \; such \; that \; W_{2,\tau}^2(P, Q) \leq \rho(\rho - \rho_n). \tag{10}$$

*Furthermore, when* $\sigma_0$ *and* $\sigma$ *are small enough depending on* P *and* $\Xi$, *with probability* $1 - \delta$,

$$\forall f \in \mathcal{F}, \quad \widehat{\mathcal{R}}_{\rho^2}^\varepsilon(f) \geq \mathbb{E}_{\xi\sim P}\mathbb{E}_{\zeta\sim\pi_\sigma(\cdot|\xi)}\left[f(\zeta)\right].$$

The first part of the theorem, (10), guarantees that the empirical robust risk is an upper-bound on the loss even with some perturbations of the true distribution. As in OT, the regularization added to the Wasserstein metric induces a bias that may prevent $W^2_{2,\tau}(\mathrm{P},\mathrm{P})$ from being null. As a result, the second part of the theorem involves a smoothed version of the true risk: the empirical robust risk provides an *exact* upper-bound the true expectation of a convolution of the loss with $\pi_\sigma$.

A few additional comments are in order:

- Our result prescribes the scaling of the regularization parameters: $\varepsilon$ and $\sigma$ should be taken proportional to $\rho$.

- The critical radius $\rho_c$ has a slighlty more intricate definition, yet the same interpretation as in the standard WDRO case in Remark 3.2; see Section D.2.

- The regularized OT distances do not suffer from the curse of dimensionality (Genevay et al., 2019). However this property does not directly carry over to regularized WDRO. Indeed, we cannot choose the same reference measure as in OT and we have to fix the measure $\pi_\sigma$, introducing a bias. As a consequence, we have to extend the analysis of the previous section to obtain the claimed guarantees that avoid the curse of dimensionality.

### 3.3 Idea of the proofs

In this section, we present the main ideas of the proofs of Theorem 3.1, Theorem 3.3, and Theorem 3.4. The full proofs are detailed in appendix; we point to relevant sections along the discussion. First, we recall the duality results for WDRO that play a crucial role in our analysis. Second, we present a rough sketch of proofs that is common to both the standard and the regularized cases. Finally, we provide a refinement of our results that is a by-product of our analysis.

**Duality in WDRO.** Duality has been a central tool in both the theoretical analyses and computational schemes of WDRO from the onset (Shafieezadeh Abadeh et al., 2015; Esfahani and Kuhn, 2018). The expressions of the dual of WDRO problems for both the standard case (Gao and Kleywegt, 2016; Blanchet and Murthy, 2019) and the regularized case (Wang et al., 2023; Azizian et al., 2023) can be written with the following dual generator function $\phi$ defined as

$$\phi(f,\xi,\lambda,\varepsilon,\sigma) := \begin{cases} \sup_{\zeta\in\Xi}\left\{f(\zeta) - \frac{\lambda}{2}\|\xi-\zeta\|^2\right\} & \text{if } \varepsilon = 0 \\ \varepsilon\log\left(\mathbb{E}_{\zeta\sim\pi_\sigma(\cdot|\xi)}\exp\left(\frac{f(\zeta)-\lambda\|\xi-\zeta\|^2/2}{\varepsilon}\right)\right) & \text{if } \varepsilon > 0, \end{cases} \tag{11}$$

where $\lambda$ is the dual variable associated to the Wasserstein constraint in (3), (5) and (6). The effect of regularization appears here clearly as a smoothing of the supremum. Note also that this function depends on the conditional reference measures $\pi_\sigma(\cdot|\xi)$ but not on other probability distributions. Then, under some general assumptions (specified in Section 2.3 in appendix), the existing strong duality results yield that the (regularized) empirical robust risk writes

$$\widehat{\mathcal{R}}^\varepsilon_{\rho^2}(f) = \inf_{\lambda\geq 0}\lambda\rho^2 + \mathbb{E}_{\xi\sim\widehat{\mathrm{P}}_n}\left[\phi(f,\xi,\lambda,\varepsilon,\sigma)\right], \tag{12}$$

and, similarly, the (regularized) true robust risk writes

$$\mathcal{R}^\varepsilon_{\rho^2}(f) = \inf_{\lambda\geq 0}\lambda\rho^2 + \mathbb{E}_{\xi\sim\mathrm{P}}\left[\phi(f,\xi,\lambda,\varepsilon,\sigma)\right]. \tag{13}$$

These expressions for the risks are the bedrock of our analysis.

**Sketch of proof.** In both the standard case and the regularized case, our proof is built on two main parts: the first part is to obtain a concentration bound on the dual problems that crucially relies on a lower bound of the dual multiplier; the second part then consists in establishing such a lower bound. All the bounds are valid with high probability, and we drop the dependency on the confidence level $\delta$ of the theorems for simplicity.

For the first part of the proof (Section B), we assume that there is a deterministic lower-bound $\underline{\lambda} > 0$ on the optimal dual multiplier in (12) that holds with high-probability. As a consequence, we can

restrict the range of $\lambda$ in (12) to obtain:

$$\widehat{\mathcal{R}}^{\varepsilon}_{\rho^2}(f) = \inf_{\lambda \geq \underline{\lambda}} \left\{ \lambda \rho^2 + \mathbb{E}_{\xi \sim \widehat{P}_n}[\phi(f, \xi, \lambda, \varepsilon, \sigma)] \right\}$$

$$= \inf_{\lambda \geq \underline{\lambda}} \left\{ \lambda \rho^2 + \mathbb{E}_{\xi \sim P}[\phi(f, \xi, \lambda, \varepsilon, \sigma)] - \lambda \frac{\mathbb{E}_{\xi \sim P}[\phi(f, \xi, \lambda, \varepsilon, \sigma)] - \mathbb{E}_{\xi \sim \widehat{P}_n}[\phi(f, \xi, \lambda, \varepsilon, \sigma)]}{\lambda} \right\}$$

$$\geq \inf_{\lambda \geq \underline{\lambda}} \left\{ \lambda \rho^2 + \mathbb{E}_{\xi \sim P}[\phi(f, \xi, \lambda, \varepsilon, \sigma)] - \lambda \sup_{\lambda' \geq \underline{\lambda}} \frac{\mathbb{E}_{\xi \sim P}[\phi(f, \xi, \lambda', \varepsilon, \sigma)] - \mathbb{E}_{\xi \sim \widehat{P}_n}[\phi(f, \xi, \lambda', \varepsilon, \sigma)]}{\lambda'} \right\}$$

$$\geq \inf_{\lambda \geq \underline{\lambda}} \left\{ \lambda \rho^2 + \mathbb{E}_{\xi \sim P}[\phi(f, \xi, \lambda, \varepsilon, \sigma)] - \lambda \overline{\rho}_n^2 \right\}$$

$$\geq \mathcal{R}^{\varepsilon}_{\rho^2 - \overline{\rho}_n^2}(f). \tag{14}$$

In the above, we used that the inner supremum, which is random, can be bounded by a deterministic and explicit quantity that we call $\overline{\rho}_n^2$, i.e.,

$$\overline{\rho}_n^2 \geq \sup_{\lambda' \geq \underline{\lambda}} \frac{\mathbb{E}_{\xi \sim P}[\phi(f, \xi, \lambda', \varepsilon, \sigma)] - \mathbb{E}_{\xi \sim \widehat{P}_n}[\phi(f, \xi, \lambda', \varepsilon, \sigma)]}{\lambda'} \quad \text{with high probability.}$$

Hence, we obtain an upper-bound on the robust risk w.r.t. the true distribution with radius $\rho^2 - \overline{\rho}_n^2$. Moreover, we show that $\overline{\rho}_n^2 = \mathcal{O}(1/(\underline{\lambda}\sqrt{n}))$ which highlights the need for a precise lower bound $\underline{\lambda}$ to control the decrease in radius.

The second part of the proof thus consists in showing that the dual variable is indeed bounded away from 0, which means that the Wasserstein constraint is *sufficiently active*. We have to handle two cases differently:

- when $\rho$ is small, i.e., close to $\rho_n$ (Section C),
- when $\rho$ is large, i.e., close to the critical radius $\rho_c$ (Section D). Note that the additional Assumption 5 is required here: where we need to control the behaviors of $f \in \mathcal{F}$ close to their maxima (see (11) for $\varepsilon = 0$ and small $\lambda$).

In both cases we obtain that $\underline{\lambda}$ scales as $1/\rho$ for the respective ranges of admissible radiuses. As a consequence $\overline{\rho}_n^2$ is bounded by $\rho \rho_n$ with $\rho_n = \mathcal{O}(1/\sqrt{n})$ and (14) becomes

$$\widehat{\mathcal{R}}^{\varepsilon}_{\rho^2}(f) \geq \mathcal{R}^{\varepsilon}_{\rho(\rho - \rho_n)}(f), \tag{15}$$

which leads to our main results.

**Extension: upper and lower bounds on the empirical robust risk.** The proof that we sketched above actually shows that $\mathcal{R}^{\varepsilon}_{\rho(\rho - \rho_n)}(f)$ is a lower bound of $\widehat{\mathcal{R}}^{\varepsilon}_{\rho^2}(f)$. This proof technique also yields an upper bound by exchanging the roles of P and $\widehat{P}_n$.

**Theorem 3.5.** *In the setting of either Theorem 3.1, Theorem 3.3 or Theorem 3.4 (with $\varepsilon = 0$ or $\varepsilon > 0$), with probability at least $1 - \delta$, it holds that*

$$\forall f \in \mathcal{F}, \quad \mathcal{R}^{\varepsilon}_{\rho(\rho - \rho_n)}(f) \leq \widehat{\mathcal{R}}^{\varepsilon}_{\rho^2}(f) \leq \mathcal{R}^{\varepsilon}_{\rho(\rho + \rho_n)}(f),$$

*with $\rho_n = \mathcal{O}\left(\sqrt{\frac{1 + \log 1/\delta}{n}}\right)$.*

This result shows how two robust objectives w.r.t. P provide upper and lower bounds on the empirical robust risk, with only slight variations in the radius. Furthermore, when the number of data points $n$ grows, both sides of the bound converge to the same quantity $\mathcal{R}^{\varepsilon}_{\rho^2}(f)$. Hence our generalization bounds of the form (15) are asymptotically tight.

As a final remark, we underline that the proofs of this theorem and of the previous ones rely on the cost being the squared Euclidean norm and the extension to more general cost functions is left as future work. In particular, the Laplace approximation of Section A.3 in the regularized case and the analysis of Section D.1 in the standard WDRO case would need further work to accomodate general cost functions.

# 4 Examples: parametric models

Our main theorems Theorems 3.1, 3.3 and 3.4 involve a general class $\mathcal{F}$ of loss functions. We explain in this section how to instantiate our results in the important class of parametric models. We then illustrate this setting with logistic regression and linear regression in Examples 4.1 and 4.2.

Let us consider the class of functions of the form

$$\mathcal{F} = \{\xi \mapsto f(\theta, \xi) : \theta \in \Theta\} \qquad \text{with } f : \Theta \times \Xi \longrightarrow \mathbb{R} \tag{16}$$

where $\Theta$, the parameter space, is a subset of $\mathbb{R}^p$ and $\Xi$, the sample space, is a subset of $\mathbb{R}^d$.

For instance, this covers the case of linear models of the form $f(\theta, \xi) = \ell(\langle \xi, \theta \rangle)$ with $\ell$ a convex loss. This class of models is studied by Shafieezadeh-Abadeh et al. (2019); Chen and Paschalidis (2018) in a slightly different setting, where they obtain a closed form for the robust objective and then establish a generalization bound similar to (9).

Let us show how to instantiate our theorems in the case of (16).

- If $f$ is twice continuously differentiable on a neighborhood of $\Theta \times \Xi$ with $\Theta$ and $\Xi$ both compact, then Assumption 2 is immediately satisfied. Therefore, Theorem 3.4 can be readily applied and its generalization guarantee hold.

- As for Assumption 4, it is equivalent to, for all $\theta \in \Theta$, $P(\nabla_\xi f(\theta, \xi) \neq 0) > 0$. Thus disregarding the degenerate case of $\nabla_\xi f(\theta, \xi)$ being null for P-almost every $\xi$ (e.g., when the loss does not depend on $\xi$), we are in the setting of Theorem 3.3.

- Satisfying Assumption 5, needed for Theorem 3.1, requires some problem-dependent developments, see the examples below. Note though that the second item of Assumption 5 is implied by the parametric Morse-Bott property (Arbel and Mairal, 2022); see Section A.5.

We discuss linear and non-linear examples of this framework. In light of the above, we focus our discussion on Assumption 5. We first present the examples of linear models, Examples 4.1 and 4.2, where the latter assumption is satisfied. We then consider several examples of nonlinear models: kernel regression (Example 4.3), smooth neural networks (Example 4.4) and families of invertible mappings (Example 4.5). In Section H, we also provide numerical illustrations for linear models.

**Example 4.1** (Logistic Regression). *For a training sample $(x, y) \in \mathbb{R}^p \times \{-1, +1\}$, the logistic loss for a parameter $\theta \in \mathbb{R}^p$ is given by $\log(1 + e^{-y\langle x, \theta \rangle})$. It fits into our framework by defining $f(\theta, \xi) = \log(1 + e^{\langle \xi, \theta \rangle})$ with $\xi$ playing the role of $-y \times x$. We assume that $\Theta$ is a compact set that does not include the origin, and, for the sake of simplicity, we take $\Xi$ as a closed Euclidean ball, i.e., $\Xi = \overline{\mathbb{B}}(0, r)$. We are going to show that Assumption 5 is satisfied, and, for this, we need the following elements. For any $\theta$, the maximizer of $f(\theta, \cdot)$ over $\Xi = \overline{\mathbb{B}}(0, r)$ is reached at $\xi^* := \frac{r\theta}{\|\theta\|}$. Besides, for any $\xi \in \Xi$, it holds that*

$$r^2 \geq \|\xi\|^2 = \|\xi^*\|^2 + 2\langle \xi^*, \xi - \xi^* \rangle + \|\xi - \xi^*\|^2,$$

*so that, since $\|\xi^*\| = r$, we have*

$$\langle \xi^*, \xi^* - \xi \rangle \geq \frac{1}{2} \|\xi - \xi^*\|^2. \tag{17}$$

*We can now turn to the verification of Assumption 5.*

1. *Take some $R > 0$ and some $\xi \in \Xi$ such that $\|\xi - \xi^*\| \geq R$. Then, (17) yields*

$$\langle \theta, \xi \rangle - \langle \theta, \xi^* \rangle = \frac{\|\theta\|}{r} \langle \xi^*, \xi - \xi^* \rangle \leq -\frac{\|\theta\|}{2r} \|\xi - \xi^*\|^2 \leq -\frac{d(0, \Theta) R^2}{2r}. \tag{18}$$

*Since $u \mapsto \log(1 + e^u)$ is increasing, this yields that $f(\theta, \xi) - f(\theta, \xi^*)$ is bounded away from 0 by a negative constant uniformly in $\theta$. The first item of Assumption 5 is thus satisfied.*

2. *Fix $\theta \in \Theta$; by Taylor expanding $u \mapsto \log(1 + e^u)$ around $\langle \theta, \xi^* \rangle$ we get*

$$f(\theta, \xi) = f(\theta, \xi^*) + \frac{1}{1 + e^{-\langle \theta, \xi^* \rangle}} \langle \theta, \xi - \xi^* \rangle + \mathcal{O}(\langle \theta, \xi - \xi^* \rangle)^2,$$

*where the big-O remainder is uniform over $\theta \in \Theta$. Using the first inequality in (18), we get for $\xi$ close enough to $\xi^*$*

$$f(\theta, \xi) \leq f(\theta, \xi^*) - \frac{1}{2(1 + e^{-\langle \theta, \xi^* \rangle})} \langle \theta, \xi - \xi^* \rangle \leq f(\theta, \xi^*) - \frac{\|\theta\|}{4r} \|\xi - \xi^*\|^2.$$

*This shows that the second item of Assumption 5 is satisfied locally around $\xi^*$. It can be made global by using the uniform Lipschitz-continuity of $f$, which introduces a term of the form $\frac{L}{6} \|\xi - \xi^*\|^3$.*

**Example 4.2** (Linear Regression). *With samples of the form $\xi = (x, y) \in \mathbb{R}^p \times \mathbb{R}$ and parameters $\theta \in \mathbb{R}^p$, the loss is given by $f(\theta, \xi) = \frac{1}{2}(\langle \theta, x \rangle - y)^2$. Similarly to the previous example, we take $\Theta$ as a compact set of $\mathbb{R}^d$ that does not include the origin and $\Xi$ of the form $\overline{\mathbb{B}}(0, r) \times [-r', r']$. The maximizers of $f(\theta, \cdot)$ on $\Xi$ are $\xi_1^* = (r\theta/\|\theta\|, -r')$ and $\xi_2^* = (-r\theta/\|\theta\|, r')$. By symmetry, one can restrict to the case of $\xi_1^*$ and $\langle \theta, x \rangle - y \geq 0$; the same rationale as above can then be applied.*

**Example 4.3** (Kernel Ridge Regression). *Using a kernel $k : \mathcal{X} \times \mathcal{X} \to \mathbb{R}$ with $\mathcal{X}$ compact and $k$ smooth, for instance Gaussian or polynomial, we consider the following class of loss functions:*

$$f(\theta, \xi) = \frac{1}{2} \left( \sum_{i=1}^m \alpha_i k(x, x_i) - y \right)^2 + \frac{\mu}{2} \|\alpha\|_2^2.$$

*where $\xi = (x, y)$, $\Xi$ is some compact subset of $\mathcal{X} \times \mathbb{R}$, $\theta = (\alpha_1, \ldots, \alpha_m, x_1, \ldots, x_m)$, $\Theta = A_m \times \mathcal{X}_m$, $m$ is a fixed integer, $A$ is a compact subset of $\mathbb{R}^m$, $\mathcal{X}_m$ can be any closed subset of $\mathcal{X}^m$ and $\mu \geq 0$ is the regularization parameter. A typical choice for $\mathcal{X}_m$ would be the datapoints of the training set. This class then fits into our framework of parametric models above. Finally, further information on the kernel would be needed to ensure that Assumption 5 is satisfied.*

**Example 4.4** (Smooth Neural Networks). *Denote by $\mathcal{NN}(x, \theta, \sigma)$ a multi-linear perceptron that takes $x$ as input, has weights $\theta$ and a smooth activation function $\sigma$, for instance the hyperbolic tangent or the Gaussian Error Linear Units (GELU). We choose $\ell(\hat{y}, y)$ a smooth loss function and we consider the loss $f(\theta, (x, y)) = \ell(\mathcal{NN}(x, \theta, \sigma), y)$ with $\theta \in \Theta$ some compact set. Provided that the inputs $(x, y)$ lie in a compact set $\Xi$, this class fits the parametric framework above. Note that we require $\sigma$ to be smooth, further work would be required for non-smooth activation functions.*

**Example 4.5** (Family of diffeomorphisms). *Consider maps $h: \Xi \to \Xi$ and $(\theta, \xi) \in \Theta \times \Xi \mapsto g_\theta(\xi) \in \Xi$ and define the parametric loss $f(\theta, \xi) = h(g_\theta(\xi))$. Assume that these functions are twice differentiable, that $h$ satisfies the second item of Assumption 5 and that, for every $\theta \in \Theta$, $g_\theta$ has a inverse $g_\theta^{-1}$ which is also continuously differentiable in a neighborhood of $\Theta \times \Xi$.*

*As before, this setting fits into the framework above. We now show that Assumption 5 is satisfied.*

1. *Since $h$ is continuous, $h$ satisfies the first item of Assumption 5. It is satisfied by $\mathcal{F}$ as well thanks to $g_\theta^{-1}$ being Lipschitz-continuous in $\xi$ uniformly in $\theta$ by compactness of $\Theta \times \Xi$.*

2. *Take $C$ such that both $g_\theta$ and $g_\theta^{-1}$ are $C$-Lipschitz in $\xi$ uniformly in $\theta$. Since $\arg\max f(\theta, \cdot) = g_\theta^{-1}(\arg\max h)$, it holds that $\min_{\zeta^\star \in \arg\max h} \|g_\theta(\xi) - \zeta^\star\| = \min_{\xi^\star \in \arg\max f(\theta, \cdot)} \|g_\theta(\xi) - g_\theta(\xi^\star)\|$ which lies between $C^{-1} \min_{\xi^\star \in \arg\max f(\theta, \cdot)} \|\xi - \xi^\star\|$ and $C \min_{\xi^\star \in \arg\max f(\theta, \cdot)} \|\xi - \xi^\star\|$. Combined with $h$ satisfying the second item of Assumption 5, this shows that $f$ satisfies this condition as well.*

## 5 Conclusion and perspectives

In this work, we provide generalization guarantees for WDRO models that improve over existing literature in the following aspects: our results avoid the curse of dimensionality, provide exact upper bounds without spurious error terms, and allow for distribution shifts during testing. We obtained these bounds through the development of an original concentration result on the dual of WDRO. Our framework is general enough to cover regularized versions of the WDRO problem: they enjoy similar generalization guarantees as standard WDRO, with less restrictive assumptions.

Our work could be naturally extended in several ways. For instance, it might be possible to relax any of the assumptions (on the sample space, the sampling process, the Wasserstein metric, and the class of functions) at the expense of additional technical work. Moreover, the crucial role played by the radius of the Wasserstein ball calls for a principled and efficient procedure to select it.

## Acknowledgments and Disclosure of Funding

This work has been supported by MIAI Grenoble Alpes (ANR-19-P3IA-0003).

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
