# Appendices

We provide here the proofs of our main results Theorems 3.1, 3.3 and 3.4, along with detailed versions that include explicit bounds. We start, in Section A, by referencing preliminary results, reformulate some of our assumptions, and introduce quantities that appear in the final bounds.

As sketched in Section 3.3, our proof is built on two main parts. The first part is to obtain a concentration bound on the dual problems (12) and (13) by leveraging on a lower bound on the dual multiplier. This concentration result is presented in Section B where we assume that such a lower-bound is given. The second part of the proof then consists in establishing the lower-bound. We have to distinguish two cases: when $\rho$ is small (Section C) and when $\rho$ is close to the critical radius $\rho_c$ (Section D). For the latter, we also need to treat separately the cases where the WDRO problem is regularized or not (respectively Section D.2 and Section D.1): this is where the Assumptions 4 and 5, that are not required in the regularized case, come into play. Putting together these two parts, we obtain our precise theorems in Section E and show how they imply our main results. Section F then complements our theorems to obtain Theorem 3.5. Finally, some variations of known results and technical computations are compiled in Section G as standalone lemmas and, in Section H, we provide numerical illustrations for linear models.

# A Preliminaries

This section presents preliminary results before we start the proofs in Section B. In the first part of this section Section A.1, we present a weaker and more detailed version of Assumption 2, namely Assumption 6, that will suffice for all the proofs in the appendix. We also introduce several quantities that will appear in the final bounds. Then, in Section A.1 we recall the dual problems introduced in Section 3.3 and justify that strong duality holds. Preliminary approximation results on the dual are then given in Section A.3. We then proceed to show the relative compactness of the class $\mathcal{F}$ w.r.t. several metrics in Section A.4. These properties provide a convenient way of ensuring that quantities involving $\mathcal{F}$ are finite, e.g., complexity measures or supremums over $\mathcal{F}$. Finally, we introduce the so-called parametric Morse-Bott condition of Arbel and Mairal (2022) and show how it implies the second item of Assumption 2 in a Riemannian setting.

## A.1 Detailed assumption on the function class and important quantities

Here we present the precise assumptions that we will refer to in the proofs. While Assumptions 1 and 3 are used as presented in the main text, we slightly weaken Assumption 2 to Assumption 6. We also introduce some quantities that we will be of interest for the proofs and the final results.

**Assumption 6** (On the function class). Consider $\mathcal{F}$ a set of real-valued non-negative continuous functions on $\Xi$. We assume that:

- the functions $f \in \mathcal{F}$ are uniformly $L_2$-smooth;

- the gradients are uniformly bounded, i.e.,
$$G := \sup_{f \in \mathcal{F}} \sup_{\xi \in \mathrm{supp}\, \mathrm{P}} \|\nabla f(\xi)\|_2 < +\infty$$

- when $\varepsilon = 0$, the supremum in (11) is finite, i.e.,
$$\widetilde{F}(\lambda) := \sup_{f \in \mathcal{F}} \sup_{\xi \in \mathrm{supp}\, \mathrm{P}} \sup_{\arg\max\{f - \frac{\lambda}{2}\|\xi - \cdot\|^2\}} f < +\infty .$$

Note that the non-negativity assumption is without loss of generality since otherwise, it suffices to consider $\widetilde{\mathcal{F}} := \{f - \min f : f \in \mathcal{F}\}$ and our results are invariant by addition of a constant.

**The blanket assumptions for the remaining of the appendix will be Assumptions 1, 3 and 6.**

The following finite quantities are relevant for the proofs and appear in the quantitative versions of Theorems 3.1, 3.3 and 3.4.

$$C^\star := \sup\left\{ \frac{1}{2}\|\xi - \xi^\star\|^2 : \xi \in \mathrm{supp}\, \mathrm{P},\ f \in \mathcal{F},\ \xi^\star \in \arg\max f \right\},$$

$$C(\sigma) := \sup_{\xi \in \mathrm{supp}\, \mathrm{P}} \mathbb{E}_{\zeta \sim \pi_\sigma(\cdot|\xi)}\left[ \frac{1}{2}\|\xi - \zeta\|^2 \right],$$

$$C_{\mathcal{F}}(\varepsilon, \sigma) := \sup_{f \in \mathrm{conv}(\mathcal{F})} \sup_{\xi \in \mathrm{supp}\, \mathrm{P}} \mathbb{E}_{\zeta \sim \pi_\sigma^{f/\varepsilon}(\cdot|\xi)} \frac{1}{2}\|\xi - \zeta\|^2 ,$$

$$\text{and} \quad \mathrm{Var}(\varepsilon, \sigma) := \sup_{f \in \mathrm{conv}(\mathcal{F})} \sup_{\xi \in \mathrm{supp}\, \mathrm{P}} \sup_{\lambda \geq 0} \mathrm{Var}_{\zeta \sim \pi_\sigma^{\frac{f - \lambda\|\xi - \cdot\|^2/2}{\varepsilon}}(\cdot|\xi)} \frac{1}{2}\|\xi - \zeta\|^2 ,$$

where $\varepsilon > 0$, $\sigma > 0$, and $\pi_\sigma$ is given by (4).

## A.2 Strong duality

As mentioned in Section 3.3, duality plays a central role in our proofs. Let us recall the central notion of dual generator functions, introduced in (11): for any $f \in \mathcal{F}, \xi \in \Xi, \lambda, \varepsilon \geq 0$ and $\sigma > 0$, the dual generator $\phi$ is given as

$$\phi(f, \xi, \lambda, \varepsilon, \sigma) := \begin{cases} \sup_{\zeta \in \Xi}\left\{ f(\zeta) - \frac{\lambda}{2}\|\xi - \zeta\|^2 \right\} & \text{if } \varepsilon = 0 \\ \varepsilon \log\left( \mathbb{E}_{\zeta \sim \pi_\sigma(\cdot|\xi)} \exp\left( \frac{f(\zeta) - \lambda\|\xi - \zeta\|^2/2}{\varepsilon} \right) \right) & \text{if } \varepsilon > 0 . \end{cases}$$

Our proofs are based on the (strong) dual formulations of WDRO, as given by the following lemma that summarizes results of the literature for the regularized and unregularized cases.

**Lemma A.1.** *Under the blanket assumptions, for $f \in \mathcal{F}$, $\rho > 0$, $\varepsilon \geq 0$ and $\sigma > 0$,*

$$\widehat{\mathcal{R}}_{\rho^2}^{\varepsilon}(f) = \inf_{\lambda \geq 0} \lambda \rho^2 + \mathbb{E}_{\xi \sim \widehat{P}_n}\left[\phi(f, \xi, \lambda, \varepsilon, \sigma)\right]$$

$$\text{and} \quad \mathcal{R}_{\rho^2}^{\varepsilon}(f) = \inf_{\lambda \geq 0} \lambda \rho^2 + \mathbb{E}_{\xi \sim P}\left[\phi(f, \xi, \lambda, \varepsilon, \sigma)\right].$$

*Proof.* See Blanchet and Murthy (2019); Gao and Kleywegt (2016) for the unregularized case and Azizian et al. (2023) for the regularized case. □

### A.3   Approximation of the dual generator $\phi$

Important preliminary results for our upcoming concentration bounds (in Section B) are quantitative approximations of the dual generator $\phi$, namely Proposition A.2 and Lemma A.3. In particular, these results also imply bounds on $\phi$ in Corollary A.5.

**Proposition A.2** (Bounding the distance between $\phi$ and $f$)**.** *There are positive constants $\lambda_1$, $\varepsilon_1$, $\sigma_1$, $c_1$, $c_2$ which depend on $G$, $R$ and $d$ such that taking some $\overline{\lambda} \geq \underline{\lambda} \geq \lambda_1 + L_2$, we have for any $f \in \mathcal{F}$, $\xi \in \operatorname{supp} P$, $\lambda \in [\underline{\lambda}, \overline{\lambda}]$, $\varepsilon \in [0, \varepsilon_1]$ and $\sigma \in (0, \sigma_1]$*

$$|\phi(f, \xi, \lambda, \varepsilon, \sigma) - f(\xi)| \leq M(\underline{\lambda}, \overline{\lambda}, \varepsilon, \sigma)$$

*where*

$$M(\underline{\lambda}, \overline{\lambda}, \varepsilon, \sigma) := \frac{1}{2\underline{\lambda}} G^2 + \frac{\varepsilon d}{2} \log\left(\frac{\overline{\lambda}}{\varepsilon} + \frac{1}{\sigma^2}\right) + \varepsilon \log 2 + \varepsilon d |\log \sigma| + \varepsilon c_1 e^{-c_2 \left(\frac{\underline{\lambda} - L_2}{\varepsilon}\right)^{\frac{1}{3}}}.$$

The proof of this result is based on the following second approximation result which gives a precise approximation of $\phi$ that will be used several times in the upcoming proof.

More precisely, we want to approximate $\phi$ by a Taylor development $\overline{\phi}$ defined for any $f \in \mathcal{F}$, $\xi \in \operatorname{supp} P$, $\lambda \geq 0$, $\varepsilon \geq 0$ and $\sigma > 0$ as

$$\overline{\phi}(f, \xi, \lambda, \varepsilon, \sigma) := f(\xi) + \frac{1}{2\left(\lambda + \frac{\varepsilon}{\sigma^2}\right)} \|\nabla f(\xi)\|_2^2 - \frac{\varepsilon d}{2} \log\left(\frac{\lambda}{\varepsilon} + \frac{1}{\sigma^2}\right) + \varepsilon \log \frac{(2\pi)^{\frac{d}{2}}}{Z(\xi, \sigma)} \quad (19)$$

where $Z(\xi, \sigma) := \int_\Xi e^{-\frac{\|\xi - \zeta\|_2^2}{2\sigma^2}} \mathrm{d}\zeta$. The distance between $\phi$ and $\overline{\phi}$ is then controlled by the following Laplace approximation lemma.

**Lemma A.3** (Approximation of $\phi$)**.** *There are positive constants $\lambda_1, \varepsilon_1, \sigma_1, c_1, c_2$ which depend on $G$, $R$ and $d$ such that $\varepsilon_1 \leq \lambda_1$ and, when $\varepsilon \in [0, \varepsilon_1]$, $\sigma \in (0, \sigma_1]$ and $\lambda \geq \lambda_1 + L_2$, we have for any $f \in \mathcal{F}$, $\xi \in \operatorname{supp} P$*

$$\overline{\phi}(f, \xi, \lambda + L_2, \varepsilon, \sigma) - \varepsilon c_1 e^{-c_2\left(\frac{\lambda + L_2}{\varepsilon}\right)^{\frac{1}{3}}} \leq \phi(f, \xi, \lambda, \varepsilon, \sigma) \leq \overline{\phi}(f, \xi, \lambda - L_2, \varepsilon, \sigma) + \varepsilon c_1 e^{-c_2\left(\frac{\lambda - L_2}{\varepsilon}\right)^{\frac{1}{3}}}.$$

*Proof.* Fix $f \in \mathcal{F}$, $\xi \in \operatorname{supp} P$, $\lambda \geq 0$, $\varepsilon \geq 0$ and $\sigma > 0$. To bound the error between $\phi$ and its approximation $\overline{\phi}$, we introduce an intermediate approximation $\widetilde{\phi}$ defined as

$$\widetilde{\phi}(f, \xi, \lambda, \varepsilon, \sigma) := \begin{cases} \varepsilon \log\left(\mathbb{E}_{\zeta \sim \pi_\sigma(\cdot|\xi)} \exp\left(\frac{f(\xi) + \langle \nabla f(\xi), \zeta - \xi \rangle - \lambda \|\xi - \zeta\|^2/2}{\varepsilon}\right)\right) & \text{if } \varepsilon > 0 \\ \sup_{\zeta \in \Xi}[f(\xi) + \langle \nabla f(\xi), \zeta - \xi \rangle - \frac{\lambda}{2}\|\xi - \zeta\|^2] & \text{if } \varepsilon = 0, \end{cases}$$

which corresponds to $\phi$ applied to the Taylor approximation of $f$ at $\xi$ (instead of $f$ itself). By smoothness of the functions in $\mathcal{F}$ (Assumption 6), we readily have that,

$$\widetilde{\phi}(f, \xi, \lambda + L_2, \varepsilon, \sigma) \leq \phi(f, \xi, \lambda, \varepsilon, \sigma) \leq \widetilde{\phi}(f, \xi, \lambda - L_2, \varepsilon, \sigma).$$

Now, all that is left to bound, is the error between $\widetilde{\phi}$ and $\overline{\phi}$. Consider first the case where $\varepsilon > 0$ and let us rewrite $\widetilde{\phi}$ by using the definition of $\pi_\sigma$:

$$\widetilde{\phi}(f, \xi, \lambda, \varepsilon, \sigma) = \varepsilon \log\left(\mathbb{E}_{\zeta \sim \pi_\sigma(\cdot|\xi)} \exp\left(\frac{f(\xi) + \langle \nabla f(\xi), \zeta - \xi \rangle - \lambda \|\xi - \zeta\|^2/2}{\varepsilon}\right)\right)$$

$$= \varepsilon \log\left(\int_\Xi \exp\left(\frac{1}{\varepsilon}\left(f(\xi) + \langle \nabla f(\xi), \zeta - \xi \rangle - \left(\lambda + \frac{\varepsilon}{\sigma^2}\right)\frac{1}{2}\|\xi - \zeta\|^2\right)\right)\mathrm{d}\zeta\right) - \varepsilon \log Z(\xi, \sigma).$$

But, looking at the inner expression, we have that

$$f(\xi) + \langle \nabla f(\xi), \zeta - \xi \rangle - \left( \lambda + \frac{\sigma^2}{\varepsilon} \right) \frac{1}{2} \| \xi - \zeta \|^2 = f(\xi) + \frac{1}{2(\lambda + \frac{\sigma^2}{\varepsilon})} \| \nabla f(\xi) \|_2^2 - \frac{1}{2\tau} \| \zeta - \zeta^\star(\tau) \|_2^2,$$

(20)

where we defined $\frac{1}{\tau} := \lambda + \frac{\varepsilon}{\sigma^2}$ and $\zeta^\star(\tau) = \xi + \tau \nabla f(\xi)$. Hence,

$$\widetilde{\phi}(f, \xi, \lambda, \varepsilon, \sigma) = \overline{\phi}(f, \xi, \lambda, \varepsilon, \sigma) + \varepsilon \log \left( \int_\Xi \exp \left( -\frac{1}{2\varepsilon\tau} \| \zeta - \zeta^\star(\tau) \|_2^2 \right) d\zeta \right) - \frac{\varepsilon d}{2} \log(2\pi\varepsilon\tau) \,.$$

Define $\lambda_1' := \frac{\sqrt{6}G}{R}$ and $\tau_1 := \frac{1}{\lambda_1'} = \frac{R}{2G}$ so that $\lambda \geq \lambda_1'$ implies that $\tau \leq \frac{1}{\lambda_1'} = \tau_1$. Let us now check that the conditions of Lemma G.1 are satisfied.

1. Since $\zeta^\star(0) = \xi$, by Assumption 3, $\overline{\mathbb{B}}(\zeta^\star(0), R)$ is contained in $\Xi$.

2. For $\tau \leq \tau_1$, we have that $\tau^2 \| \nabla f(\xi) \|_2^2 \leq \frac{G^2}{(\lambda_1')^2} = \frac{R^2}{6}$ by definition.

Hence, we can apply Lemma G.1 to get that, for any $\lambda \geq \lambda_1'$

$$\varepsilon \log \left( 1 - 6^{d/2} e^{-\frac{R^2}{12\varepsilon\tau}} \right) \leq \widetilde{\phi}(f, \xi, \lambda, \varepsilon, \sigma) - \overline{\phi}(f, \xi, \lambda, \varepsilon, \sigma) \leq \varepsilon \log \left( 1 + 6^{d/2} e^{-\frac{R^2}{12\varepsilon\tau}} \right) \,.$$

Now, using Lemma G.5, we get that there are positive constants $\varepsilon_1$, $\lambda_1$, $c_1, c_2$ depending on $R$, $G$ and $d$ such that, if $\varepsilon \leq \varepsilon_1$ and $\lambda \geq \lambda_1$, then

$$|\widetilde{\phi}(f, \xi, \lambda, \varepsilon, \sigma) - \overline{\phi}(f, \xi, \lambda, \varepsilon, \sigma)| \leq \varepsilon c_1 e^{-c_2 \left( \frac{\lambda}{\varepsilon} \right)^{\frac{1}{3}}} \,.$$

Moreover, $\varepsilon_1$ can be reduced so that it is less than $\lambda_1$ if it is not the case originally.

To finish the proof, let us now come back to the case $\varepsilon = 0$. First, note that (20) is still valid even with $\varepsilon = 0$ so that we have

$$\widetilde{\phi}(f, \xi, \lambda, 0) = \overline{\phi}(f, \xi, \lambda, 0) - \frac{1}{2\tau} \inf_{\zeta \in \Xi} \| \zeta - \zeta^\star(\tau) \|_2^2 \,.$$

But as seen above, for $\tau = \lambda^{-1} \leq \tau_1$, $\zeta^\star(\tau)$ is inside $\overline{\mathbb{B}}(\zeta^\star(0), R)$ so that $\widetilde{\phi}(f, \xi, \lambda, 0) = \overline{\phi}(f, \xi, \lambda, 0)$.

We conclude the proof by noticing that the obtained bounds are valid for any $0 < \varepsilon \leq \varepsilon_1$, $\lambda \geq \lambda_1 + L_2$, $f \in \mathcal{F}$ and $\xi \in \operatorname{supp} \mathrm{P}$. $\qquad\square$

The following lemma is needed for the proof of Proposition A.2.

**Lemma A.4.** *There is a positive constant $\sigma_1 > 0$ which depends on $R$ and $d$ such that, for $\sigma \in (0, \sigma_1]$ and $\xi \in \operatorname{supp} \mathrm{P}$,*

$$\left| \log \frac{Z(\xi, \sigma)}{(2\pi)^{d/2}} \right| \leq d |\log \sigma| + \log 2 \,.$$

*Proof.* It suffices to show that

$$\frac{(2\pi\sigma^2)^{d/2}}{2} \leq Z(\xi, \sigma) \leq (2\pi\sigma^2)^{d/2} \,,$$

(21)

for any $\sigma \in (0, \sigma_1]$ with some $\sigma_1 > 0$ suitably defined. We prove the right-hand side (RHS) by removing the constraint $\Xi$ in the integral defining $Z(\xi, \sigma)$:

$$Z(\xi, \sigma) = \int_\Xi e^{-\frac{\| \xi - \zeta \|^2}{2\sigma^2}} d\zeta \leq \int_{\mathbb{R}^d} e^{-\frac{\| \xi - \zeta \|^2}{2\sigma^2}} d\zeta = (2\pi\sigma^2)^{d/2} \,.$$

For the left-hand side (LHS), we invoke Lemma G.1 using Assumption 3 to get that $Z(\xi, \sigma) \geq \sigma^d$ when $\sigma \leq \sigma_1$ with $\sigma_1 > 0$ satisfying

$$1 - 6^{d/2} e^{-\frac{R^2}{12\sigma_1^2}} \geq \frac{1}{2} \,.$$

$\qquad\square$

We are now in a position to prove the main result of the section.

*Proof of Proposition A.2.* Applying Lemmas A.3 and A.4 and using the definition of $G$ readily gives us that

$$|\phi(f,\xi,\lambda,\varepsilon,\sigma) - f(\xi)| \leq \frac{1}{2\lambda}G^2 + \frac{\varepsilon d}{2}\left|\log\left(\frac{\lambda}{\varepsilon} + \frac{1}{\sigma^2}\right)\right| + \varepsilon(d|\log\sigma| + \log 2) + \varepsilon\overline{Z}(\sigma) + \varepsilon c_1 e^{-c_2\left(\frac{\lambda-L_2}{\varepsilon}\right)^{\frac{1}{3}}}.$$

Since $\lambda$ is always greater or equal than $\varepsilon$, $\log\left(\frac{\lambda}{\varepsilon} + \frac{1}{\sigma^2}\right)$ is always non-negative and, with $\lambda$ belonging to $[\underline{\lambda}, \overline{\lambda}]$, we get that

$$|\phi(f,\xi,\lambda,\varepsilon,\sigma) - f(\xi)| \leq \frac{1}{2\underline{\lambda}}G^2 + \frac{\varepsilon d}{2}\log\left(\frac{\overline{\lambda}}{\varepsilon} + \frac{1}{\sigma^2}\right) + \varepsilon(d|\log\sigma| + \log 2) + \varepsilon c_1 e^{-c_2\left(\frac{\underline{\lambda}-L_2}{\varepsilon}\right)^{\frac{1}{3}}},$$

which is the desired result. $\qquad\square$

As a consequence of this result, we have the following bound on the dual generator.

**Corollary A.5.** *For any $f \in \mathcal{F}$, $\xi \in \operatorname{supp} P$, $\lambda \in [\underline{\lambda}, \overline{\lambda}]$, $\varepsilon \geq 0$ and $\sigma > 0$, the bound*

$$-a(\underline{\lambda}, \overline{\lambda}, \varepsilon, \sigma) \leq \phi(f,\xi,\lambda,\varepsilon,\sigma) \leq \widetilde{F}(\underline{\lambda}),$$

*holds where*

$$a(\underline{\lambda}, \overline{\lambda}, \varepsilon, \sigma) := \begin{cases} 0 & \text{when } \varepsilon = 0 \\ M(\underline{\lambda}, \overline{\lambda}, \varepsilon, \sigma) & \text{when } 0 < \varepsilon \leq \varepsilon_1,\ 0 < \sigma \leq \sigma_1,\ \underline{\lambda} \geq \lambda_1 + L_2 \\ G\sqrt{2C(\sigma)} + \left(L_2 + \overline{\lambda}\right)C(\sigma) & \text{otherwise.} \end{cases}$$

*with $M(\underline{\lambda}, \overline{\lambda}, \varepsilon, \sigma)$ the bounding term appearing in Proposition A.2, as well as $\varepsilon_1$, $\sigma_1$, $\lambda_1$.*

*Proof.* For the upper-bound, it suffices to note that

$$\phi(f,\xi,\lambda,\varepsilon,\sigma) \leq \phi(f,\xi,\lambda,0) \leq \phi(f,\xi,\underline{\lambda},0) \leq \widetilde{F}(\underline{\lambda})$$

by definition of $\widetilde{F}(\underline{\lambda})$. Let us now turn to the lower bound.

When $\varepsilon = 0$, we have that $\phi(f,\xi,\lambda,\varepsilon,\sigma) \geq f(\xi) \geq 0$.

When $0 < \varepsilon \leq \varepsilon_1$ and $\underline{\lambda} \geq \lambda_1 + L_2$, we have from Proposition A.2

$$\phi(f,\xi,\lambda,\varepsilon,\sigma) \geq f(\xi) - M(\underline{\lambda}, \overline{\lambda}, \varepsilon, \sigma) \geq -M(\underline{\lambda}, \overline{\lambda}, \varepsilon, \sigma).$$

Otherwise, the bound comes from the smoothness of $f$ and Jensen's inequality as

$$\phi(f,\xi,\lambda,\varepsilon,\sigma) \geq \varepsilon\log\left(\mathbb{E}_{\zeta\sim\pi_\sigma(\cdot|\xi)}\exp\left(\frac{f(\xi) + \langle\nabla f(\xi),\zeta-\xi\rangle - (L_2+\lambda)\|\xi-\zeta\|^2/2}{\varepsilon}\right)\right)$$

$$\geq \mathbb{E}_{\zeta\sim\pi_\sigma(\cdot|\xi)}\left[f(\xi) + \langle\nabla f(\xi),\zeta-\xi\rangle - (L_2+\lambda)\frac{1}{2}\|\xi-\zeta\|^2\right]$$

$$\geq -\left(G\sqrt{2C(\sigma)} + (L_2+\lambda)C(\sigma)\right) \geq -\left(G\sqrt{2C(\sigma)} + \left(L_2+\overline{\lambda}\right)C(\sigma)\right).$$

$\qquad\square$

## A.4 Relative compactness of the class $\mathcal{F}$ of loss functions

In this section we prove the relative compactness of the class $\mathcal{F}$ w.r.t. several metrics. First, we show in Lemma A.6 that, under our blanket assumptions, $\mathcal{F}$ is relatively compact for for the infinity norm over $\Xi$, defined by i.e., $\|f\|_\infty := \sup_{\xi\in\Xi}|f(\xi)|$. Then, in Lemma A.7, we establish the equivalence between the first item of Assumption 5 and the relative compactness of $\mathcal{F}$ w.r.t. another distance that we introduce, as mentioned below Assumption 5 in Section 3.1. Finally, we leverage these compactness properties to ensure that the Dudley integral of $\mathcal{F}$ w.r.t. those metrics, a standard complexity measure in concentration theory, is finite in Lemma A.10.

**Lemma A.6.** $\mathcal{F}$ *and* $\mathrm{conv}(\mathcal{F})$ *are relatively compact for the topology of the uniform convergence.*

*Proof.* First, the functions of $\mathcal{F}$ are uniformly Lipschitz-continuous: fix $\xi \in \mathrm{supp}\, \mathrm{P}$, then, for any $\zeta \in \Xi\ \|\nabla f(\zeta)\| \leq L_2 \|\xi - \zeta\| + G \leq L_2 \sup_{(\xi,\zeta)\in\Xi} \|\xi - \zeta\| + G$ which is finite by compactness. Using the compactness of $\Xi$ again, the functions in $\mathcal{F}$ are also uniformly bounded. As a consequence, the functions in $\mathrm{conv}(\mathcal{F})$ are also uniformly Lipschitz-continuous and uniformly bounded. By the Arzelà-Ascoli theorem, see e.g., (Rudin, 1987, Thm. 11.28), $\mathcal{F}$ and $\mathrm{conv}(\mathcal{F})$ are then relatively compact for the topology of uniform convergence. $\qquad\square$

Recall that, for a set $A \subset \Xi$ and a point $\xi \in \Xi$, we denote by $d(\xi, A)$ the distance between $\xi$ and $A$, i.e., $d(\xi, A) = \inf_{\zeta \in A} \|\xi - \zeta\|$.

**Lemma A.7.** *Consider the distance, defined on continuous functions on $\Xi$ by*

$$D(f, g) := \|f - g\|_\infty + D_H(\arg\max f, \arg\max g)$$

*where $D_H$ denotes the Hausdorff distance between sets associated to d, i.e., for $A, B \subset \Xi$,*

$$D_H(A, B) := \max\left(\sup_{\xi \in A} d(\xi, A), \sup_{\xi \in B} d(\xi, B)\right).$$

*Under the blanket assumptions, we have that Item 1 of Assumption 5, i.e., that for any $R > 0$, there exists $\Delta > 0$ such that,*

$$\forall f \in \mathcal{F}, \forall \zeta \in \Xi, d(\zeta, \arg\max f) \geq R \implies f(\zeta) - \max f \leq -\Delta, \tag{22}$$

*is equivalent to $\mathcal{F}$ being relatively compact for $D$.*

*Proof.* ( $\implies$ ) Let us begin by showing that (22) implies the relative compactness of $\mathcal{F}$ for $D$, i.e., that the adherence of $\mathcal{F}$ is compact for $D$.

Take $(f_t)_{t=1,2,...}$ a sequence of functions from $\mathcal{F}$, and we will show that there is a subsequence which converges to some function in $\mathcal{F}$ for $D$. By compactness of $\mathcal{F}$ for the infinity norm, Lemma A.6, there readily is a subsequence of $(f_t)_{t=1,2,...}$ that converges uniformly to some continuous function $f : \Xi \to \mathbb{R}$. Without loss of generality, let us assume that the whole sequence $(f_t)_{t=1,2,...}$ converges uniformly to $f$, i.e., that $\|f - f_t\|_\infty \to 0$ as $t \to +\infty$. As a consequence, it holds also holds that $\max_\Xi f_t$ converges to $\max_\Xi f$.

We now show that $D_H(\arg\max f_t, \arg\max f)$ converges to 0. $\mathcal{F}$ satisfy (22) by assumption. Hence, for any fixed $\eta > 0$, we can invoke (22) with $R \leftarrow \eta$ and it gives us some $\Delta > 0$. Now, since $f$ is continuous, $\{\zeta \in \Xi : d(\zeta, \arg\max f) \geq \eta\}$ is a closed set inside a compact and therefore is compact as well. Hence, $f$ reaches its maximum over this set and it is strictly less than $\max_\Xi f$ by construction. Substituting $\Delta$ with $\min(\Delta, \max_\Xi f - \max\{f(\zeta) : \zeta \in \Xi, d(\zeta, \arg\max f) \geq \eta\})$ which is still positive, we get that, for any $\zeta \in \Xi$, both,

$$d(\zeta, \arg\max f) \geq \eta \implies f(\zeta) - \max f \leq -\Delta,$$

and, for any $t = 1, 2, \ldots,$

$$d(\zeta, \arg\max f_t) \geq \eta \implies f_t(\zeta) - \max f_t \leq -\Delta.$$

By convergence of the sequence, as mentioned above, there is some $T \geq 1$ such that, for any $t \geq T$, $\|f - f_t\|_\infty \leq \Delta/3$ and $|\max_\Xi f_t - \max_\Xi f| \leq \Delta/3$. These two inequalities imply that, for any $\xi \in \arg\max f$,

$$\max_\Xi f_t - f_t(\xi) \leq \max_\Xi f + \frac{\Delta}{3} - f(\xi) + \frac{\Delta}{3} = \frac{2\Delta}{3}.$$

Therefore, by definition of $\Delta$, it holds that $d(\xi, \arg\max f_t) < \eta$. Similarly, when $\xi \in \arg\max f_t$, one shows that $\max_\Xi f - f(\xi) \leq 2\Delta/3$ so that we have $d(\xi, \arg\max f) < \eta$ as well. Hence, for any $t \geq T$, $D_H(\arg\max f_t, \arg\max f)$ is at most $\eta$.

Therefore, we have shown that $D_H(\arg\max f_t, \arg\max f)$ goes to zero. Since $\|f - f_t\|_\infty$ converges to zero as well by construction, this means that $D(f_t, f)$ converges to zero, which concludes the proof.

$(\Longleftarrow)$ Let us proceed by contradiction, i.e., assume that there is some $R > 0$, some sequence $(f_t)_{t=1,2,\ldots}$ of functions from $\mathcal{F}$ and some sequence $(\xi_t)_{t=1,2,\ldots}$ of points from $\Xi$ such that,

$$\forall t = 1, 2, \ldots, d(\arg\max f_t, \xi_t) \geq R \quad \text{yet} \quad f_t(\xi_t) - \max_{\Xi} f_t \to 0 \text{ as } t \to +\infty\,.$$

Since $\Xi$ is compact and since we assume $\mathcal{F}$ to be relatively compact for $D$, without loss of generality, we can assume that $(\xi_t)_{t=1,2,\ldots}$ converges to some $\xi \in \Xi$ while $(f_t)_{t=1,2,\ldots}$ converges to some continuous function $f$ for $D$. On the one hand, by definition of the Hausdorff distance, we have that, for any $t = 1, 2, \ldots$,

$$\begin{aligned}
d(\arg\max f, \xi) &\geq d(\arg\max f_t, \xi) - D_H(\arg\max f_t, \arg\max f) \\
&\geq d(\arg\max f_t, \xi_t) - (d(\xi, \xi_t) + D_H(\arg\max f_t, \arg\max f))\,,
\end{aligned}$$

so that, by taking $t \to +\infty$, we get that $d(\arg\max f, \xi) \geq R$. On the other hand, by uniform convergence, one has that

$$f(\xi) - \max f = \lim_{t \to +\infty} f_t(\xi_t) - \max f_t = 0\,,$$

which yields the contradiction since $\xi$ cannot belong to $\arg\max f$.

$\square$

Note that, for parametric models (Section 4), this lemma gives a computation-free approach to verifying the second item of Assumption 5.

**Corollary A.8.** *Consider $\Theta$ a compact subset of $\mathbb{R}^p$ and $f : \Theta \times \Xi \to \mathbb{R}$ a continuous function. If the map $\theta \in \Theta \mapsto f(\theta, \cdot)$ is continuous from $\Theta$ to the space of continuous functions on $\Xi$ equipped with the distance $D$ defined in Lemma A.7, then $\mathcal{F} := \{f(\theta, \cdot) : \theta \in \Theta\}$ is compact for $D$.*

In particular, this corollary allows one to easily check that Examples 4.1 and 4.2 satisfy the second item of Assumption 5.

We finally introduce Dudley's integral, which is a standard complexity measure in concentration theory.

**Definition A.9.** *Dudley's entropy integral $\mathcal{I}(\mathcal{X}, \mathrm{dist})$ is defined for a metric space $(\mathcal{X}, \mathrm{dist})$ as*

$$\mathcal{I}(\mathcal{X}, \mathrm{dist}) := \int_0^{+\infty} \sqrt{\log N(t, \mathcal{X}, \mathrm{dist})} \mathrm{d}t$$

*where $N(t, \mathcal{X}, \mathrm{dist})$ denotes the $t$-packing number of $\mathcal{X}$, which is the maximal number of points in $\mathcal{X}$ which are at least at a distance $t$ from each other.*

**Lemma A.10.** *The Dudley integral of $\mathcal{F}$ w.r.t. $\|\cdot\|_\infty$, that we denote by $\mathcal{I}(\mathcal{F}, \|\cdot\|_\infty)$, is finite. Under Assumption 5, the Dudley integral of $\mathcal{F}$ w.r.t. $D$, denoted by $\mathcal{I}(\mathcal{F}, D)$ is finite as well.*

*Proof.* Lemma A.6 shows that $\mathcal{F}$ is relatively compact for the norm $\|\cdot\|_\infty$ and in particular bounded. Since Dudley's entropy integral is finite for balls (Wainwright, 2019, Ex. 5.18) and $\mathcal{F}$ is now included in some ball for $\|\cdot\|_\infty$, the integral $\mathcal{I}(\mathcal{F}, \|\cdot\|_\infty)$ is indeed finite. The second assertion is proven using the same reasoning and Lemma A.7. $\square$

### A.5 Parametric Morse-Bott objectives

In this section, we discuss the quadratic growth condition of the second item of Assumption 5 and its relation to the parametric Morse-Bott assumption of Arbel and Mairal (2022). Indeed, in the context of smooth manifolds and parametric models, we prove that the parametric Morse-Bott assumption implies the quadratic growth condition of Assumption 5. In Assumption 7, we introduce the Riemannian and parametric settings that are necessary to formulate the parametric Morse-Bott condition and we then present a version of this condition adapted to our context. We refer to Lee (2018) for definitions relevant to Riemannian geometry. The main result of this section is then Proposition A.11, which relies on Lemma A.12 for its proof.

**Assumption 7** (Parametric Morse-Bott). Let $\mathcal{F} = \{\xi \mapsto f(\theta, \xi) : \theta \in \Theta\}$ where :

- $\Xi$, $\Theta$ are smooth compact (connected embedded) submanifolds of $\mathbb{R}^d$ and $\mathbb{R}^p$ respectively, endowed with the induced Euclidean metric.

- $f : \Theta \times \Xi \to \mathbb{R}$ is thrice continuously differentiable on the product manifold.

- $f$ is a parametric Morse-Bott function (Arbel and Mairal, 2022, Def. 2): the set of augmented critical points of $\mathcal{F}$, defined as

$$M := \{(\theta, \xi) \in \Theta \times \Xi : \mathrm{grad}_\xi f(\theta, \xi) = 0\}.$$

  is a smooth (embedded) submanifold of $\Theta \times \Xi \setminus \mathrm{bd}\,\Xi$ whose dimension at $(\theta, \xi) \in M$ is $\dim_\theta(\Theta) + \dim(\ker \mathrm{Hess}_\xi f(\theta, \xi))$.

Under this assumption Assumption 7, the following result thus guarantees that the quadratic growth condition of Assumption 5 holds.

**Proposition A.11.** *Under Assumption 7 and the first item of Assumption 5, the second item of Assumption 5 holds, i.e., there exists $\mu, L_3 > 0$ such that, for all $\theta \in \Theta$, $\xi \in \Xi$ and $\xi^* \in \arg\max f$ a projection of $\xi$ on $\arg\max f$, i.e., $\xi^* \in \arg\min_{\arg\max f}\|\xi - \cdot\|$, it holds that*

$$f(\theta, \xi^*) \geq f(\theta, \xi) + \frac{\mu}{2}\|\xi - \xi^*\|^2 - \frac{L_3}{6}\|\xi - \xi^*\|^3.$$

To show this result, we rely on the following lemma that relates Assumption 7 to a local quadratic growth condition.

**Lemma A.12.** *Under Assumption 7, for any $(\theta_0, \xi_0) \in M$ such that $\xi_0$ is a local maximum of $f(\theta_0, \cdot)$ and any neighborhood $\mathcal{W}$ of $(\theta_0, \xi_0)$ in $M$, there exists a neighborhood $\mathcal{U}$ of $(\theta_0, \xi_0)$ in $\Theta \times \xi$ and $\mu > 0$ such that, for any $(\theta, \xi) \in \mathcal{U}$, there exists $\xi^* \in \Xi$ such that $(\theta, \xi^*) \in \mathcal{W}$ and*

$$f(\theta, \xi^*) \geq f(\theta, \xi) + \frac{\mu}{2}\|\xi - \xi^*\|^2.$$

*Proof.* By assumption, the tangent space of $M$ at $(\theta, \xi)$ is given by

$$\mathrm{T}_{(\theta, \xi)}\,M = \mathrm{T}_\theta\,\Theta \times \ker \mathrm{Hess}_\xi f(\theta, \xi),$$

and so its normal space (in $\Theta \times \Xi$) is equal to

$$\mathrm{N}_{(\theta, \xi)}\,M = \{0\} \times (\ker \mathrm{Hess}_\xi f(\theta, \xi))^\perp \subset \mathrm{T}_\theta\,\Theta \times \mathrm{T}_\xi\,\Xi.$$

Applying the inverse function theorem to the normal exponential map $E : (\theta, \xi, (0, z)) \in NM \mapsto (\theta, \exp_\xi(z))$ following the proof of Lee (2018, Thm. 5.25), there exists $\eta > 0$ and a neighborhood $\mathcal{U}_\eta$ of $M$ in $\Theta \times \xi$ such that, with

$$\mathcal{V}_\eta := \left\{(\theta, \xi, z) : (\theta, \xi) \in M, z \in (\ker \mathrm{Hess}_\xi f(\theta, \xi))^\perp, \|z\| < \eta\right\},$$

the normal exponential map $E$ is a diffeomorphism from $\mathcal{V}_\eta$ to $\mathcal{U}_\eta$. Note that $\mathcal{V}_\eta$ is relatively compact and, as a consequence, the third derivative of $t \in [0, 1] \mapsto f(\theta, \exp_\xi(tz))$ is a continuous function of $t \in [0, 1]$ and $(\theta, \xi, z) \in \mathcal{V}_\eta$ and as a consequence is bounded uniformly by some constant $L_3 > 0$. Fix $(\theta_0, \xi_0) \in M$ such that $\xi_0$ is a local maximum of $f(\theta, \cdot)$. Consider the map

$$\nu : (\theta, \xi) \mapsto \inf\left\{\langle z, -\mathrm{Hess}_\xi f(\theta, \xi)z\rangle : z \in (\ker \mathrm{Hess}\, f(\theta, \xi))^\perp, \|z\| = 1\right\}.$$

If $\nu(\theta_0, \xi_0)$ is $+\infty$, i.e., if $\ker \mathrm{Hess}\, f(\theta_0, \xi_0)$ is equal to the whole $\mathrm{T}_{\xi_0}\,\Xi$, then, since the dimension of a manifold is locally constant, there is a neighborhood of $(\theta_0, \xi_0)$ in $\Theta \times \Xi$ on which $\nu$ is identically equal to $+\infty$. Otherwise, if $\nu(\theta_0, \xi_0)$ is finite, then it is positive by construction. Hence, the continuity of $\nu$ implies there is a positive constant $\mu > 0$ and a neighborhood of $(\theta_0, \xi_0)$ in $\Theta \times \Xi$ on which $\nu$ is lower-bounded by $\mu$.

Hence, in both cases, there is $\mu > 0$ and $\mathcal{V}'$ a neighborhood of $(\theta_0, \xi_0)$ in $\Theta \times \Xi$ such that $\nu$ is at least greater or equal to $\mu$ on $\mathcal{V}'$. Finally, take

$$\mathcal{U} := \mathcal{U}_\eta \cap E\left(\left\{(\theta, \xi, z) \in \mathcal{V}_\eta : (\theta, \xi) \in \mathcal{V}' \cap \mathcal{W}, \|z\| < \frac{3\mu}{4L_3}\right\}\right).$$

We are now in a position to prove the result. Take $(\theta, \xi) \in \mathcal{U}$. Since $\mathcal{U}$ is included in $\mathcal{U}_\eta$, there is some $\xi^* \in \Xi$ and $z \in (\ker \operatorname{Hess} f(\theta, \xi))^\perp$ such that $(\theta, \xi^*) \in M \cap \mathcal{V}' \cap \mathcal{W}$, $\|z\| < \frac{3\mu}{4L_3}$ and $\exp_{\xi^*}(z) = \xi$. Let $\gamma(t) := \exp_{\xi^*}(tz)$ for $t \in [0, 1]$ denotes the geodesic curve going from $\xi^*$ to $\xi$. Then, by the Taylor inequality applied to $t \mapsto f(\xi, \gamma(t))$ (see Boumal (2023, § 5.9)) and by definition of $L_3$,

$$f(\theta, \xi) \leq f(\theta, \xi^*) + \langle \operatorname{grad}_\xi f(\theta, \xi^*), z \rangle + \frac{1}{2} \langle \operatorname{Hess}_\xi f(\theta, \xi^*) z, z \rangle$$
$$+ \frac{1}{2} \langle \operatorname{grad}_\xi f(\theta, \xi^*), \gamma''(0) \rangle + \frac{L_3}{6} \|z\|^3 .$$

But $\gamma''(t)$ is null since $\gamma$ is a geodesic and $\operatorname{grad}_\xi f(\theta, \xi^*)$ too by definition. Moreover, since $(\theta, \xi^*) \in \mathcal{V}'$ and $z \in (\ker \operatorname{Hess}_\xi f(\theta, \xi^*))^\perp$, the term $\langle \operatorname{Hess}_\xi f(\theta, \xi^*) z, z \rangle$ is bounded by $-\mu \|z\|^2$. But $\|z\|$ is also equal to $\|\xi - \xi^*\|$ by definition of $z$ so we get,

$$f(\theta, \xi) \leq f(\theta, \xi^*) - \frac{\mu}{2} \|\xi - \xi^*\|^2 + \frac{L_3}{6} \|\xi - \xi^*\|^3$$
$$\leq f(\theta, \xi^*) - \frac{\mu}{4} \|\xi - \xi^*\|^2 ,$$

since $\|z\| = \|\xi - \xi^*\| \leq \frac{3\mu}{4L_3}$, which gives the result. $\square$

We are now ready to prove Proposition A.11.

*Proof of Proposition A.11.* We build upon the result of Lemma A.12. Fix $(\theta_0, \xi_0) \in M$ such that $\xi_0$ is a maximum of $f(\theta_0, \cdot)$ and let $r > 0$ such that $\overline{\mathbb{B}}((\theta_0, \xi_0), r) \cap M$ is diffeomorphic to an Euclidean ball. Invoke the first item of Assumption 5, with $R \leftarrow r/2$ and let $\Delta > 0$ be the given positive quantity. Let $\mathcal{U}, \mu$ be given by Lemma A.12 invoked with $\mathcal{W} := \overline{\mathbb{B}}((\theta_0, \xi_0), \frac{r}{2}) \cap \{(\theta, \xi) \in M : f(\theta, \xi) > \max_\Xi f(\theta, \cdot) - \Delta\}$.

Hence, for any $(\theta, \xi) \in \mathcal{U}$, there is $\xi^* \in \Xi$ such that $(\theta, \xi^*) \in M \cap \mathcal{W}$ and

$$f(\theta, \xi^*) \geq f(\theta, \xi) + \frac{\mu}{2} \|\xi - \xi^*\|^2 . \tag{23}$$

But $(\theta, \xi^*)$ also satisfies $f(\theta, \xi^*) > \max_\Xi f(\theta, \cdot) - \Delta$ so that $d(\xi^*, \arg\max_\Xi f(\theta, \cdot)) < \frac{r}{2}$ by definition of $\Delta$, i.e., there exists $\xi^{**}$ that is a maximizer of $f(\theta, \cdot)$ and that is at distance at most $< \frac{r}{2}$ from $\xi^*$. But then both $\xi^*$ and $\xi^{**}$ belong to $\overline{\mathbb{B}}((\theta_0, \xi_0), r)$ that is diffeomorphic to an Euclidean ball. Hence, since the derivative of $f(\theta, \cdot)$ is null on $M$, $f(\theta, \xi^*) = f(\theta, \xi^{**}) = \max_\Xi f(\theta, \cdot)$ so that $\xi^*$ is a maximizer of $f(\theta, \cdot)$ too. Therefore, (23) becomes

$$\max_\Xi f(\theta, \cdot) = f(\theta, \xi^*) \geq f(\theta, \xi) + \frac{\mu}{2} \|\xi - \xi^*\|^2 \geq f(\theta, \xi) + \frac{\mu}{2} d^2 \left( \xi, \arg\max_\Xi f(\theta, \cdot) \right) .$$

The final statement of the proposition follows by compactness and uniform Lipschitz-continuity of $\mathcal{F}$ (see the proof of Lemma A.6). $\square$

# B  From empirical to true risk via duality

In this part of the proof, our objective is to show that: if the dual variable $\lambda$ in (12) can be bounded uniformly in $[\underline{\lambda}, \overline{\lambda}]$ with probability $1 - \delta$, then we can concentrate the empirical expectation in (12) towards the one in (13). The concentration error induces a loss in the radius, fortunately, captured by the variable $\overline{\rho}_n^2(\delta, \underline{\lambda}, \overline{\lambda}, \varepsilon, \sigma)$ that we take as

$$\overline{\rho}_n^2(\delta, \underline{\lambda}, \overline{\lambda}, \varepsilon, \sigma) := \frac{117}{\sqrt{n}\underline{\lambda}} \left( \mathcal{I}(\mathcal{F}, \|\cdot\|_\infty) + \max\left( \widetilde{F}(\underline{\lambda}), a(\underline{\lambda}, \overline{\lambda}, \varepsilon, \sigma) \right) \left( 1 + \sqrt{\log \frac{1}{\delta}} \right) \right), \tag{24}$$

where $\mathcal{I}(\mathcal{F}, \|\cdot\|_\infty)$ is the Dudley integral of $\mathcal{F}$ w.r.t. the infinity norm (Definition A.9), $\widetilde{F}$ is defined in Assumption 6, and $a(\underline{\lambda}, \overline{\lambda}, \varepsilon, \sigma)$ is the bounding term appearing in Corollary A.5.

The main result of this part is Proposition B.1, stated below, and the remainder of the section will consist in proving it.

**Proposition B.1.** *for $\rho > 0$, $\varepsilon \geq 0$, $\sigma > 0$ and $\delta \in (0,1)$, assume that there is some $0 < \underline{\lambda} \leq \overline{\lambda} < +\infty$ such that, with probability at least $1 - \frac{\delta}{2}$,*

$$\forall f \in \mathcal{F}, \quad \widehat{\mathcal{R}}^{\varepsilon}_{\rho^2}(f) = \inf_{\underline{\lambda} \leq \lambda \leq \overline{\lambda}} \lambda \rho^2 + \mathbb{E}_{\xi \sim \widehat{P}_n}\left[\phi(f, \xi, \lambda, \varepsilon, \sigma)\right]. \tag{25}$$

*then, when $\rho^2 \geq \overline{\rho}_n^2(\delta, \underline{\lambda}, \overline{\lambda}, \varepsilon, \sigma)$, with probability $1 - \delta$,*

$$\forall f \in \mathcal{F}, \quad \widehat{\mathcal{R}}^{\varepsilon}_{\rho^2}(f) \geq \mathcal{R}^{\varepsilon}_{\rho^2 - \overline{\rho}_n^2(\delta, \underline{\lambda}, \overline{\lambda}, \varepsilon, \sigma)}(f).$$

The proof of this result mainly consists in verifying that under our standing assumptions, we can apply the concentration result presented in Lemma G.2 in order to concentrate $\widehat{\mathcal{R}}^{\varepsilon}_{\rho^2}(f)$ towards $\mathcal{R}^{\varepsilon}_{\rho^2}(f)$ through their dual formulations.

We begin by showing that the dual generator *divided by $\lambda$* is Lipchitz continuous in $f$ and in $\lambda^{-1}$ (for convenience, we use the notation $\mu = \lambda^{-1}$).

**Lemma B.2.** *Fix some $\overline{\lambda} \geq \underline{\lambda} > 0$. For any $\xi \in \Xi$, $\varepsilon \geq 0$ and $\sigma > 0$ we have that*

(a) *for any $\lambda \in [\underline{\lambda}, \overline{\lambda}]$, $f \mapsto \lambda^{-1}\phi(f, \xi, \lambda, \varepsilon, \sigma)$ is $\underline{\lambda}^{-1}$-Lipschitz continuous w.r.t. the norm $\|\cdot\|_\infty$;*

(b) *for any $f \in \mathcal{F}$, $\mu \mapsto \mu\phi(f, \xi, \mu^{-1}, \varepsilon, \sigma)$ is $\max\left(\widetilde{F}(\underline{\lambda}), a(\underline{\lambda}, \overline{\lambda}, \varepsilon, \sigma)\right)$-Lipschitz continuous on $\left[\overline{\lambda}^{-1}, \underline{\lambda}^{-1}\right]$.*

*Proof.* Item (a). When $\varepsilon = 0$, $f \mapsto \phi(f, \xi, \lambda, \varepsilon, \sigma)$ is a supremum of 1-Lipschitz functions and is thus 1-Lipschitz. For $\varepsilon > 0$, take $f, g \in \mathcal{F}$ and, for $t \in [0,1]$, define $f_t = f + t(g - f)$. Differentiating $t \mapsto \phi(f_t, \xi, \lambda, \varepsilon, \sigma)$ yields

$$\left|\frac{\mathrm{d}}{\mathrm{d}t}\phi(f_t, \xi, \lambda, \varepsilon, \sigma)\right| = \left|\frac{\mathbb{E}_{\zeta \sim \pi_\sigma(\cdot|\xi)}\left[(g(\zeta) - f(\zeta))e^{\frac{f_t(\zeta) - \lambda\|\xi - \zeta\|^2/2}{\varepsilon}}\right]}{\mathbb{E}_{\zeta' \sim \pi_\sigma(\cdot|\xi)}\left[e^{\frac{f_t(\zeta) - \lambda\|\xi - \zeta'\|^2/2}{\varepsilon}}\right]}\right| \leq \|g - f\|_\infty,$$

which gives that $f \mapsto \phi(f, \xi, \lambda, \varepsilon, \sigma)$ is 1-Lipschitz continuous w.r.t. the norm $\|\cdot\|_\infty$.

Since this bound is uniform in $\lambda$, we immediately get that $f \mapsto \lambda^{-1}\phi(f, \xi, \lambda, \varepsilon, \sigma)$ is $\underline{\lambda}^{-1}$-Lipschitz continuous for all $\lambda \geq \underline{\lambda}$.

Item (b). Fix $f \in \mathcal{F}$, $\xi \in \Xi$, $\varepsilon \geq 0$, $\sigma > 0$ and define $g(\lambda) \coloneqq \lambda \mapsto \lambda^{-1}\phi(f, \xi, \lambda, \varepsilon, \sigma)$.

Let us first begin with the case $\varepsilon = 0$. Take $\lambda, \lambda' \in [\underline{\lambda}, \overline{\lambda}]$. Without loss of generality, we can suppose that $g(\lambda) \geq g(\lambda')$. Since $f$ is continuous and $\Xi$ is a compact set, choose $\zeta \in \arg\max_{\zeta \in \Xi}\left\{f(\zeta) - \frac{\lambda}{2}\|\xi - \zeta\|^2\right\}$. Then, the claim comes from the fact that

$$0 \leq g(\lambda) - g(\lambda') \leq \lambda^{-1}f(\zeta) - \frac{1}{2}\|\xi - \zeta\|^2 - \left(\lambda'^{-1}f(\zeta) - \frac{1}{2}\|\xi - \zeta\|^2\right) \leq |\lambda^{-1} - \lambda'^{-1}|\widetilde{F}(\underline{\lambda}),$$

where we use that since $f$ is non-negative by assumption, $\left|\widetilde{F}(\lambda)\right| = \widetilde{F}(\lambda) \leq \widetilde{F}(\underline{\lambda})$.

Let us now turn to the case where $\varepsilon > 0$, for which $g$ is differentiable on $[\underline{\lambda}, \overline{\lambda}]$ with derivative

$$g'(\lambda) = -\frac{1}{\lambda^2}\phi(f, \xi, \lambda, \varepsilon, \sigma) - \frac{1}{\lambda}\frac{\mathbb{E}_{\zeta \sim \pi_\sigma(\cdot|\xi)}\left[\frac{1}{2}\|\xi - \zeta'\|^2 e^{\frac{f(\zeta) - \lambda\|\xi - \zeta\|^2/2}{\varepsilon}}\right]}{\mathbb{E}_{\zeta' \sim \pi_\sigma(\cdot|\xi)}\left[e^{\frac{f(\zeta') - \lambda\|\xi - \zeta'\|^2/2}{\varepsilon}}\right]}.$$

Since the claimed result is the Lipchitz continuity of $h : \mu \mapsto g(\mu^{-1})$, it suffices to bound its derivative, i.e., to bound $-\lambda^{-2}g'(\lambda)$ for all $\overline{\lambda} \geq \lambda \geq \underline{\lambda}$. On the one hand, thanks to Lemma G.7, it is bounded above as

$$-\frac{1}{\lambda^2}g'(\lambda) \leq \frac{\mathbb{E}_{\zeta \sim \pi_\sigma(\cdot|\xi)}\left[(f(\zeta) - \frac{\lambda}{2}\|\xi - \zeta\|^2)e^{\frac{f(\zeta) - \lambda\|\xi - \zeta\|^2/2}{\varepsilon}}\right]}{\mathbb{E}_{\zeta' \sim \pi_\sigma(\cdot|\xi)}\left[e^{\frac{f(\zeta') - \lambda\|\xi - \zeta'\|^2/2}{\varepsilon}}\right]} + \lambda\frac{\mathbb{E}_{\zeta \sim \pi_\sigma(\cdot|\xi)}\left[\frac{1}{2}\|\xi - \zeta\|^2 e^{\frac{f(\zeta) - \lambda\|\xi - \zeta\|^2/2}{\varepsilon}}\right]}{\mathbb{E}_{\zeta' \sim \pi_\sigma(\cdot|\xi)}\left[e^{\frac{f(\zeta') - \lambda\|\xi - \zeta'\|^2/2}{\varepsilon}}\right]}$$

$$\leq \phi(f, \xi, \lambda, 0) \leq \widetilde{F}(\lambda) \leq \widetilde{F}(\underline{\lambda}).$$

On the other hand, invoking Corollary A.5 also yields that

$$-\frac{1}{\lambda^2}g'(\lambda) \geq \varepsilon \log\left(\mathbb{E}_{\zeta \sim \pi_\sigma(\cdot|\xi)} e^{\frac{f(\zeta)-\lambda\|\xi-\zeta\|^2/2}{\varepsilon}}\right) \geq -a(\underline{\lambda}, \overline{\lambda}, \varepsilon, \sigma),$$

which concludes the proof. $\qquad\square$

We can now apply standard concentration for bounded Lipschitz quantities to bound the difference between the expectation of the dual generator over the empirical distribution $\widehat{P}_n$ and true one P.

**Lemma B.3.** *For $\rho > 0$, $\varepsilon \geq 0$, $\sigma > 0$, $\delta \in (0,1)$ and some $0 < \underline{\lambda} \leq \overline{\lambda} < +\infty$, we have with probability at least $1 - \frac{\delta}{2}$ that*

$$\sup_{(f,\lambda)\in\mathcal{F}\times[\underline{\lambda},\overline{\lambda}]} \left\{ \frac{\mathbb{E}_{\xi\sim P}[\phi(f,\xi,\lambda,\varepsilon,\sigma)] - \mathbb{E}_{\xi\sim\widehat{P}_n}[\phi(f,\xi,\lambda,\varepsilon,\sigma)]}{\lambda} \right\} \leq \bar{\rho}_n^2(\delta, \underline{\lambda}, \overline{\lambda}, \varepsilon, \sigma).$$

*Proof.* Our objective is to bound the quantity

$$\sup_{(f,\lambda)\in\mathcal{F}\times[\underline{\lambda},\overline{\lambda}]} \left\{ \frac{\mathbb{E}_{\xi\sim P}[\phi(f,\xi,\lambda,\varepsilon,\sigma)] - \mathbb{E}_{\xi\sim\widehat{P}_n}\left[\phi(f,\widehat{P}_n,\lambda,\varepsilon,\sigma)\right]}{\lambda} \right\}$$

$$= \sup_{(f,\mu)\in\mathcal{F}\times\left[\overline{\lambda}^{-1},\underline{\lambda}^{-1}\right]} \left\{ \mathbb{E}_{\xi\sim P}\left[\mu\,\phi(f,\xi,\mu^{-1},\varepsilon,\sigma)\right] - \mathbb{E}_{\xi\sim\widehat{P}_n}\left[\mu\,\phi(f,\xi,\mu^{-1},\varepsilon,\sigma)\right] \right\}$$

$$= \sup_{(f,\mu)\in\mathcal{X}} \left\{ \mathbb{E}_{\xi\sim P}[X((f,\mu),\xi)] - \mathbb{E}_{\xi\sim\widehat{P}_n}[X((f,\mu),\xi)] \right\},$$

where we used again the notation $\mu = \lambda^{-1}$ and defined

$$\mathcal{X} := \mathcal{F} \times \left[\overline{\lambda}^{-1}, \underline{\lambda}^{-1}\right] \quad \text{and} \quad X((f,\mu),\xi) := \mu\,\phi(f,\xi,\mu^{-1},\varepsilon,\sigma).$$

Let us endow $\mathcal{X}$ with the distance,

$$\mathrm{dist}((f,\mu),(f',\mu')) := \underline{\lambda}^{-1}\|f - f'\|_\infty + \max\left(\widetilde{F}(\underline{\lambda}), a(\underline{\lambda}, \overline{\lambda}, \varepsilon, \sigma)\right)|\mu - \mu'|.$$

We now wish to apply Lemma G.2 and check its three requirements:

1. For any $(f,\mu) \in \mathcal{F} \times \left[\overline{\lambda}^{-1}, \underline{\lambda}^{-1}\right]$, $X((f,\mu),\cdot)$ is measurable since the functions of $\mathcal{F}$ are continuous and thus *a fortiori* measurable;

2. By Lemma B.2, for any $\varepsilon \geq 0$ and any $\xi \in \mathrm{supp}\,P$, $X(\cdot,\xi)$ is 1-Lipschitz w.r.t. $\mathrm{dist}$;

3. Thanks to Corollary A.5, for any $(f,\mu) \in \mathcal{X}$, $\xi \in \mathrm{supp}\,P$, $\varepsilon \geq 0$ and $\sigma > 0$, we have

$$-\frac{a(\underline{\lambda}, \overline{\lambda}, \varepsilon, \sigma)}{\underline{\lambda}} \leq X((f,\mu),\xi) \leq \frac{\widetilde{F}(\underline{\lambda})}{\underline{\lambda}}.$$

As a consequence, applying statement $(b)$ of Lemma G.2 yields that, with probability at least $1 - \frac{\delta}{2}$,

$$\sup_{(f,\lambda)\in\mathcal{F}\times[\underline{\lambda},\overline{\lambda}]} \left\{ \frac{\mathbb{E}_{\xi\sim P}[\phi(f,\xi,\lambda,\varepsilon,\sigma)] - \mathbb{E}_{\xi\sim\widehat{P}_n}\left[\phi(f,\widehat{P}_n,\lambda,\varepsilon,\sigma)\right]}{\lambda} \right\}$$

$$\leq \frac{48\mathcal{I}(\mathcal{X},\mathrm{dist})}{\sqrt{n}} + \frac{2}{\underline{\lambda}}\left(\widetilde{F}(\underline{\lambda}) + a(\underline{\lambda}, \overline{\lambda}, \varepsilon, \sigma)\right)\sqrt{\frac{\log\frac{2}{\delta}}{2n}}.$$

We now proceed to bound $\mathcal{I}(\mathcal{X}, \mathrm{dist})$. Exploiting the product space structure of $\mathcal{X}$ and $\mathrm{dist}$ with Lemma G.3, one has that,

$$\mathcal{I}(\mathcal{X},\mathrm{dist}) \leq \underline{\lambda}^{-1}\mathcal{I}(\mathcal{F}, \|\cdot\|_\infty) + \max\left(\widetilde{F}(\underline{\lambda}), a(\underline{\lambda}, \overline{\lambda}, \varepsilon, \sigma)\right)\mathcal{I}([0, \underline{\lambda}^{-1}], |\cdot|)$$

$$\leq \underline{\lambda}^{-1}\left(\mathcal{I}(\mathcal{F}, \|\cdot\|_\infty) + \max\left(\widetilde{F}(\underline{\lambda}), a(\underline{\lambda}, \overline{\lambda}, \varepsilon, \sigma)\right)\frac{1 + 2\log 2}{2}\right),$$

where we used Lemma G.4. Hence, we have shown that with probability at least $1 - \frac{\delta}{2}$,

$$\sup_{(f,\lambda)\in\mathcal{F}\times[\underline{\lambda},\overline{\lambda}]}\left\{\frac{\mathbb{E}_{\xi\sim\mathrm{P}}[\phi(f,\xi,\lambda,\varepsilon,\sigma)]-\mathbb{E}_{\xi\sim\widehat{\mathrm{P}}_n}[\phi(f,\xi,\lambda,\varepsilon,\sigma)]}{\lambda}\right\}\leq\overline{\rho}_n^2(\delta,\underline{\lambda},\overline{\lambda},\varepsilon,\sigma)$$

where some numerical constants have been simplified. $\qquad\square$

*Proof of Proposition B.1.* Building on Lemma B.3, we can now conclude the main result of this section. Using our boundedness assumption on $\lambda$, we have that, with probability $1 - \delta$, the two following statements hold simultaneously

- $\forall f \in \mathcal{F}, \quad \widehat{\mathcal{R}}_{\rho^2}^{\varepsilon}(f) = \inf_{\underline{\lambda}\leq\lambda\leq\overline{\lambda}} \lambda\rho^2 + \mathbb{E}_{\xi\sim\widehat{\mathrm{P}}_n}[\phi(f,\xi,\lambda,\varepsilon,\sigma)]$ ;

- $\displaystyle\sup_{(f,\lambda)\in\mathcal{F}\times[\underline{\lambda},\overline{\lambda}]}\left\{\frac{\mathbb{E}_{\xi\sim\mathrm{P}}[\phi(f,\xi,\lambda,\varepsilon,\sigma)]-\mathbb{E}_{\xi\sim\widehat{\mathrm{P}}_n}[\phi(f,\xi,\lambda,\varepsilon,\sigma)]}{\lambda}\right\}\leq\overline{\rho}_n^2(\delta,\underline{\lambda},\overline{\lambda},\varepsilon,\sigma)\,.$

As a consequence, on this event, for any $f \in \mathcal{F}$,

$$\widehat{\mathcal{R}}_{\rho^2}^{\varepsilon}(f) = \inf_{\underline{\lambda}\leq\lambda\leq\overline{\lambda}}\left\{\lambda\rho^2 + \mathbb{E}_{\xi\sim\widehat{\mathrm{P}}_n}[\phi(f,\xi,\lambda,\varepsilon,\sigma)]\right\}$$

$$= \inf_{\underline{\lambda}\leq\lambda\leq\overline{\lambda}}\left\{\lambda\rho^2 + \mathbb{E}_{\xi\sim\mathrm{P}}[\phi(f,\xi,\lambda,\varepsilon,\sigma)] - \lambda\frac{\mathbb{E}_{\xi\sim\mathrm{P}}[\phi(f,\xi,\lambda,\varepsilon,\sigma)]-\mathbb{E}_{\xi\sim\widehat{\mathrm{P}}_n}[\phi(f,\xi,\lambda,\varepsilon,\sigma)]}{\lambda}\right\}$$

$$\geq \inf_{\underline{\lambda}\leq\lambda\leq\overline{\lambda}}\left\{\lambda\rho^2 + \mathbb{E}_{\xi\sim\mathrm{P}}[\phi(f,\xi,\lambda,\varepsilon,\sigma)] - \lambda\sup_{\underline{\lambda}\leq\lambda'\leq\overline{\lambda}}\frac{\mathbb{E}_{\xi\sim\mathrm{P}}[\phi(f,\xi,\lambda',\varepsilon,\sigma)]-\mathbb{E}_{\xi\sim\widehat{\mathrm{P}}_n}[\phi(f,\xi,\lambda',\varepsilon,\sigma)]}{\lambda'}\right\}$$

$$\geq \inf_{\underline{\lambda}\leq\lambda\leq\overline{\lambda}}\left\{\lambda\rho^2 + \mathbb{E}_{\xi\sim\mathrm{P}}[\phi(f,\xi,\lambda,\varepsilon,\sigma)] - \lambda\overline{\rho}_n^2(\delta,\underline{\lambda},\overline{\lambda},\varepsilon,\sigma)\right\}$$

$$\geq \mathcal{R}_{\rho^2-\overline{\rho}_n^2(\delta,\underline{\lambda},\overline{\lambda},\varepsilon,\sigma)}^{\varepsilon}(f)$$

where $\rho^2 - \overline{\rho}_n^2(\delta,\underline{\lambda},\overline{\lambda},\varepsilon,\sigma) \geq 0$ by assumption. $\qquad\square$

**Remark B.4.** *Note that the proof of Proposition B.1 actually gives us the slightly stronger result at the penultimate equation: with probability at least $1 - \delta$, for any $f \in \mathcal{F}$,*

$$\widehat{\mathcal{R}}_{\rho^2}^{\varepsilon}(f) \geq \inf_{\underline{\lambda}\leq\lambda\leq\overline{\lambda}}\left\{\lambda(\rho^2 - \overline{\rho}_n^2(\delta,\underline{\lambda},\overline{\lambda},\varepsilon,\sigma)) + \mathbb{E}_{\xi\sim\mathrm{P}}[\phi(f,\xi,\lambda,\varepsilon,\sigma)]\right\}\,,$$

*that we will require later.*

## C Dual bound when $\rho$ is small

In this section, we show how the condition (25) of Proposition B.1 can be obtained when the robustness radius $\rho$ is small enough. The results of this section cover both the standard WDRO setting of Theorems 3.1 and 3.3 and the regularized case of Theorem 3.4.

In the following Assumption 8, we precise how small $\rho$ has to be; we also take $\varepsilon$ and $\sigma$ proportional to $\rho$ in order to get close to the true risk with $\rho$, $\varepsilon$ and $\sigma$ "small" at the same time. The main result of this section is Proposition C.1, whose proof relies on Lemma C.2.

**Assumption 8** ($\rho$ is small)**.** Take $\varepsilon = \varepsilon_0\rho$, $\sigma = \sigma_0\rho$ with $\varepsilon_0 \geq 0$, $\sigma_0 > 0$ and define

$$\lambda_0^* := \varepsilon_0 d + \sqrt{(\varepsilon_0 d)^2 + 8\inf_{f\in\mathcal{F}}\mathbb{E}_{\mathrm{P}}\|\nabla f\|_2^2} \qquad \mu^* := \frac{8\inf_{f\in\mathcal{F}}\mathbb{E}_{\mathrm{P}}\|\nabla f\|_2^2}{(\lambda_0^*)^3} + \frac{2\varepsilon_0 d}{(\lambda_0^{*2})}\,.$$

Moreover, assume that $\varepsilon_0$ and $\sigma_0$ satisfy

$$\frac{\varepsilon_0}{\sigma_0^2} \leq \frac{\lambda_0^*}{8}\,.$$

Assume that $\rho > 0$ is small enough so that,

$$\rho \le \min\left( \frac{\varepsilon_1}{\varepsilon_0}, \frac{\lambda_0^*}{32(\lambda_1 + L_2)}, \frac{\mu^*(\lambda_0^*)^2}{4096 L_2}, \sqrt{\frac{c_2^3 \lambda_0^*}{8\varepsilon_0}} \left( \log\left( \frac{4096\varepsilon_0 c_1}{\mu^*(\lambda_0^*)^2} \right) \right)_+^{-\frac{3}{2}} \right)$$

where $\lambda_1, \varepsilon_1, c_1, c_2$ are positive constant given by Lemma A.3 and $\sigma$ comes from (4).

Note that $\lambda_0^*$ and $\mu^*$ are both always positive, be it thanks to Assumption 5 or the regularization with $\varepsilon_0 > 0$. For such values of $\rho$, the main result of this section Proposition C.1 shows that the dual variable of (12) can be bounded with high probability.

**Proposition C.1.** *Let Assumption 8 hold and fix a threshold $\delta \in (0, 1)$. Assume in addition that*

$$\rho \ge \frac{8192}{\sqrt{n}\,\mu^*(\lambda_0^*)^2} \left( 12\mathcal{I}(\mathcal{F}, \|\cdot\|_\infty) + (\widetilde{F}(0) + \overline{M}(\rho))\sqrt{1 + \log\frac{1}{\delta}} \right)$$

*where $\mathcal{I}(\mathcal{F}, \|\cdot\|_\infty), \widetilde{F}$ are defined in Section A.1 and $M(\underline{\lambda}, \overline{\lambda}, \varepsilon, \sigma)$, the bounding term appearing in Proposition A.2, is used to define*

$$\overline{M}(\rho) := \sup_{\rho' \in (0,\rho]} M\left( \max\left( \frac{\lambda_0^*}{32\rho'}, \lambda_1 + L_2 \right), \frac{\lambda_0^*}{2\rho'}, \varepsilon_0\rho', \sigma_0\rho' \right).$$

*Then, with probability at least $1 - \delta$, we have*

$$\forall f \in \mathcal{F}, \quad \widehat{\mathcal{R}}_{\rho^2}^\varepsilon(f) = \inf_{\frac{\lambda_0^*}{32\rho} \le \lambda \le \frac{\lambda_0^*}{2\rho}} \lambda\rho^2 + \mathbb{E}_{\xi \sim \widehat{P}_n}\left[ \phi(f, \xi, \lambda, \varepsilon, \sigma) \right].$$

To show Proposition C.1, we need the following helper lemma.

**Lemma C.2.** *Let Assumption 8 hold. Then,*

$$\left( \frac{\lambda_0^*}{4\rho} - \frac{\varepsilon_0\rho}{\sigma^2} + L_2 \right)\rho^2 + \mathbb{E}_{\xi \sim P}\left[ \phi\left( f, \xi, \frac{\lambda_0^*}{4\rho} - \frac{\varepsilon_0\rho}{\sigma^2} + L_2, \varepsilon \right) \right] + \frac{\rho\mu^*}{1024}(\lambda_0^*)^2$$

$$\le \min\left( \left( \frac{\lambda_0^*}{8\rho} - \frac{\varepsilon_0\rho}{2\sigma^2} - L_2 \right)\rho^2 + \mathbb{E}_{\xi \sim P}\left[ \phi\left( f, \xi, \frac{\lambda_0^*}{8\rho} - \frac{\varepsilon_0\rho}{2\sigma^2} - L_2, \varepsilon \right) \right], \right.$$

$$\left. \left( \frac{\lambda_0^*}{2\rho} - \frac{2\varepsilon_0\rho}{\sigma^2} + L_2 \right)\rho^2 + \mathbb{E}_{\xi \sim P}\left[ \phi\left( f, \xi, \frac{\lambda_0^*}{2\rho} - \frac{2\varepsilon_0\rho}{\sigma^2} + L_2, \varepsilon \right) \right] \right)$$

*and* $\quad \max\left( \frac{\lambda_0^*}{32\rho}, \lambda_1 + L_2 \right) \le \frac{\lambda_0^*}{8\rho} - \frac{\varepsilon_0\rho}{2\sigma^2} - L_2 \le \frac{\lambda_0^*}{4\rho} - \frac{\varepsilon_0\rho}{\sigma^2} + L_2 \le \frac{\lambda_0^*}{2\rho}$

*Proof.* Fix $f \in \mathcal{F}$. Consider the function $\overline{\psi_\rho} : \lambda \mapsto \lambda\rho^2 + \mathbb{E}_{\xi \sim P}\left[ \overline{\phi}(f, \xi, \lambda, \varepsilon, \sigma) \right]$ where $\overline{\phi}$ is defined in (19). By Lemma G.8 invoked with $a \leftarrow \rho^2$, $b \leftarrow \frac{1}{2}\mathbb{E}_P\left[ \|\nabla f\|_2^2 \right]$, $c \leftarrow \frac{\varepsilon d}{2}$ and $r \leftarrow \frac{\varepsilon}{\sigma^2}$, its unique minimizer is

$$\lambda^\star := \left[ \frac{\varepsilon_0 d + \sqrt{(\varepsilon_0 d)^2 + 8\mathbb{E}_P\|\nabla f\|_2^2}}{4\rho} - \frac{\varepsilon_0\rho}{\sigma^2} \right]_+ = \left[ \frac{\lambda_0^*}{4\rho} - \frac{\varepsilon_0\rho}{\sigma^2} \right]_+.$$

where we used that $\varepsilon = \varepsilon_0\rho$. And, since $\frac{\varepsilon_0\rho^2}{\sigma^2} = \frac{\varepsilon_0}{\sigma_0^2} \le \frac{\lambda_0^*}{8}$ by Assumption 8, $\lambda^\star$ actually satisfies

$$\frac{\lambda_0^*}{8\rho} \le \lambda^\star \le \frac{\lambda_0^*}{4\rho}. \tag{26}$$

Moreover, Lemma G.8 also shows that, on $[0, 2\lambda^\star]$, $\overline{\psi_\rho}$ is strongly convex with modulus

$$\frac{\mathbb{E}_P\left[ \|\nabla f\|_2^2 \right]}{(2\lambda^\star + \frac{\varepsilon_0\rho}{\sigma^2})^3} + \frac{\varepsilon_0 d\rho}{2(2\lambda^\star + \frac{\varepsilon_0\rho}{\sigma^2})^2} = \frac{\mathbb{E}_P\left[ \|\nabla f\|_2^2 \right]}{(\frac{\lambda_0^*}{2\rho} - \frac{\varepsilon_0\rho}{\sigma^2})^3} + \frac{\varepsilon_0 d\rho}{2(\frac{\lambda_0^*}{2\rho} - \frac{\varepsilon_0\rho}{\sigma^2})^2} \ge \rho^3\mu^*.$$

Now, we notice that $\varepsilon = \varepsilon_0 \rho \leq \varepsilon_1$ by Assumption 8. Then, if $\lambda \in [\lambda_1 + L_2, 2\lambda^\star - L_2]$, then Lemma A.3 (applied twice) and the strong convexity of $\overline{\psi}_\rho$ yield

$$\lambda \rho^2 + \mathbb{E}_{\xi \sim P}[\phi(f, \xi, \lambda, \varepsilon, \sigma)]$$

$$\geq \lambda \rho^2 + \mathbb{E}_{\xi \sim P}\left[\overline{\phi}(f, \xi, \lambda + L_2, \varepsilon, \sigma)\right] - \varepsilon c_1 e^{-c_2\left(\frac{\lambda + L_2}{\varepsilon}\right)^{\frac{1}{3}}}$$

$$\geq \lambda^\star \rho^2 + \mathbb{E}_{\xi \sim P}\left[\overline{\phi}(f, \xi, \lambda^\star, \varepsilon, \sigma)\right] - \rho^2 L_2 + \frac{\rho^3 \mu^*}{2}(\lambda^\star - (\lambda + L_2))^2 - \varepsilon c_1 e^{-c_2\left(\frac{\lambda + L_2}{\varepsilon}\right)^{\frac{1}{3}}}$$

$$\geq \lambda^\star \rho^2 + \mathbb{E}_{\xi \sim P}[\phi(f, \xi, \lambda^\star + L_2, \varepsilon, \sigma)] - \rho^2 L_2 + \frac{\rho^3 \mu^*}{2}(\lambda^\star - (\lambda + L_2))^2 - \varepsilon c_1 e^{-c_2\left(\frac{\lambda + L_2}{\varepsilon}\right)^{\frac{1}{3}}} - \varepsilon c_1 e^{-c_2\left(\frac{\lambda^\star}{\varepsilon}\right)^{\frac{1}{3}}}.$$
(27)

We first wish to choose $\lambda = \frac{\lambda^\star}{2} - L_2$. By (26), since $\rho \leq \frac{\lambda_0^*}{32(\lambda_1 + L_2)}$ by Assumption 8, this choice of $\lambda$ is indeed greater than or equal to $\lambda_1 + L_2$ and (27) leads to

$$\left(\frac{\lambda^\star}{2} - L_2\right) \rho^2 + \mathbb{E}_{\xi \sim P}\left[\phi(f, \xi, \frac{\lambda^\star}{2} - L_2, \varepsilon, \sigma)\right]$$

$$\geq \lambda^\star \rho^2 + \mathbb{E}_{\xi \sim P}[\phi(f, \xi, \lambda^\star + L_2, \varepsilon, \sigma)] - \rho^2 L_2 + \frac{\rho^3 \mu^*}{8}(\lambda^\star)^2 - \varepsilon c_1 e^{-c_2\left(\frac{\lambda^\star}{2\varepsilon}\right)^{\frac{1}{3}}} - \varepsilon c_1 e^{-c_2\left(\frac{\lambda^\star}{\varepsilon}\right)^{\frac{1}{3}}}$$

$$\geq (\lambda^\star + L_2)\rho^2 + \mathbb{E}_{\xi \sim P}[\phi(f, \xi, \lambda^\star + L_2, \varepsilon, \sigma)] - 2\rho^2 L_2 + \frac{\rho \mu^*}{512}(\lambda_0^*)^2 - 2\varepsilon c_1 e^{-c_2\left(\frac{\lambda_0^*}{8\varepsilon \rho}\right)^{\frac{1}{3}}}.$$
(28)

where we used (26) again for the last inequality.

To obtain the other inequality we pick $\lambda = 2\lambda^\star - L_2$, which is greater or equal to $\lambda_1 + L_2$ by Assumption 8 and (26) as above. Then, (27) yields

$$(2\lambda^\star - L_2)\rho^2 + \mathbb{E}_{\xi \sim P}[\phi(f, \xi, 2\lambda^\star - L_2, \varepsilon, \sigma)]$$

$$\geq \lambda^\star \rho^2 + \mathbb{E}_{\xi \sim P}[\phi(f, \xi, \lambda^\star + L_2, \varepsilon, \sigma)] - \rho^2 L_2 + \frac{\rho^3 \mu^*}{2}(\lambda^\star)^2 - \varepsilon c_1 e^{-c_2\left(\frac{2\lambda^\star}{\varepsilon}\right)^{\frac{1}{3}}} - \varepsilon c_1 e^{-c_2\left(\frac{\lambda^\star}{\varepsilon}\right)^{\frac{1}{3}}}$$

$$\geq (\lambda^\star + L_2)\rho^2 + \mathbb{E}_{\xi \sim P}[\phi(f, \xi, \lambda^\star + L_2, \varepsilon, \sigma)] - 2\rho^2 L_2 + \frac{\rho \mu^*}{512}(\lambda_0^*)^2 - 2\varepsilon c_1 e^{-c_2\left(\frac{\lambda_0^*}{8\varepsilon \rho}\right)^{\frac{1}{3}}}$$

where we used (26) again, and degraded the constants to match those of (28).

Thus, we have that $\lambda^\star = \frac{\lambda_0^*}{4\rho} - \frac{\varepsilon_0 \rho}{\sigma^2}$ and

$$(\lambda^\star + L_2)\rho^2 + \mathbb{E}_{\xi \sim P}[\phi(f, \xi, \lambda^\star + L_2, \varepsilon, \sigma)] - 2\rho^2 L_2 + \frac{\rho \mu^*}{512}(\lambda_0^*)^2 - 2\varepsilon c_1 e^{-c_2\left(\frac{\lambda_0^*}{8\varepsilon \rho}\right)^{\frac{1}{3}}}$$

$$\leq \min\left(\left(\frac{\lambda^\star}{2} - L_2\right)\rho^2 + \mathbb{E}_{\xi \sim P}\left[\phi(f, \xi, \frac{\lambda^\star}{2} - L_2, \varepsilon, \sigma)\right], (2\lambda^\star - L_2)\rho^2 + \mathbb{E}_{\xi \sim P}[\phi(f, \xi, 2\lambda^\star - L_2, \varepsilon, \sigma)]\right)$$

All that is left to show for the main result of the lemma is that

$$- 2\rho^2 L_2 + \frac{\rho \mu^*}{512}(\lambda_0^*)^2 - 2\varepsilon c_1 e^{-c_2\left(\frac{\lambda_0^*}{8\varepsilon \rho}\right)^{\frac{1}{3}}} \geq \frac{\rho \mu^*}{1024}(\lambda_0^*)^2$$

$$\Leftrightarrow \quad 2\rho L_2 + 2\varepsilon_0 c_1 e^{-c_2\left(\frac{\lambda_0^*}{8\varepsilon_0 \rho^2}\right)^{\frac{1}{3}}} \leq \frac{\mu^*}{1024}(\lambda_0^*)^2.$$
(29)

This is a consequence of Assumption 8 which states that

$$\rho \leq \frac{\mu^*}{4096 L_2}(\lambda_0^*)^2 \quad \text{and} \quad \rho \leq \sqrt{\frac{c_2^3 \lambda_0^*}{8\varepsilon_0}}\left(\log\left(\frac{4096\varepsilon_0 c_1}{\mu^*(\lambda_0^*)^2}\right)\right)_+^{-\frac{3}{2}},$$

which imply that $2\rho L_2 \leq \frac{\mu^*}{2048}(\lambda_0^*)^2$, and $2\varepsilon_0 c_1 e^{-c_2\left(\frac{\lambda_0^*}{8\varepsilon_0 \rho^2}\right)^{\frac{1}{3}}} \leq \frac{\mu^*}{2048}(\lambda_0^*)^2$,

so that (29) indeed holds, concluding the proof of the first part of the result.

The supplementary bounds follow directly from (26) and our assumptions on $\rho$. $\qquad \square$

We are now in a position to show our main result when $\rho$ is small, namely Proposition C.1.

*Proof of Proposition C.1.* Let us first take any $\lambda \in [\max\left(\frac{\lambda_0^*}{32\rho}, \lambda_1 + L_2\right), \frac{\lambda_0^*}{2\rho}]$. We want to instante Lemma G.2 with $X(f,\xi) \leftarrow \phi(f,\xi,\lambda,\varepsilon,\sigma)$, $(\mathcal{X}, \mathrm{dist}) \leftarrow (\mathcal{F}, \|\cdot\|_\infty)$, whose requirements are checked since:

1. For any $f \in \mathcal{F}$, $\phi(f,\xi,\lambda,\varepsilon,\sigma)$ is measurable since the functions of $\mathcal{F}$ are continuous and thus *a fortiori* measurable;

2. By the proof of Lemma B.2(a), we have that for any $\varepsilon \geq 0$, $\sigma > 0$ and any $\xi \in \mathrm{supp}\, \mathrm{P}$, $f \mapsto \phi(f,\xi,\lambda,\varepsilon,\sigma)$ is 1-Lipschitz continuous w.r.t. the norm $\|\cdot\|_\infty$;

3. With Proposition A.2 with $\underline{\lambda} \leftarrow \max\left(\frac{\lambda_0^*}{32\rho}, \lambda_1 + L_2\right), \overline{\lambda} \leftarrow \frac{\lambda_0^*}{2\rho}$, for any $f \in \mathcal{F}, \xi \in \mathrm{supp}\, \mathrm{P}$ and $\varepsilon \in [0, \varepsilon_1]$ (by Assumption 8), we have
$$-\overline{M}(\rho) \leq f(\xi) - M(\underline{\lambda}, \overline{\lambda}, \varepsilon, \sigma) \leq \phi(f,\xi,\lambda,\varepsilon,\sigma) \leq f(\xi) + M(\underline{\lambda}, \overline{\lambda}, \varepsilon, \sigma) \leq \widetilde{F}(0) + \overline{M}(\rho)$$
where $\overline{M}(\rho)$ is defined in Proposition C.1.

Since $\frac{\lambda_0^*}{8\rho} - \frac{\varepsilon_0\rho}{2\sigma^2} - L_2 \geq \lambda_1 + L_2$ by Lemma C.2, we can apply statement $(b)$ of Lemma G.2 with $\lambda \leftarrow \frac{\lambda_0^*}{8\rho} - \frac{\varepsilon_0\rho}{2\sigma^2} - L_2$ and $\delta \leftarrow \frac{\delta}{4}$ to have that, with probability at least $1 - \frac{\delta}{4}$, for all $f \in \mathcal{F}$
$$\mathbb{E}_{\xi\sim\mathrm{P}}\left[\phi\left(f,\xi,\frac{\lambda_0^*}{8\rho} - \frac{\varepsilon_0\rho}{2\sigma^2} - L_2,\varepsilon\right)\right] - \mathbb{E}_{\xi\sim\widehat{\mathrm{P}}_n}\left[\phi\left(f,\xi,\frac{\lambda_0^*}{8\rho} - \frac{\varepsilon_0\rho}{2\sigma^2} - L_2,\varepsilon\right)\right]$$
$$\leq \frac{48\mathcal{I}(\mathcal{F}, \|\cdot\|_\infty)}{\sqrt{n}} + 4(\widetilde{F}(0) + \overline{M}(\rho))\sqrt{\frac{\log\frac{4}{\delta}}{2n}}.$$

Similarly, we can apply statement $(a)$ of Lemma G.2 with $\lambda \leftarrow \frac{\lambda_0^*}{4\rho} - \frac{\varepsilon_0\rho}{\sigma^2} + L_2$ and $\delta \leftarrow \frac{\delta}{4}$ to get that, with probability at least $1 - \frac{\delta}{4}$, for all $f \in \mathcal{F}$,
$$\mathbb{E}_{\xi\sim\widehat{\mathrm{P}}_n}\left[\phi\left(f,\xi,\frac{\lambda_0^*}{4\rho} - \frac{\varepsilon_0\rho}{\sigma^2} + L_2,\varepsilon\right)\right] - \mathbb{E}_{\xi\sim\mathrm{P}}\left[\phi\left(f,\xi,\frac{\lambda_0^*}{4\rho} - \frac{\varepsilon_0\rho}{2\sigma^2} + L_2,\varepsilon\right)\right]$$
$$\leq \frac{48\mathcal{I}(\mathcal{F}, \|\cdot\|_\infty)}{\sqrt{n}} + 4(\widetilde{F}(0) + \overline{M}(\rho))\sqrt{\frac{\log\frac{4}{\delta}}{2n}}.$$

Combining the two statements above and using Lemma C.2, we get that, with probability at least $1 - \frac{\delta}{2}$, for any $f \in \mathcal{F}$,
$$\left(\frac{\lambda_0^*}{8\rho} - \frac{\varepsilon_0\rho}{2\sigma^2} - L_2\right)\rho^2 + \mathbb{E}_{\xi\sim\widehat{\mathrm{P}}_n}\left[\phi\left(f,\xi,\frac{\lambda_0^*}{8\rho} - \frac{\varepsilon_0\rho}{2\sigma^2} - L_2,\varepsilon\right)\right]$$
$$\geq \left(\frac{\lambda_0^*}{8\rho} - \frac{\varepsilon_0\rho}{2\sigma^2} - L_2\right)\rho^2 + \mathbb{E}_{\xi\sim\mathrm{P}}\left[\phi\left(f,\xi,\frac{\lambda_0^*}{8\rho} - \frac{\varepsilon_0\rho}{2\sigma^2} - L_2,\varepsilon\right)\right]$$
$$- \frac{48\mathcal{I}(\mathcal{F}, \|\cdot\|_\infty)}{\sqrt{n}} - 4(\widetilde{F}(0) + \overline{M}(\rho))\sqrt{\frac{\log\frac{4}{\delta}}{2n}}$$
$$\geq \left(\frac{\lambda_0^*}{4\rho} - \frac{\varepsilon_0\rho}{\sigma^2} + L_2\right)\rho^2 + \mathbb{E}_{\xi\sim\mathrm{P}}\left[\phi\left(f,\xi,\frac{\lambda_0^*}{4\rho} - \frac{\varepsilon_0\rho}{\sigma^2} + L_2,\varepsilon\right)\right]$$
$$+ \frac{\rho\mu^*}{1024}(\lambda_0^*)^2 - \frac{48\mathcal{I}(\mathcal{F}, \|\cdot\|_\infty)}{\sqrt{n}} - 4(\widetilde{F}(0) + \overline{M}(\rho))\sqrt{\frac{\log\frac{4}{\delta}}{2n}}$$
$$\geq \left(\frac{\lambda_0^*}{4\rho} - \frac{\varepsilon_0\rho}{\sigma^2} + L_2\right)\rho^2 + \mathbb{E}_{\xi\sim\widehat{\mathrm{P}}_n}\left[\phi\left(f,\xi,\frac{\lambda_0^*}{4\rho} - \frac{\varepsilon_0\rho}{\sigma^2} + L_2,\varepsilon\right)\right]$$
$$+ \frac{\rho\mu^*}{1024}(\lambda_0^*)^2 - \frac{96\mathcal{I}(\mathcal{F}, \|\cdot\|_\infty)}{\sqrt{n}} - 8(\widetilde{F}(0) + \overline{M}(\rho))\sqrt{\frac{\log\frac{4}{\delta}}{2n}}.$$

Noting that the assumption on $\rho$ in Proposition C.1 implies that

$$\frac{\rho\mu^*}{1024}(\lambda_0^*)^2 \geq \frac{96\mathcal{I}(\mathcal{F}, \|\cdot\|_\infty)}{\sqrt{n}} + 8\left(\widetilde{F}(0) + \overline{M}(\rho)\right)\sqrt{\frac{\log\frac{4}{\delta}}{2n}}$$

we have proven that, with probability at least $1 - \frac{\delta}{2}$, for any $f \in \mathcal{F}$,

$$\psi_\rho\left(\frac{\lambda_0^*}{8\rho} - \frac{\varepsilon_0\rho}{2\sigma^2} - L_2\right) \geq \psi_\rho\left(\frac{\lambda_0^*}{4\rho} - \frac{\varepsilon_0\rho}{\sigma^2} + L_2\right)$$

where $\psi_\rho : \lambda \mapsto \lambda\rho^2 + \mathbb{E}_{\xi\sim\widehat{P}_n}[\phi(f, \xi, \lambda, \varepsilon, \sigma)]$. Now, since $\psi_\rho$ is convex, this means that its minimizers on $\mathbb{R}_+$ are greater than

$$\frac{\lambda_0^*}{8\rho} - \frac{\varepsilon_0\rho}{2\sigma^2} - L_2 \geq \frac{\lambda_0^*}{32\rho}$$

where the inequality comes from Lemma C.2.

Using the same reasoning, one can get that with probability at least $1 - \frac{\delta}{2}$ the minimizers are no greater than

$$\frac{\lambda_0^*}{4\rho} - \frac{\varepsilon_0\rho}{\sigma^2} + L_2 \leq \frac{\lambda_0^*}{2\rho}.$$

Thus, we have shown that with probability at least $1 - \frac{\delta}{2}$, for any $f \in \mathcal{F}$,

$$\widehat{\mathcal{R}}_{\rho^2}^\varepsilon(f) = \inf_{\lambda\geq 0} \lambda\rho^2 + \mathbb{E}_{\xi\sim\widehat{P}_n}\left[\phi(f, \xi, \lambda, \varepsilon, \sigma)\right]$$

$$= \inf_{\frac{\lambda_0^*}{32\rho}\leq\lambda\leq\frac{\lambda_0^*}{2\rho}} \lambda\rho^2 + \mathbb{E}_{\xi\sim\widehat{P}_n}\left[\phi(f, \xi, \lambda, \varepsilon, \sigma)\right].$$

$\square$

# D Dual bound when $\rho$ is close to this maximal radius

Complementary to the previous section Section C, we consider the case where $\rho$ is close than the critical radius. Though the bounds of this section are much worse that the one of Section C when $\rho$ goes to zero, they hold for the whole ranges of $\rho$ considered in the theorems.

As mentioned in Remark 3.2, as $\rho$ grows, the Wasserstein ball constraint can stop being active, leading to a null dual variable. Thus, it is essential that $\rho$ be lower then the critical radius to stay in the distributionally robust regime and to avoid the worst-case regime. In that case, we are able to lower-bound the dual multiplier $\lambda$.

We defined the critical radius in standard WDRO case in Remark 3.2 and we extend it here to cover the regularized case:

$$\rho_c^2(\varepsilon, \sigma) := \begin{cases} \inf_{f\in\mathcal{F}} \mathbb{E}_{\xi\sim P}\left[\mathbb{E}_{\zeta\sim\pi_\sigma^{f/\varepsilon}(\cdot|\xi)}\left[\frac{1}{2}\|\xi - \zeta\|^2\right]\right] & \text{if } \varepsilon > 0 \\ \inf_{f\in\mathcal{F}} \mathbb{E}_{\xi\sim P}\left[\min\{\frac{1}{2}\|\xi - \zeta\|^2 : \zeta \in \arg\max f\}\right] & \text{otherwise} \end{cases} \tag{30}$$

where $\pi_\sigma^g(\mathrm{d}\zeta|\xi) \propto e^{g(\zeta)}\pi_\sigma(\mathrm{d}\zeta|\xi)$ is a conditional probability distribution parametrized by an $\Xi \to \mathbb{R}$ function $g$, i.e.,

$$\mathbb{E}_{\zeta\sim\pi_\sigma^g(\cdot|\xi)}[h(\xi, \zeta)] = \frac{\mathbb{E}_{\zeta\sim\pi_\sigma(\cdot|\xi)}\left[e^{g(\zeta)}h(\xi, \zeta)\right]}{\mathbb{E}_{\zeta'\sim\pi_\sigma(\cdot|\xi)}\left[e^{g(\zeta')}\right]}.$$

For this part of the proof, the case when $\varepsilon = 0$ differs from the regularized one $\varepsilon > 0$. We thus present them in separate sections Sections D.1 and D.2.

## D.1 Standard WDRO case

The main result of this section in the standard WDRO case is Proposition D.1 below.

**Proposition D.1.** *Let Assumption 5 hold and fix a threshold $\delta \in (0,1)$. Assume that*

$$\rho^2 \leq \rho_c^2(0,0) - \frac{2B(\delta)}{\sqrt{n}}$$

$$\text{with} \quad L := \frac{16 \sup_{f \in \mathcal{F}} \mathbb{E}_{\xi \sim P}\left[\frac{1}{2}d^2(\xi, \arg\max f)\right]}{\mu}$$

$$\text{and} \quad B(\delta) := 48\mathcal{I}(\mathcal{F}, D) + 2\sqrt{C^\star \log 1/\delta}.$$

*where $\mathcal{I}(\mathcal{F}, D)$ and $D$ are defined in Section A.4, and $\lambda_2 > 0$ is a constant depending on $\Xi$, $\mathcal{F}$, $L_3$, $\mu$ and $C^\star$.*

*Then, with probability at least $1 - \delta$, we have*

$$\forall f \in \mathcal{F}, \quad \widehat{\mathcal{R}}_{\rho^2}^\varepsilon(f) = \inf_{\underline{\lambda} \leq \lambda} \lambda\rho^2 + \mathbb{E}_{\xi \sim \widehat{P}_n}\left[\phi(f, \xi, \lambda, 0)\right]$$

*where the dual bound $\underline{\lambda}$ is defined as*

$$\underline{\lambda} := \min\left(\lambda_2, \frac{\rho_c^2(0,0) - \rho^2}{2L}\right).$$

Before proceeding with the proof, we need to prove the following lemma which leverages Assumption 5.

**Lemma D.2.** *Fix $f \in \mathcal{F}$ and $\xi \in \operatorname{supp} P$. There exists a constant $\lambda_2 > 0$ depending on $\Xi$, $\mathcal{F}$, $L_3$, $\mu$ and $C^\star$ such that, for $\lambda \in [0, \lambda_2]$,*

$$\min\left\{\frac{1}{2}\|\xi - \zeta\|^2 : \zeta \in \arg\max_\Xi f - \frac{\lambda}{2}\|\xi - \cdot\|^2\right\} \geq \left(1 - \frac{16\lambda}{\mu}\right)\min\left\{\frac{1}{2}\|\xi - \zeta\|^2 : \zeta \in \arg\max_\Xi f\right\}.$$

*Proof.* Fix $f \in \mathcal{F}$ and $\xi \in \operatorname{supp} P$. Define, for convenience, $\Xi^\star := \arg\max f$ and

$$Y(\lambda) := \min\left\{\frac{1}{2}\|\xi - \zeta\|^2 : \zeta \in \arg\max_\Xi f - \frac{\lambda}{2}\|\xi - \cdot\|^2\right\}.$$

Step 1: Localization in a $\mathcal{O}(1)$-neighborhood of $\Xi^\star$. For a fixed $R^* > 0$ that will be chosen later, we show that, for $\lambda$ small enough, $Y(\lambda)$ is equal to

$$\min\left\{\frac{1}{2}\|\xi - \zeta\|^2 : \zeta \in \arg\max_{(\Xi^\star)^{R^*}} f - \frac{\lambda}{2}\|\xi - \cdot\|^2\right\} \quad \text{where} \quad (\Xi^\star)^{R^*} := \{\xi \in \Xi : d(\xi, \Xi^\star) \leq R^*\}.$$

Indeed, by Assumption 5, there is some $\Delta(R^*) > 0$ such that for all $f \in \mathcal{F}$ and $\zeta \in \Xi \setminus (\Xi^\star)^{R^*}$,

$$f(\zeta) - \max f - \frac{\lambda}{2}\|\xi - \zeta\|^2 \leq f(\zeta) - \max f \leq -\Delta(R^*),$$

while, for any $\xi^\star \in \Xi^\star$,

$$f(\xi^\star) - \max f - \frac{\lambda}{2}\|\xi - \xi^\star\|^2 = -\frac{\lambda}{2}\|\xi - \xi^\star\|^2 \geq -\lambda C^\star.$$

Hence, for $\lambda \leq \frac{\Delta(R^*)}{C^\star}$, $f(\zeta) - \frac{\lambda}{2}\|\xi - \zeta\|^2 \leq -\Delta(R^*) + \max f \leq -\lambda C^\star + \max f \leq f(\xi^\star) - \frac{\lambda}{2}\|\xi - \xi^\star\|^2$. This means that points in $\Xi \setminus (\Xi^\star)^{R^*}$ cannot maximize $f - \frac{\lambda}{2}\|\xi - \cdot\|^2$ and so it suffices to consider the $\arg\max$ over $(\Xi^\star)^{R^*}$ in the definition of $Y(\lambda)$.

Step 2: Localization in a $\mathcal{O}(\lambda)$-neighborhood of $\Xi^\star$. Take $\zeta^\star \in \arg\max_{(\Xi^\star)^{R^*}} f - \frac{\lambda}{2}\|\xi - \cdot\|^2$. Since $d(\zeta^\star, \Xi^\star) \leq R^*$, the Euclidean projection of $\zeta^\star$ on $\Xi^\star$, that we denote by $\xi^\star$, is at most at distance

$R^*$ of $\zeta^\star$ and $\zeta^\star - \xi^\star \in \widehat{\mathrm{NC}}_{\xi^\star}(\Xi^\star)$, see e.g., Rockafellar and Wets (1998, Thm. 6.12). By the growth condition of Assumption 5, we get that

$$f(\xi^\star) \geq f(\zeta^\star) + \frac{\mu}{2}\|\zeta^\star - \xi^\star\|^2 - \frac{L_3}{6}\|\zeta^\star - \xi^\star\|^3 \,. \tag{31}$$

But, by definition of $\zeta^\star$, we also have that

$$f(\zeta^\star) - \frac{\lambda}{2}\|\xi - \zeta^\star\|^2 \geq f(\xi^\star) - \frac{\lambda}{2}\|\xi - \xi^\star\|^2 \,.$$

Plugging (31) we get that

$$-\frac{\lambda}{2}\|\xi - \zeta^\star\|^2 \geq -\frac{\lambda}{2}\|\xi - \xi^\star\|^2 + \frac{\mu}{2}\|\zeta^\star - \xi^\star\|^2 - \frac{L_3}{6}\|\zeta^\star - \xi^\star\|^3 \,.$$

Rearranging and developing $\frac{1}{2}\|\xi - \xi^\star\|^2$ yields

$$\frac{L_3}{6}\|\zeta^\star - \xi^\star\|^3 + \lambda\langle\xi - \zeta^\star, \zeta^\star - \xi^\star\rangle \geq \frac{\lambda + \mu}{2}\|\xi^\star - \zeta^\star\|^2 \,,$$

which gives, by Cauchy-Schwarz inequality,

$$\frac{L_3}{6}\|\zeta^\star - \xi^\star\|^3 + \lambda\|\xi - \zeta^\star\|\|\zeta^\star - \xi^\star\| \geq \frac{\lambda + \mu}{2}\|\xi^\star - \zeta^\star\|^2 \,, \tag{32}$$

We now wish to obtain a bound on $u^\star := \|\zeta^\star - \xi^\star\|$. If it is zero, there is nothing to do. Otherwise, assuming that it is positive, (32) gives the inequation

$$\frac{\mu + \lambda}{2}u^\star \leq \frac{L_3}{6}(u^\star)^2 + \lambda\|\xi - \xi^\star\| \,.$$

When $\frac{(\mu+\lambda)^2}{4} - \frac{2L_3\lambda}{3}\|\xi - \xi^\star\|$ is non-negative, this inequation is satisfied for

$$u^\star \notin \left[\frac{(\mu + \lambda) \pm \sqrt{(\mu + \lambda)^2 - 8L_3\lambda\|\xi - \xi^\star\|/3}}{L_3/3}\right] \,.$$

Hence, in particular, if $u^\star \leq \frac{3\mu}{L_3}$, then $u^\star$ must be less or equal than

$$\frac{(\mu + \lambda) - \sqrt{(\mu + \lambda)^2 - 8L_3\lambda\|\xi - \xi^\star\|/3}}{L_3/3} = \frac{3(\mu + \lambda)}{L_3}\left(1 - \sqrt{1 - \frac{8L_3\lambda\|\xi - \xi^\star\|}{3(\mu + \lambda)^2}}\right) \leq \frac{8\lambda\|\xi - \xi^\star\|}{\mu + \lambda}$$

when $\frac{8L_3\lambda\|\xi - \xi^\star\|}{3(\mu+\lambda)^2} \leq 1$, using that $1 - \sqrt{1 - x} \leq x$ for $x \in [0, 1]$.

Thus, assuming that $\lambda$ is small enough so that $8L_3\lambda C^\star \leq 3(\mu)^2$ and choosing $R^* := \frac{3\mu}{L_3}$ so that $u^\star \leq \frac{3\mu}{L_3}$ by construction, we have that for any $\zeta^\star \in \arg\max_{(\Xi^\star)^{R^*}} f - \frac{\lambda}{2}\|\xi - \cdot\|^2$, there is a point $\xi^\star \in \Xi^\star$ such that

$$\|\zeta^\star - \xi^\star\| \leq \frac{8\lambda\|\xi - \xi^\star\|}{\mu} \,.$$

Step 3: Conclusion. Defining the constant

$$\lambda_2 := \min\left(\frac{\Delta(R^*)}{C^\star}, \frac{3\mu^2}{8L_3 C^\star}, \frac{\mu}{16}\right),$$

and using the previous steps, we have for any $\lambda \in [0, \lambda_2]$ and any $\zeta^\star \in \arg\max_\Xi f - \frac{\lambda}{2}\|\xi - \cdot\|^2$

$$\frac{1}{2}\|\xi - \zeta^\star\|^2 = \frac{1}{2}\|\xi - \zeta^\star\|^2 = \frac{1}{2}\|\xi - \xi^\star\|^2 + \frac{1}{2}\|\xi^\star - \zeta^\star\|^2 - \langle\xi - \xi^\star, \xi^\star - \zeta^\star\rangle$$

$$\geq \frac{1}{2}\|\xi - \xi^\star\|^2 - \|\xi - \xi^\star\|\|\xi^\star - \zeta^\star\|$$

$$\geq \left(1 - \frac{16\lambda}{\mu}\right)\frac{1}{2}\|\xi - \xi^\star\|^2$$

which concludes the proof. $\qquad\square$

We can now turn to the proof of our proposition.

*Proof of Proposition D.1.* Let $0 \leq \lambda \leq \underline{\lambda}$. For $f \in \mathcal{F}$ and $\lambda \geq 0$, we define $\hat{\psi}_\rho : \lambda \mapsto \lambda \rho^2 + \mathbb{E}_{\xi \sim \widehat{P}_n} [\phi(f, \xi, \lambda, 0)]$ and its (right-sided) derivative $\partial_\lambda \hat{\psi}_\rho$. This derivative is given by,

$$\partial_\lambda \hat{\psi}_\rho(\lambda) = \rho^2 - \mathbb{E}_{\xi \sim \widehat{P}_n} \left[ \min \left\{ \frac{1}{2} \|\xi - \zeta\|^2 : \zeta \in \arg\max_\Xi f - \frac{\lambda}{2} \|\xi - \cdot\|^2 \right\} \right]$$

$$\leq \rho^2 - \left( 1 - \frac{16\lambda}{\mu} \right) \mathbb{E}_{\xi \sim \widehat{P}_n} \left[ \min \left\{ \frac{1}{2} \|\xi - \zeta\|^2 : \zeta \in \arg\max_\Xi f \right\} \right] \qquad (33)$$

where we used Lemma D.2 with $\lambda \leq \underline{\lambda} \leq \lambda_2$.

We then instantiate Lemma G.2 with $X(f, \xi) \leftarrow \frac{1}{2} d^2(\xi, \arg\max f)$, $(\mathcal{X}, \text{dist}) \leftarrow (\mathcal{F}, D)$, whose requirements are checked since:

1. For any $f \in \mathcal{F}$, $X(f, \cdot)$ is measurable since the functions of $\mathcal{F}$ are continuous and thus $\arg\max f$ is *a fortiori* measurable;

2. By definition of $D$, for any $\xi \in \text{supp} \, P$, $f \mapsto d(\xi, \arg\max f)$ is 1-Lipschitz w.r.t. this distance so that $X(\xi, \cdot)$ is $\sqrt{2C^\star}$-Lipschitz.

3. By construction, the range of values $X$ is included in $[0, C^\star]$.

We can thus apply statement $(b)$ of Lemma G.2 to have that, with probability at least $1 - \delta$, for all $f \in \mathcal{F}$,

$$\mathbb{E}_{\xi \sim \widehat{P}_n} \left[ \frac{1}{2} d^2(\xi, \arg\max f) \right] \geq \mathbb{E}_{\xi \sim P} \left[ \frac{1}{2} d^2(\xi, \arg\max f) \right] - \frac{B(\delta)}{\sqrt{n}}$$

Hence, putting this bound together with (33) yields

$$\partial_\lambda \hat{\psi}_\rho(\lambda) \leq \rho^2 - \left( 1 - \frac{16\lambda}{\mu} \right) \mathbb{E}_{\xi \sim P} \left[ \frac{1}{2} d^2(\xi, \arg\max f) \right] + \frac{B(\delta)}{\sqrt{n}}$$

$$\leq \rho^2 - \rho_c^2 + L\lambda + \frac{B(\delta)}{\sqrt{n}} \, ,$$

which is non-negative for $\lambda \leq \underline{\lambda}$. $\qquad \square$

## D.2 Regularized case

The main bound on $\lambda$ of this section are given by Proposition D.3.

**Proposition D.3.** *Fix a threshold $\delta \in (0, 1)$. Assume that $\varepsilon_0 > 0$ and that*

$$\rho^2 \leq \rho_c^2(\varepsilon, \sigma) - \left( \frac{48\sqrt{\text{Var}(\varepsilon, \sigma)} \mathcal{I}(\mathcal{F}, \|\cdot\|_\infty)}{\varepsilon \sqrt{n}} + 2C_\mathcal{F}(\varepsilon, \sigma) \sqrt{\frac{\log \frac{1}{\delta}}{2n}} \right)$$

*where $\mathcal{I}(\mathcal{F}, \|\cdot\|_\infty)$ is defined in Section A.1.*

*Then, with probability at least $1 - \delta$, we have*

$$\forall f \in \mathcal{F}, \quad \widehat{\mathcal{R}}_{\rho^2}^\varepsilon(f) = \inf_{\underline{\lambda}_n \leq \lambda \leq \overline{\lambda}} \lambda \rho^2 + \mathbb{E}_{\xi \sim \widehat{P}_n} [\phi(f, \xi, \lambda, \varepsilon, \sigma)]$$

*where the dual bounds are defined by*

$$\underline{\lambda}_n := \frac{\varepsilon}{\text{Var}(\varepsilon, \sigma)} \left( \rho_c^2(\varepsilon, \sigma) - \rho^2 - \left( \frac{48\sqrt{\text{Var}(\varepsilon, \sigma)} \mathcal{I}(\mathcal{F}, \|\cdot\|_\infty)}{\varepsilon \sqrt{n}} + 2C_\mathcal{F}(\varepsilon, \sigma) \sqrt{\frac{\log \frac{1}{\delta}}{2n}} \right) \right)$$

*and* $\quad \overline{\lambda} := \max \left( \frac{12\varepsilon}{R^2} \log(2 \times 6^{d/2}), e^{\frac{\sup_{f \in \mathcal{F}} \|f\|_\infty}{\varepsilon}} \frac{\varepsilon_0}{\rho} \right).$

*Proof.* Lower-bound: By Assumption 6, for any $f \in \mathcal{F}$, $\xi \in \operatorname{supp} \mathrm{P}$, $\lambda \mapsto \phi(f, \xi, \lambda, \varepsilon, \sigma)$ is twice differentiable and its derivatives are for any $\lambda \geq 0$

$$\partial_\lambda \phi(f, \xi, \lambda, \varepsilon, \sigma) = -\mathbb{E}_{\zeta \sim \pi_\sigma^{\frac{f - \lambda \|\xi - \cdot\|^2 / 2}{\varepsilon}} (\cdot | \xi)} \left[ \frac{1}{2} \|\xi - \zeta\|^2 \right]$$

$$\partial_\lambda^2 \phi(f, \xi, \lambda, \varepsilon, \sigma) = \frac{1}{\varepsilon} \operatorname{Var}_{\zeta \sim \pi_\sigma^{\frac{f - \lambda \|\xi - \cdot\|^2 / 2}{\varepsilon}} (\cdot | \xi)} [\frac{1}{2} \|\xi - \zeta\|^2],$$

and using $\operatorname{Var}(\varepsilon, \sigma)$ which is defined in Section A.1, we get that, for any $\lambda \geq 0$,

$$0 \leq \partial_\lambda^2 \phi(f, \xi, \lambda, \varepsilon, \sigma) \leq \frac{1}{\varepsilon} \operatorname{Var}(\varepsilon, \sigma).$$

As a consequence,

$$\partial_\lambda \left\{ \lambda \rho^2 + \mathbb{E}_{\xi \sim \widehat{\mathrm{P}}_n} [\phi(f, \xi, \lambda, \varepsilon, \sigma)] \right\} = \rho^2 + \mathbb{E}_{\xi \sim \widehat{\mathrm{P}}_n} [\partial_\lambda \phi(f, \xi, \lambda, \varepsilon, \sigma)]$$

$$\leq \rho^2 + \mathbb{E}_{\xi \sim \widehat{\mathrm{P}}_n} [\partial_\lambda \phi(f, \xi, 0, \varepsilon, \sigma)] + \frac{\lambda}{\varepsilon} \operatorname{Var}(\varepsilon, \sigma)$$

$$= \rho^2 - \mathbb{E}_{\xi \sim \widehat{\mathrm{P}}_n} \left[ \mathbb{E}_{\zeta \sim \pi_\sigma^{f/\varepsilon} (\cdot | \xi)} \left[ \frac{1}{2} \|\xi - \zeta\|^2 \right] \right] + \frac{\lambda}{\varepsilon} \operatorname{Var}(\varepsilon, \sigma).$$

$$(34)$$

Now, we want to instante Lemma G.2 with $X(f, \xi) := \mathbb{E}_{\zeta \sim \pi_\sigma^{f/\varepsilon} (\cdot | \xi)} \frac{1}{2} \|\xi - \zeta\|^2$, $(\mathcal{X}, \operatorname{dist}) \leftarrow (\mathcal{F}, \|\cdot\|_\infty)$, whose requirements are checked since:

1. For any $f \in \mathcal{F}$, $\mathbb{E}_{\zeta \sim \pi_\sigma^{f/\varepsilon} (\cdot | \xi)} \frac{1}{2} \|\xi - \zeta\|^2$ is measurable since the functions of $\mathcal{F}$ are continuous and thus *a fortiori* measurable;

2. To show that $f \mapsto X(f, \xi)$ is $\frac{1}{\varepsilon} \sqrt{\operatorname{Var}(\varepsilon, \sigma)}$-Lipschitz, we take $f, g \in \mathcal{F}$ and define, for $t \in [0, 1]$, $f_t = f + t(g - f)$. Since, $\|f - g\|_\infty < +\infty$ and $\sup_{\xi \in \operatorname{supp} \mathrm{P}} \mathbb{E}_{\zeta \sim \pi_\sigma^{f_t/\varepsilon} (\cdot | \xi)} \frac{1}{2} \|\xi - \zeta\|^2 < +\infty$ by compactness of $\Xi$, Assumption 1, $t \mapsto X(f_t, \xi)$ is differentiable with derivative,

$$\frac{\mathrm{d}}{\mathrm{d}t} X(f_t, \xi) = \frac{1}{\varepsilon} \mathbb{E}_{\zeta \sim \pi_\sigma^{f_t/\varepsilon} (\cdot | \xi)} \left[ \frac{1}{2} \|\xi - \zeta\|^2 (g(\zeta) - f(\zeta)) \right]$$

$$- \frac{1}{\varepsilon} \mathbb{E}_{\zeta \sim \pi_\sigma^{f_t/\varepsilon} (\cdot | \xi)} \left[ \frac{1}{2} \|\xi - \zeta\|^2 \right] \mathbb{E}_{\zeta \sim \pi_\sigma^{f_t/\varepsilon} (\cdot | \xi)} [g(\zeta) - f(\zeta)]$$

$$= \frac{1}{\varepsilon} \mathbb{E}_{\zeta \sim \pi_\sigma^{f_t/\varepsilon} (\cdot | \xi)} \left[ \left( \frac{1}{2} \|\xi - \zeta\|^2 - \mathbb{E}_{\zeta' \sim \pi_\sigma^{f_t/\varepsilon} (\cdot | \xi)} \left[ \frac{1}{2} \|\xi - \zeta'\|^2 \right] \right) \right.$$

$$\left. \times \left( (g(\zeta) - f(\zeta)) - \mathbb{E}_{\zeta' \sim \pi_\sigma^{f_t/\varepsilon} (\cdot | \xi)} [g(\zeta') - f(\zeta')] \right) \right].$$

By using Cauchy-Schwarz inequality, we get that,

$$\frac{\mathrm{d}}{\mathrm{d}t} X(f_t, \xi) \leq \frac{1}{\varepsilon} \sqrt{\operatorname{Var}_{\zeta \sim \pi_\sigma^{f_t/\varepsilon} (\cdot | \xi)} [\frac{1}{2} \|\xi - \zeta\|^2]} \sqrt{\operatorname{Var}_{\zeta \sim \pi_\sigma^{f_t/\varepsilon} (\cdot | \xi)} [g(\zeta) - f(\zeta)]}$$

$$\leq \frac{1}{\varepsilon} \sqrt{\operatorname{Var}_{\zeta \sim \pi_\sigma^{f_t/\varepsilon} (\cdot | \xi)} [\frac{1}{2} \|\xi - \zeta\|^2]} \sqrt{\mathbb{E}_{\zeta \sim \pi_\sigma^{f_t/\varepsilon} (\cdot | \xi)} \left[ (g(\zeta) - f(\zeta))^2 \right]}$$

$$\leq \frac{1}{\varepsilon} \sqrt{\operatorname{Var}_{\zeta \sim \pi_\sigma^{f_t/\varepsilon} (\cdot | \xi)} [\frac{1}{2} \|\xi - \zeta\|^2]} \|g - f\|_\infty,$$

which gives the desired Lipschitz condition;

3. The random variables $X(f, \xi)$ lie between 0 and $C_\mathcal{F}(\varepsilon, \sigma)$, which is defined in Section A.1.

We can thus apply statement $(b)$ of Lemma G.2 to have that, with probability at least $1 - \delta$, for all $f \in \mathcal{F}$

$$\mathbb{E}_{\xi \sim \mathrm{P}}\left[\mathbb{E}_{\zeta \sim \pi_\sigma^{f/\varepsilon}(\cdot|\xi)}\frac{1}{2}\|\xi - \zeta\|^2\right] - \mathbb{E}_{\xi \sim \widehat{\mathrm{P}}_n}\left[\mathbb{E}_{\zeta \sim \pi_\sigma^{f/\varepsilon}(\cdot|\xi)}\frac{1}{2}\|\xi - \zeta\|^2\right]$$

$$\leq \frac{48\sqrt{\mathrm{Var}(\varepsilon,\sigma)}\mathcal{I}(\mathcal{F}, \|\cdot\|_\infty)}{\varepsilon\sqrt{n}} + 2C_\mathcal{F}(\varepsilon,\sigma)\sqrt{\frac{\log\frac{1}{\delta}}{2n}}. \tag{35}$$

Combining (34) and (35), we obtain that with probability at least $1 - \delta$

$$\partial_\lambda\left\{\lambda\rho^2 + \mathbb{E}_{\xi \sim \widehat{\mathrm{P}}_n}\left[\phi(f, \xi, \lambda, \varepsilon, \sigma)\right]\right\}$$

$$\leq \rho^2 - \mathbb{E}_{\xi \sim \mathrm{P}}\left[\mathbb{E}_{\zeta \sim \pi_\sigma^{f/\varepsilon}(\cdot|\xi)}\left[\frac{1}{2}\|\xi - \zeta\|^2\right]\right] + \frac{\lambda}{\varepsilon}\mathrm{Var}(\varepsilon,\sigma) + \frac{48\sqrt{\mathrm{Var}(\varepsilon,\sigma)}\mathcal{I}(\mathcal{F}, \|\cdot\|_\infty)}{\varepsilon\sqrt{n}} + 2C_\mathcal{F}(\varepsilon,\sigma)\sqrt{\frac{\log\frac{1}{\delta}}{2n}}$$

$$\leq \rho^2 - \rho_c^2(\varepsilon,\sigma) + \frac{\lambda}{\varepsilon}\mathrm{Var}(\varepsilon,\sigma) + \frac{48\sqrt{\mathrm{Var}(\varepsilon,\sigma)}\mathcal{I}(\mathcal{F}, \|\cdot\|_\infty)}{\varepsilon\sqrt{n}} + 2C_\mathcal{F}(\varepsilon,\sigma)\sqrt{\frac{\log\frac{1}{\delta}}{2n}}$$

$$= \frac{1}{\varepsilon}\mathrm{Var}(\varepsilon,\sigma)\left(\lambda - \underline{\lambda}_n\right)$$

where $\underline{\lambda}_n \geq 0$ is as defined in the statement of the result.

Hence, for all $0 \leq \lambda \leq \underline{\lambda}_n$, the derivative of $\lambda \mapsto \lambda\rho^2 + \mathbb{E}_{\xi \sim \widehat{\mathrm{P}}_n}\left[\phi(f, \xi, \lambda, \varepsilon, \sigma)\right]$ is negative; and since this function is convex, this means that its minimizers are greater than $\underline{\lambda}_n$ with probability at least $1 - \delta$ which is our result.

Upper-bound: Almost surely, for any $f \in \mathcal{F}$, let us begin by bounding the $\partial_\lambda\phi(f, \xi, \lambda, \varepsilon, \sigma)$ for $\lambda \geq 0$, $f \in \mathcal{F}$ and $\xi \in \mathrm{supp}\,\mathrm{P}$. Its expression is given by

$$-\partial_\lambda\phi(f, \xi, \lambda, \varepsilon, \sigma) = \mathbb{E}_{\zeta \sim \pi_\sigma^{\frac{f - \lambda\|\xi - \cdot\|^2/2}{\varepsilon}}(\cdot|\xi)}\left[\frac{1}{2}\|\xi - \zeta\|^2\right] \leq e^{\frac{\|f\|_\infty}{\varepsilon}}\frac{\int_\Xi \frac{1}{2}\|\xi - \zeta\|^2 e^{-\left(\frac{\lambda}{\varepsilon} + \frac{1}{\sigma^2}\right)\frac{1}{2}\|\xi - \zeta\|^2}\mathrm{d}\zeta}{\int_\Xi e^{-\left(\frac{\lambda}{\varepsilon} + \frac{1}{\sigma^2}\right)\frac{1}{2}\|\xi - \zeta\|^2}\mathrm{d}\zeta}.$$

On the one hand, we lower-bound the denominator using Lemma G.1 and Assumption 3 as

$$\frac{1}{(2\pi)^{d/2}}\left(\frac{\lambda}{\varepsilon} + \frac{1}{\sigma^2}\right)^{d/2}\int_\Xi e^{-\left(\frac{\lambda}{\varepsilon} + \frac{1}{\sigma^2}\right)\frac{1}{2}\|\xi - \zeta\|^2}\mathrm{d}\zeta \geq 1 - 6^{d/2}e^{-\frac{R^2}{12}\left(\frac{\lambda}{\varepsilon} + \frac{1}{\sigma^2}\right)} \geq \frac{1}{2},$$

where we used that $\lambda \geq \frac{12\varepsilon}{R^2}\log(2 \times 6^{d/2})$.

On the other hand, the denominator is upper-bounded as

$$\frac{1}{(2\pi)^{d/2}}\left(\frac{\lambda}{\varepsilon} + \frac{1}{\sigma^2}\right)^{d/2}\int_\Xi \frac{1}{2}\|\xi - \zeta\|^2 e^{-\left(\frac{\lambda}{\varepsilon} + \frac{1}{\sigma^2}\right)\frac{1}{2}\|\xi - \zeta\|^2}\mathrm{d}\zeta \leq \frac{1}{2}\left(\frac{\lambda}{\varepsilon} + \frac{1}{\sigma^2}\right)^{-1} \leq \frac{\varepsilon}{2\lambda}.$$

Hence, we have shown that $-\partial_\lambda\phi(f, \xi, \lambda, \varepsilon, \sigma) \leq e^{\frac{\|f\|_\infty}{\varepsilon}}\frac{\varepsilon}{\lambda}$ and, as a consequence,

$$\rho^2 + \mathbb{E}_{\xi \sim \widehat{\mathrm{P}}_n}[\partial_\lambda\phi(f, \xi, \lambda, \varepsilon, \sigma)] \geq \rho^2 - e^{\frac{\|f\|_\infty}{\varepsilon}}\frac{\varepsilon}{\lambda},$$

which is non-negative for $\lambda \geq e^{\frac{\|f\|_\infty}{\varepsilon}}\frac{\varepsilon_0}{\rho}$.

Hence, for

$$\lambda \geq \overline{\lambda} := \max\left(\frac{12\varepsilon}{R^2}\log(2 \times 6^{d/2}), e^{\frac{\sup_{f \in \mathcal{F}}\|f\|_\infty}{\varepsilon}}\frac{\varepsilon_0}{\rho}\right),$$

the derivative of $\lambda \mapsto \lambda\rho^2 + \mathbb{E}_{\xi \sim \widehat{\mathrm{P}}_n}[\phi(f, \xi, \lambda, \varepsilon, \sigma)]$ is non-negative, which means that its minimizers are smaller than $\overline{\lambda}$. $\qquad\square$

# E  Proof of the main results

In this section, we present our main results with explicit constants. In Section E.1 we treat the case of standard WDRO, i.e., the setting of Theorems 3.1 and 3.3, while in Section E.2 we handle the regularized setting of Theorem 3.4.

## E.1 Standard WDRO case

The main results of this section are Theorems E.1 and E.3 which are more precise versions of Theorems 3.1 and 3.3 respectively.

**Theorem E.1** (Extended version of Theorem 3.1). *Under Assumptions 1, 3 and 6 and the additional Assumptions 4 and 5, with $\rho_c = \rho_c(0,0)$ defined in (30) for any $\delta \in (0,1)$ and $n \geq 1$, if*

$$\max\left(\rho_n, \frac{8192}{\sqrt{n}\mu^*(\lambda_0^*)^2}\left(12\mathcal{I}(\mathcal{F}, \|\cdot\|_\infty) + \left(\widetilde{F}(0) + \overline{M}(\rho_c)\right)\sqrt{1 + \log\frac{4}{\delta}}\right)\right) \leq \rho$$

$$and \qquad \rho \leq \frac{\rho_c}{2} - \frac{96\mathcal{I}(\mathcal{F},D) + 4\sqrt{C^\star \log 1/\delta}}{\sqrt{n}}.$$

*where*

$$\rho_{\text{thres}} := \min\left(\frac{\lambda_0^*}{8(\lambda_1 + L_2)}, \frac{\mu^*(\lambda_0^*)^2}{4096 L_2}\right)$$

$$\overline{a} := \sup_{0 < \rho' \leq \rho_{\text{thres}}} a\left(\frac{1}{\rho'}\min\left(\frac{\lambda_0^*}{32}, \rho_{\text{thres}}\lambda_2, \frac{3\rho_c^2\rho_{\text{thres}}}{8L}\right), \frac{\lambda_0^*}{2\rho'}, 0, 0\right)$$

$$\rho_n := \frac{117\left(\mathcal{I}(\mathcal{F}, \|\cdot\|_\infty) + \max\left(\widetilde{F}\left(\frac{\lambda_0^*}{32\rho_c}\right), \overline{a}\right)\left(1 + \sqrt{\log\frac{1}{\delta}}\right)\right)}{\sqrt{n}\min\left(\frac{\lambda_0^*}{32}, \rho_{\text{thres}}\lambda_2, \frac{3\rho_c^2\rho_{\text{thres}}}{8L}\right)},$$

*then, with probability $1 - \delta$,*

$$\forall f \in \mathcal{F}, \quad \widehat{\mathcal{R}}_{\rho^2}(f) \geq \mathbb{E}_{\xi \sim Q}[f(\xi)] \qquad \text{for all } Q \text{ such that } W_2^2(P, Q) \leq \rho(\rho - \rho_n).$$

*In particular, with probability $1 - \delta$, we have*

$$\forall f \in \mathcal{F}, \quad \widehat{\mathcal{R}}_{\rho^2}(f) \geq \mathbb{E}_{\xi \sim P}[f(\xi)].$$

The proof of Theorem E.1 relies on Lemma E.2 that combines the results of the previous sections, namely propositions B.1,C.1 and, D.1.

**Lemma E.2.** *Under the blanket assumptions Assumptions 1, 3 and 6 and with the additional Assumption 5, for any threshold $\delta \in (0,1)$, define*

$$\underline{\lambda}(\rho) = \begin{cases} \frac{\lambda_0^*}{32\rho} \text{ if } \rho \leq \rho_{\text{thres}} = \min\left(\frac{\lambda_0^*}{32(\lambda_1 + L_2)}, \frac{\mu^*(\lambda_0^*)^2}{4096 L_2}\right) \\ \min\left(\lambda_2, \frac{\rho_c^2(0,0) - \rho^2}{2L}\right) \text{ otherwise} \end{cases} \tag{36}$$

$$\overline{\lambda}(\rho) = \frac{\lambda_0^*}{2\rho}.$$

*Assume that*

$$\rho \geq \frac{8192}{\sqrt{n}\mu^*(\lambda_0^*)^2}\left(12\mathcal{I}(\mathcal{F}, \|\cdot\|_\infty) + \left(\widetilde{F}(0) + \overline{M}(\rho)\right)\sqrt{1 + \log\frac{4}{\delta}}\right), \tag{37}$$

*and that*

$$\rho^2 \leq \rho_c^2(0,0) - \frac{2B(\delta)}{\sqrt{n}}.$$

*Then, with probability at least $1 - \frac{\delta}{2}$,*

$$\forall f \in \mathcal{F}, \quad \widehat{\mathcal{R}}_{\rho^2}(f) = \inf_{\underline{\lambda}(\rho) \leq \lambda \leq \overline{\lambda}(\rho)} \lambda\rho^2 + \mathbb{E}_{\xi \sim \widehat{P}_n}[\phi(f, \xi, \lambda, 0)]$$

*and when $\rho^2 \geq \overline{\rho}_n^2(\delta, \underline{\lambda}(\rho), \overline{\lambda}(\rho), 0)$, with probability $1 - \delta$, it holds,*

$$\widehat{\mathcal{R}}_{\rho^2}(f) \geq \mathcal{R}_{\rho^2 - \overline{\rho}_n^2(\delta, \underline{\lambda}(\rho), \overline{\lambda}(\rho), 0)}(f),.$$

*Furthermore, with probability $1 - \delta$,*

$$\forall f \in \mathcal{F}, \quad \widehat{\mathcal{R}}_{\rho^2}(f) \geq \sup\{\mathbb{E}_Q[f] : Q \in \mathcal{P}(\Xi), W_2^2(P, Q) \leq \rho^2 - \overline{\rho}_n^2(\delta, \underline{\lambda}(\rho), \overline{\lambda}(\rho), 0)\}.$$

*Proof.* This result is a consequence of Propositions C.1 and D.1 both applied with $\delta \leftarrow \delta/4$ and of Proposition B.1. Note that the upper-bound on the dual variable given by Proposition C.1 holds for any $\rho$ since the optimal dual variable is non-increasing as a function of $\rho$. □

*Proof of Theorem E.1.* The proof consists in simplifying both the assumptions and the result of Lemma E.2.

We begin by showing that $\underline{\lambda}(\rho)$ can always be lower-bounded by a quantity proportional to $1/\rho$. Indeed, by definition of $\underline{\lambda}(\rho)$, (36) in Lemma E.2, and using that $\rho$ is in particular less than $\frac{\rho_c}{2}$, it holds that,

$$\underline{\lambda}(\rho) \geq \frac{1}{\rho}\min\left(\frac{\lambda_0^*}{32}, \rho_{\text{thres}}\lambda_2, \frac{3\rho_c^2\rho_{\text{thres}}}{8L}\right) \tag{38}$$

Let us now turn our attention to the condition $\rho^2 \geq \overline{\rho}_n^2(\delta, \underline{\lambda}(\rho), \overline{\lambda}(\rho), 0, 0)$, whose RHS was defined by (24) in Section B. We have that, by definition (Assumption 6), $\sup_{0<\rho\leq\rho_c} \widetilde{F}(\lambda_0^*/(32\rho)) = \widetilde{F}(\lambda_0^*/(32\rho_c)) < +\infty$ and,

$$\sup_{0<\rho\leq\rho_c} a\big(\underline{\lambda}_n(\rho), \overline{\lambda}(\rho), 0, 0\big) \leq \sup_{0<\rho'\leq\rho_{\text{thres}}} a\left(\frac{1}{\rho}\min\left(\frac{\lambda_0^*}{32}, \rho_{\text{thres}}\lambda_2, \frac{3\rho_c^2\rho_{\text{thres}}}{8L}\right), \frac{\lambda_0^*}{2\rho'}, 0, 0\right)$$
$$= \overline{a} < +\infty\,,$$

by definition and non-decreasingness of $a$ in its first argument (see Corollary A.5) and (38). Hence, the following bound holds

$$\overline{\rho}_n^2(\delta, \underline{\lambda}(\rho), \overline{\lambda}(\rho), \varepsilon, \sigma)$$
$$\leq \frac{117}{\sqrt{n}\underline{\lambda}(\rho)}\left(\mathcal{I}(\mathcal{F}, \|\cdot\|_\infty) + \max\left(\widetilde{F}\left(\frac{\lambda_0^*}{32\rho_c}\right), \overline{a}\right)\left(1 + \sqrt{\log\frac{1}{\delta}}\right)\right)$$
$$\leq \rho_n\rho\,,$$

where we plugged (38).

Finally, since $\rho$ is in particular bounded by $\rho_c$, the condition (37) is implied by

$$\rho \geq \frac{8192}{\sqrt{n}\mu^*(\lambda_0^*)^2}\left(12\mathcal{I}(\mathcal{F}, \|\cdot\|_\infty) + \left(\widetilde{F}(0) + \overline{M}(\rho_c)\right)\sqrt{1 + \log\frac{4}{\delta}}\right),$$

with $\overline{M}(\rho_c) < +\infty$ by definition (Proposition C.1). □

**Theorem E.3** (Extended version of Theorem 3.3). *Under Assumptions 1, 3 and 6, for any $\delta \in (0, 1)$ and $n \geq 1$, if*

$$\max\left(\rho_n, \frac{8192}{\sqrt{n}\mu^*(\lambda_0^*)^2}\left(12\mathcal{I}(\mathcal{F}, \|\cdot\|_\infty) + \left(\widetilde{F}(0) + \overline{M}(\rho_c)\right)\sqrt{1 + \log\frac{2}{\delta}}\right)\right) \leq \rho$$

$$\text{and} \qquad \rho \leq \min\left(\rho_{\text{thres}}, \frac{\rho_c}{2} - \frac{96\mathcal{I}(\mathcal{F}, D) + 4\sqrt{C^\star \log 1/\delta}}{\sqrt{n}}\right).$$

*where*

$$\rho_{\text{thres}} := \min\left(\frac{\lambda_0^*}{8(\lambda_1 + L_2)}, \frac{\mu^*(\lambda_0^*)^2}{4096L_2}\right)$$

$$\overline{a} := \sup_{0<\rho'\leq\rho_{\text{thres}}} a\left(\frac{\lambda_0^*}{32\rho'}, \frac{\lambda_0^*}{2\rho'}, 0, 0\right)$$

$$\rho_n := \frac{3744\left(\mathcal{I}(\mathcal{F}, \|\cdot\|_\infty) + \max\left(\widetilde{F}\left(\frac{\lambda_0^*}{32\rho_{\text{thres}}}\right), \overline{a}\right)\left(1 + \sqrt{\log\frac{1}{\delta}}\right)\right)}{\sqrt{n}\lambda_0^*,}\,,$$

*then, with probability $1 - \delta$,*

$$\forall f \in \mathcal{F}, \quad \widehat{\mathcal{R}}_{\rho^2}(f) \geq \mathbb{E}_{\xi\sim Q}[f(\xi)] \qquad \text{for all } Q \text{ such that } W_2^2(P, Q) \leq \rho(\rho - \rho_n)\,.$$

*In particular, with probability $1 - \delta$, we have*

$$\forall f \in \mathcal{F}, \quad \widehat{\mathcal{R}}_{\rho^2}(f) \geq \mathbb{E}_{\xi\sim P}[f(\xi)]\,.$$

The proof of Theorem E.3 leverages results from the previous sections, combined in Lemma E.4.

**Lemma E.4.** *Under the blanket assumptions Assumptions 1, 3 and 6, for any threshold $\delta \in (0,1)$, define*

$$\underline{\lambda}(\rho) = \frac{\lambda_0^*}{32\rho}, \qquad \overline{\lambda}(\rho) = \frac{\lambda_0^*}{2\rho}.$$

*Assume that*

$$\rho \geq \frac{8192}{\sqrt{n}\mu^*(\lambda_0^*)^2}\left(12\mathcal{I}(\mathcal{F}, \|\cdot\|_\infty) + \left(\widetilde{F}(0) + \overline{M}(\rho)\right)\sqrt{1 + \log\frac{2}{\delta}}\right), \tag{39}$$

*and that*

$$\rho^2 \leq \min\left(\rho_c^2(0,0) - \frac{2B(\delta)}{\sqrt{n}}, \min\left(\frac{\lambda_0^*}{32(\lambda_1 + L_2)}, \frac{\mu^*(\lambda_0^*)^2}{4096L_2}\right)^2\right).$$

*Then, with probability at least $1 - \frac{\delta}{2}$,*

$$\forall f \in \mathcal{F}, \quad \widehat{\mathcal{R}}_{\rho^2}(f) = \inf_{\underline{\lambda}(\rho) \leq \lambda \leq \overline{\lambda}(\rho)} \lambda\rho^2 + \mathbb{E}_{\xi \sim \widehat{P}_n}\left[\phi(f, \xi, \lambda, 0)\right]$$

*and when $\rho^2 \geq \overline{\rho}_n^2(\delta, \underline{\lambda}(\rho), \overline{\lambda}(\rho), 0)$, with probability $1 - \delta$, it holds,*

$$\widehat{\mathcal{R}}_{\rho^2}(f) \geq \mathcal{R}_{\rho^2 - \overline{\rho}_n^2(\delta, \underline{\lambda}(\rho), \overline{\lambda}(\rho), 0)}(f),.$$

*Furthermore, with probability $1 - \delta$,*

$$\forall f \in \mathcal{F}, \quad \widehat{\mathcal{R}}_{\rho^2}(f) \geq \sup\left\{\mathbb{E}_Q[f] : Q \in \mathcal{P}(\Xi), W_2^2(P, Q) \leq \rho^2 - \overline{\rho}_n^2(\delta, \underline{\lambda}(\rho), \overline{\lambda}(\rho), 0)\right\}.$$

*Proof.* This result follows directly from Proposition C.1 that we invoke with $\delta \leftarrow \delta/2$ and of Proposition B.1. $\qquad\square$

*Proof of Theorem E.3.* The proof consists in simplifying both the assumptions and the result of Lemma E.4 and follows the same structure as the proof of Theorem E.1.

We begin by examining the condition $\rho^2 \geq \overline{\rho}_n^2(\delta, \underline{\lambda}(\rho), \overline{\lambda}(\rho), 0, 0)$, whose RHS was defined by (24) in Section B. We have that, by definition (Assumption 6), $\sup_{0 < \rho \leq \rho_c} \widetilde{F}(\lambda_0^*/(32\rho)) = \widetilde{F}(\lambda_0^*/(32\rho_c)) < +\infty$ and,

$$\sup_{0 < \rho \leq \rho_\text{thres}} a\left(\underline{\lambda}_n(\rho), \overline{\lambda}(\rho), 0, 0\right) = \sup_{0 < \rho' \leq \rho_\text{thres}} a\left(\frac{\lambda_0^*}{32\rho'}, \frac{\lambda_0^*}{2\rho'}, 0, 0\right) = \overline{a} < +\infty$$

by definition (see Corollary A.5). Hence, we have that

$$\overline{\rho}_n^2(\delta, \underline{\lambda}(\rho), \overline{\lambda}(\rho), 0, 0)$$
$$\leq \frac{117}{\sqrt{n}\underline{\lambda}(\rho)}\left(\mathcal{I}(\mathcal{F}, \|\cdot\|_\infty) + \max\left(\widetilde{F}\left(\frac{\lambda_0^*}{32\rho_c}\right), \overline{a}\right)\left(1 + \sqrt{\log\frac{1}{\delta}}\right)\right)$$
$$\leq \rho_n\rho,$$

by definition of $\rho_n$ and with $117 \times 32 = 3744$.

Finally, m (39) is implied by

$$\rho \geq \frac{8192}{\sqrt{n}\mu^*(\lambda_0^*)^2}\left(12\mathcal{I}(\mathcal{F}, \|\cdot\|_\infty) + \left(\widetilde{F}(0) + \overline{M}(\rho_\text{thres})\right)\sqrt{1 + \log\frac{2}{\delta}}\right),$$

since $\rho \leq \rho_\text{thres}$ and $\overline{M}(\rho_\text{thres}) < +\infty$ by definition (Proposition C.1). $\qquad\square$

## E.2 Regularized WDRO case

**Theorem E.5** (Extended version of Theorem 3.4). *For $\sigma = \sigma_0\rho$ with $\sigma_0 > 0$, $\varepsilon = \varepsilon_0\rho$ with $\varepsilon_0 > 0$ such that $\varepsilon_0/\sigma_0^2 \leq \lambda_0^*/8$, and for any $\delta \in (0,1)$ and $n \geq 1$, define,*

$$\rho_c := \inf\{\rho_c\left(\varepsilon_0\rho', \sigma_0\rho'\right) : \rho_{\mathrm{thres}} \leq \rho' \leq \rho_c\left(\varepsilon_0\rho_{\mathrm{thres}}, \sigma_0\rho_{\mathrm{thres}}\right)\}$$

*and*

$$\rho_{\mathrm{thres}} := \min\left(\frac{\varepsilon_1}{\varepsilon_0}, \frac{\lambda_0^*}{32(\lambda_1 + L_2)}, \frac{\mu^*(\lambda_0^*)^2}{4096 L_2}, \sqrt{\frac{c_2^3\lambda_0^*}{8\varepsilon_0}}\left(\log\left(\frac{4096\varepsilon_0 c_1}{\mu^*(\lambda_0^*)^2}\right)\right)_+^{-\frac{3}{2}}\right)$$

$$\overline{\mathrm{Var}} := \sup_{\rho_{\mathrm{thres}} \leq \rho' \leq \rho_c} \mathrm{Var}(\varepsilon_0\rho', \sigma_0\rho')$$

$$\overline{C}_{\mathcal{F}} := \sup_{\rho_{\mathrm{thres}} \leq \rho' \leq \rho_c} C_{\mathcal{F}}(\varepsilon_0\rho', \sigma_0\rho')$$

$$\overline{a} := \sup_{0 < \rho' \leq \rho_c} a\left(\frac{1}{\rho'}\min\left(\frac{\lambda_0^*}{32}, \frac{\varepsilon_0\rho_{\mathrm{thres}}^2\rho_c^2}{4\overline{\mathrm{Var}}}\right), \max\left(\frac{\lambda_0^*}{2\rho'}, \frac{12\varepsilon_0\rho_c\log(2 \times 6^{d/2})}{R^2}, \frac{e^{\frac{\sup_{f \in \mathcal{F}}\|f\|_\infty}{\varepsilon_0\rho_{\mathrm{thres}}}}\frac{\varepsilon_0}{\rho_{\mathrm{thres}}}}{}\right), \varepsilon_0\rho', \sigma_0\rho'\right)$$

$$\rho_n := \frac{117\left(\mathcal{I}(\mathcal{F}, \|\cdot\|_\infty) + \max\left(\widetilde{F}\left(\frac{\lambda_0^*}{32\rho_c}\right), \overline{a}\right)\left(1 + \sqrt{\log\frac{1}{\delta}}\right)\right)}{\sqrt{n}\min\left(\frac{\lambda_0^*}{32}, \frac{\varepsilon_0\rho_{\mathrm{thres}}^2\rho_c^2}{4\overline{\mathrm{Var}}}\right)};$$

*when*

$$\max\left(\rho_n, \frac{8192}{\mu^*(\lambda_0^*)^2\sqrt{n}}\left(12\mathcal{I}(\mathcal{F}, \|\cdot\|_\infty) + \left(\widetilde{F}(0) + \overline{M}(\rho_c)\right)\sqrt{\log\frac{4}{\delta}}\right), \frac{384\sqrt{\overline{\mathrm{Var}}}\mathcal{I}(\mathcal{F}, \|\cdot\|_\infty)}{\varepsilon_0\rho_c^2\sqrt{n}}\right) \leq \rho$$

$$\rho \leq \frac{\rho_c}{2} - \frac{384\sqrt{\overline{\mathrm{Var}}}\mathcal{I}(\mathcal{F}, \|\cdot\|_\infty)}{\varepsilon_0\rho_c^2\sqrt{n}}$$

*and* $\quad \rho_c \geq \max\left(\left(\frac{192\sqrt{\overline{\mathrm{Var}}}\mathcal{I}(\mathcal{F}, \|\cdot\|_\infty)}{\varepsilon_0\sqrt{n}}\right)^{1/3}, 2\sqrt{\overline{C}_{\mathcal{F}}}\left(\frac{\log\frac{4}{\delta}}{2n}\right)^{1/4}\right),$

*then, with probability at least $1 - \delta$,*

$$\forall f \in \mathcal{F}, \quad \widehat{\mathcal{R}}_{\rho^2}^\varepsilon(f) \geq \mathbb{E}_{\xi \sim Q}[f(\xi)] \qquad \text{for all } Q \text{ such that } W_{2,\tau(\rho)}^2(P, Q) \leq \rho(\rho - \rho_n),$$

*where* $\tau(\rho) \leq \frac{\varepsilon\rho}{\min\left(\frac{\lambda_0^*}{32}, \frac{\varepsilon_0\rho_{\mathrm{thres}}^2\rho_c^2}{4\overline{\mathrm{Var}}}\right)}$. *Furthermore, when $\sigma_0 \leq 1$ and $\sigma \leq \sigma_1$ (defined in Proposition A.2), with probability $1 - \delta$,*

$$\forall f \in \mathcal{F}, \quad \widehat{\mathcal{R}}_{\rho^2}^\varepsilon(f) \geq \mathbb{E}_{\xi \sim P}\mathbb{E}_{\zeta \sim \pi_\sigma(\cdot|\xi)}[f(\zeta)].$$

The proof of Theorem E.5 relies on Lemma E.6 that makes the regularized Wasserstein distance appear. It also uses Lemma E.7, to guarantee that a smoothed version of the true distribution is inside the right neighborhood.

**Lemma E.6.** *Fix a confidence threshold $\delta \in (0,1)$, take $\varepsilon = \varepsilon_0\rho$, $\sigma = \sigma_0\rho$ with $\varepsilon_0$ and $\sigma_0$ positive constants satisfying $\varepsilon_0/\sigma_0^2 \leq \lambda_0^*/8$ and, define $\underline{\lambda}_n(\rho)$ and $\overline{\lambda}(\rho)$ as functions of $\rho$ by*

- *If*

$$\rho \leq \min\left(\frac{\varepsilon_1}{\varepsilon_0}, \frac{\lambda_0^*}{32(\lambda_1 + L_2)}, \frac{\mu^*(\lambda_0^*)^2}{4096 L_2}, \sqrt{\frac{c_2^3\lambda_0^*}{8\varepsilon_0}}\left(\log\left(\frac{4096\varepsilon_0 c_1}{\mu^*(\lambda_0^*)^2}\right)\right)_+^{-\frac{3}{2}}\right),$$

  *then $\underline{\lambda}_n(\rho) = \frac{\lambda_0^*}{32\rho}$ and $\overline{\lambda}(\rho) = \frac{\lambda_0^*}{2\rho}$,*

- *Otherwise,*

$$\underline{\lambda}_n(\rho) = \frac{\varepsilon_0\rho}{\mathrm{Var}(\varepsilon_0\rho, \sigma_0\rho)}\left(\rho_c(\varepsilon_0\rho, \sigma_0\rho)^2 - \rho^2 - \left(\frac{48\sqrt{\mathrm{Var}(\varepsilon_0\rho, \sigma_0\rho)}\mathcal{I}(\mathcal{F}, \|\cdot\|_\infty)}{\varepsilon_0\rho\sqrt{n}} + 2C_{\mathcal{F}}(\varepsilon_0\rho, \sigma_0\rho)\sqrt{\frac{\log\frac{4}{\delta}}{2n}}\right)\right)$$

$$\overline{\lambda}(\rho) = \max\left(\frac{12\varepsilon_0\rho}{R^2}\log(2 \times 6^{d/2}), e^{\frac{\sup_{f \in \mathcal{F}}\|f\|_\infty}{\varepsilon_0\rho}}\frac{\varepsilon_0}{\rho}\right).$$

*Assume that*

$$\rho \geq \frac{8192}{\sqrt{n}\mu^*(\lambda_0^*)^2} \left(12\mathcal{I}(\mathcal{F},\|\cdot\|_\infty) + \left(\widetilde{F}(0) + \overline{M}(\rho)\right)\sqrt{1 + \log\frac{4}{\delta}}\right), \tag{40}$$

*Then, with probability at least $1 - \frac{\delta}{2}$,*

$$\forall f \in \mathcal{F}, \quad \widehat{\mathcal{R}}_{\rho^2}^\varepsilon(f) = \inf_{\underline{\lambda}_n(\rho)\leq\lambda\leq\overline{\lambda}(\rho)} \lambda\rho^2 + \mathbb{E}_{\xi\sim\widehat{P}_n}\left[\phi(f,\xi,\lambda,\varepsilon,\sigma)\right]$$

*and when $\rho^2 \geq \overline{\rho}_n^2(\delta,\underline{\lambda}_n(\rho),\overline{\lambda}(\rho),\varepsilon,\sigma)$, with probability $1 - \delta$, it holds,*

$$\widehat{\mathcal{R}}_{\rho^2}^\varepsilon(f) \geq \mathcal{R}_{\rho^2 - \overline{\rho}_n^2(\delta,\underline{\lambda}_n(\rho),\overline{\lambda}(\rho),\varepsilon,\sigma)}^\varepsilon(f),.$$

*Furthermore, with probability $1 - \delta$,*

$$\forall f \in \mathcal{F}, \quad \widehat{\mathcal{R}}_{\rho^2}^\varepsilon(f) \geq \sup\left\{\mathbb{E}_Q[f] : Q \in \mathcal{P}(\Xi), W_{2,\tau(\rho)}^2(P,Q) \leq \rho^2 - \overline{\rho}_n^2(\delta,\underline{\lambda}_n(\rho),\overline{\lambda}(\rho),\varepsilon,\sigma)\right\},$$

*with $\tau(\rho) := \frac{\varepsilon_0\rho}{\underline{\lambda}_n(\rho)}$.*

*Proof.* The first part of this result is a consequence of the combination of Propositions C.1 and D.3, both applied with $\delta \leftarrow \delta/4$, and of Proposition B.1. For the second part, note that Remark B.4 implies that the above argument actually gives the slightly stronger result: with probability $1 - \delta$, for any $f \in \mathcal{F}$,

$$\widehat{\mathcal{R}}_{\rho^2}^\varepsilon(f) \geq \inf_{\underline{\lambda}_n(\rho)\leq\lambda\leq\overline{\lambda}(\rho)} \lambda(\rho^2 - \overline{\rho}_n^2(\delta,\underline{\lambda}_n(\rho),\overline{\lambda}(\rho),\varepsilon,\sigma)) + \mathbb{E}_{\xi\sim P}[\phi(f,\xi,\lambda,\varepsilon,\sigma)]$$

Next, take $Q \in \mathcal{P}(\Xi)$ such that $W_{2,\tau(\rho)}^2(P,Q) \leq \rho^2 - \overline{\rho}_n^2(\delta,\underline{\lambda}_n(\rho),\varepsilon,\sigma)$. With a similar argument as in the proof of Proposition B.1, we get that

$$\begin{aligned}
\widehat{\mathcal{R}}_{\rho^2}^\varepsilon(f) &\geq \inf_{\underline{\lambda}_n(\rho)\leq\lambda\leq\overline{\lambda}(\rho)} \lambda(\rho^2 - \overline{\rho}_n^2(\delta,\underline{\lambda}_n(\rho),\overline{\lambda}(\rho),\varepsilon,\sigma) + \mathbb{E}_{\xi\sim P}[\phi(f,\xi,\lambda,\varepsilon,\sigma)] \\
&= \mathbb{E}_Q[f] + \inf_{\underline{\lambda}_n(\rho)\leq\lambda\leq\overline{\lambda}(\rho)} \lambda(\rho^2 - \overline{\rho}_n^2(\delta,\underline{\lambda}_n(\rho),\overline{\lambda}(\rho),\varepsilon,\sigma) - \{\mathbb{E}_Q[f] - \mathbb{E}_{\xi\sim P}[\phi(f,\xi,\lambda,\varepsilon,\sigma)]\} \\
&= \mathbb{E}_Q[f] + \inf_{\underline{\lambda}_n(\rho)\leq\lambda\leq\overline{\lambda}(\rho)} \lambda(\rho^2 - \overline{\rho}_n^2(\delta,\underline{\lambda}_n(\rho),\overline{\lambda}(\rho),\varepsilon,\sigma)) - \sup_{f'\in\mathcal{F}}\{\mathbb{E}_Q[f'] - \mathbb{E}_{\xi\sim P}[\phi(f',\xi,\lambda,\varepsilon,\sigma)]\}.
\end{aligned}$$

We now proceed to show, and this will conclude the proof, that

$$\sup_{f\in\mathcal{F}}\{\mathbb{E}_Q[f] - \mathbb{E}_{\xi\sim P}[\phi(f,\xi,\lambda,\varepsilon,\sigma)]\} \leq \lambda W_{2,\tau(\rho)}^2(P,Q),$$

for $\lambda \geq \underline{\lambda}_n(\rho)$.

Indeed,

$$\begin{aligned}
\sup_{f\in\mathcal{F}}\{\mathbb{E}_Q[f] - \mathbb{E}_{\xi\sim P}[\phi(f,\xi,\lambda,\varepsilon,\sigma)]\} &\leq \sup_{f\in\mathcal{C}(\Xi)}\{\mathbb{E}_Q[f] - \mathbb{E}_{\xi\sim P}[\phi(f,\xi,\lambda,\varepsilon,\sigma)]\} \\
&= \sup_{f\in\mathcal{C}(\Xi)}\left\{\mathbb{E}_Q[f] - \mathbb{E}_{\xi\sim P}\left[\log\left(\mathbb{E}_{\zeta\sim\pi_\sigma(\cdot|\xi)}\left[e^{\frac{f(\zeta)-\lambda\|\xi-\zeta\|^2/2}{\varepsilon}}\right]\right)\right]\right\} \\
&= \lambda\sup_{f\in\mathcal{C}(\Xi)}\left\{\mathbb{E}_Q[f] - \mathbb{E}_{\xi\sim P}\left[\log\left(\mathbb{E}_{\zeta\sim\pi_\sigma(\cdot|\xi)}\left[e^{\frac{f(\zeta)-\|\xi-\zeta\|^2/2}{\varepsilon/\lambda}}\right]\right)\right]\right\}.
\end{aligned} \tag{41}$$

where we performed the change of variable $f \leftarrow f/\lambda$. We now show the following equality that will allow us to rewrite the RHS of (41).

$$-\mathbb{E}_{\xi\sim P}\left[\log\left(\mathbb{E}_{\zeta\sim\pi_\sigma(\cdot|\xi)}\left[e^{\frac{f(\zeta)-\frac{1}{2}\|\xi-\zeta\|^2}{\varepsilon/\lambda}}\right]\right)\right] = \sup_{g\in\mathcal{C}(\Xi)} \mathbb{E}_P[g] - \frac{\varepsilon}{\lambda}\left(\mathbb{E}_{(\xi,\zeta)\sim\pi_\sigma}\left[e^{\frac{g(\xi)+f(\zeta)-\frac{1}{2}\|\xi-\zeta\|^2}{\varepsilon/\lambda}}\right] - 1\right). \tag{42}$$

Solving the optimality condition of the concave problem of the RHS of (42) gives that its maximum is reached for

$$g(\xi) = -\log\left(\mathbb{E}_{\zeta\sim\pi_\sigma(\cdot|\xi)}\left[e^{\frac{f(\zeta)-\frac{1}{2}\|\xi-\zeta\|^2}{\varepsilon/\lambda}}\right]\right)$$

so that (42) holds. Hence, we get that

$$\sup_{f\in\mathcal{C}(\Xi)}\left\{\mathbb{E}_Q[f] - \mathbb{E}_{\xi\sim P}\left[\log\left(\mathbb{E}_{\zeta\sim\pi_\sigma(\cdot|\xi)}\left[e^{\frac{f(\zeta)-\frac{1}{2}\|\xi-\zeta\|^2}{\varepsilon/\lambda}}\right]\right)\right]\right\}$$

$$= \sup_{f,g\in\mathcal{C}(\Xi)}\left\{\mathbb{E}_Q[f] + \mathbb{E}_P[g] - \frac{\varepsilon}{\lambda}\left(\mathbb{E}_{(\xi,\zeta)\sim\pi_\sigma}\left[e^{\frac{g(\xi)+f(\zeta)-\frac{1}{2}\|\xi-\zeta\|^2}{\varepsilon/\lambda}}\right] - 1\right)\right\}$$

$$= W_{2,\varepsilon/\lambda}^2(P,Q)\,,$$

by the duality formula for regularized OT (Peyré and Cuturi, 2019)[2].

Combining this equality with the bound of (41) gives

$$\sup_{f\in\mathcal{F}}\{\mathbb{E}_Q[f] - \mathbb{E}_{\xi\sim P}[\phi(f,\xi,\lambda,\varepsilon,\sigma)]\} \leq \lambda W_{2,\varepsilon/\lambda}^2(P,Q)\,,$$

which yields the result since $W_{2,\tau}^2(P,Q)$ is non-decreasing in $\tau$. $\qquad\square$

**Lemma E.7.** *In the setting of Theorem E.5, when* $Q_\sigma$ *denotes the second marginal of*

$$P(d\xi)\,\pi_\sigma(d\zeta|\xi)\,,$$

*and when* $\sigma \leq \sigma_1$*, it holds*

$$W_{2,\tau(\rho)}^2(P,Q_\sigma) \leq \sigma^2\,.$$

*Proof.* Consider the transport plan $\pi = P(d\xi)\,\pi_\sigma(d\zeta|\xi)$. To show this lemma, it suffices to prove that

$$\mathbb{E}_\pi\left[d^2\right] + \tau(\rho)\,\mathrm{KL}(\pi\,|\,\pi) = \mathcal{O}\left(\sigma^2\right)\,,$$

i.e., that $\mathbb{E}_\pi\left[d^2\right] = \mathcal{O}\left(\sigma^2\right)$. Let us first fix $\xi \in \mathrm{supp}\,P$ and consider $\mathbb{E}_{\zeta\sim\pi_\sigma(\cdot|\xi)}\left[\frac{1}{2}\|\xi-\zeta\|^2\right]$, which is equal to

$$\mathbb{E}_{\zeta\sim\pi_\sigma(\cdot|\xi)}\left[\frac{1}{2}\|\xi-\zeta\|^2\right] = \frac{\int_\Xi \frac{1}{2}\|\xi-\zeta\|^2 e^{-\frac{\|\xi-\zeta\|^2}{2\sigma^2}}d\zeta}{\int_\Xi e^{-\frac{\|\xi-\zeta\|^2}{2\sigma^2}}d\zeta}\,.$$

The numerator can be upper-bounded as follows:

$$\int_\Xi \frac{1}{2}\|\xi-\zeta\|^2 e^{-\frac{\|\xi-\zeta\|^2}{2\sigma^2}}d\zeta \leq \int_{\mathbb{R}^d} \frac{1}{2}\|\xi-\zeta\|^2 e^{-\frac{\|\xi-\zeta\|^2}{2\sigma^2}}d\zeta = (2\pi\sigma^2)^{d/2}\frac{\sigma^2}{2}\,.$$

For the denominator, we have seen in the proof of Lemma A.4, and more precisely (21), that

$$\left(\int_\Xi e^{-\frac{\|\xi-\zeta\|^2}{2\sigma^2}}d\zeta\right)^{-1} \leq \frac{2}{(2\pi\sigma^2)^{d/2}}\,,$$

when $\sigma \leq \sigma_1$. Hence, we have the bound

$$\mathbb{E}_{\zeta\sim\pi_\sigma(\cdot|\xi)}\left[\frac{1}{2}\|\xi-\zeta\|^2\right] \leq \sigma^2\,,$$

and integrating w.r.t. $\xi \sim P$ yields the result. $\qquad\square$

*Proof of Theorem E.5.* Since we will only consider radii in particular bounded by $\rho_c$, the condition (40) is implied by

$$\rho \geq \frac{8192}{\mu^*(\lambda_0^*)^2\sqrt{n}}\left(12\mathcal{I}(\mathcal{F},\|\cdot\|_\infty) + \left(\widetilde{F}(0) + \overline{M}(\rho_c)\right)\sqrt{\log\frac{4}{\delta}}\right)\,,$$

---

[2]To get this exact result for a regularization w.r.t. an arbitrary measure, one can readily combine Paty and Cuturi (2020, Cor. 1) and Feydy et al. (2019, Prop. 7). Also, note that we essentially reproved the semi-duality formula of Genevay et al. (2016, Prop. 2.1) except that the regularization is taken w.r.t. a general measure.

with $\overline{M}(\rho_c) < +\infty$.

We now show that $\underline{\lambda}_n(\rho)$ can always be lower-bounded by a quantity proportional to $1/\rho$, i.e., that

$$\underline{\lambda}_n(\rho) \geq \frac{1}{\rho} \min\left(\frac{\lambda_0^*}{32}, \frac{\varepsilon_0 \rho_{\mathrm{thres}}^2 \rho_c^2}{4\overline{\mathrm{Var}}}\right). \qquad (43)$$

Let us discuss separately the cases where $\rho \leq \rho_{\mathrm{thres}}$ holds or not.

- When $\rho \leq \rho_{\mathrm{thres}}$, (43) holds by definition of $\underline{\lambda}_n(\rho)$.

- When $\rho > \rho_{\mathrm{thres}}$, by definition, $\underline{\lambda}_n(\rho)$ is lower bounded as

$$\underline{\lambda}_n(\rho) \geq \frac{\varepsilon_0 \rho}{\overline{\mathrm{Var}}}\left(\rho_c^2 - \rho^2 - \left(\frac{48\sqrt{\overline{\mathrm{Var}}}\mathcal{I}(\mathcal{F}, \|\cdot\|_\infty)}{\varepsilon_0 \rho\sqrt{n}} + 2\overline{C}_\mathcal{F}\sqrt{\frac{\log\frac{4}{\delta}}{2n}}\right)\right).$$

Applying Lemma G.9 with $\overline{\rho} \leftarrow \frac{\rho_c}{2}$ and $c \leftarrow \frac{48\sqrt{\overline{\mathrm{Var}}}\mathcal{I}(\mathcal{F}, \|\cdot\|_\infty)}{\varepsilon_0\sqrt{n}}$, we obtain that, when

$$\rho_c \geq \left(\frac{192\sqrt{\overline{\mathrm{Var}}}\mathcal{I}(\mathcal{F}, \|\cdot\|_\infty)}{\varepsilon_0\sqrt{n}}\right)^{1/3}$$

and

$$\frac{384\sqrt{\overline{\mathrm{Var}}}\mathcal{I}(\mathcal{F}, \|\cdot\|_\infty)}{\varepsilon_0 \rho_c^2\sqrt{n}} \leq \rho \leq \frac{\rho_c}{2} - \frac{384\sqrt{\overline{\mathrm{Var}}}\mathcal{I}(\mathcal{F}, \|\cdot\|_\infty)}{\varepsilon_0 \rho_c^2\sqrt{n}},$$

the following lower-bound holds,

$$\underline{\lambda}_n(\rho) \geq \frac{\varepsilon_0 \rho}{\overline{\mathrm{Var}}}\left(\frac{3\rho_c^2}{4} - 2\overline{C}_\mathcal{F}\sqrt{\frac{\log\frac{4}{\delta}}{2n}}\right) \geq \frac{\varepsilon_0 \rho \rho_c^2}{4\overline{\mathrm{Var}}} \geq \frac{\varepsilon_0 \rho_{\mathrm{thres}}^2 \rho_c^2}{4\overline{\mathrm{Var}}\rho},$$

where we used successively that $\frac{\rho_c^2}{2} \geq 2\overline{C}_\mathcal{F}\sqrt{\frac{\log\frac{4}{\delta}}{2n}}$ and $\rho \geq \rho_{\mathrm{thres}}$. This concludes the proof of (43). Note that it implies the bound on $\tau(\rho)$ in the statement.

Let us finally turn our attention to the condition $\rho^2 \geq \overline{\rho}_n^2(\delta, \underline{\lambda}_n(\rho), \overline{\lambda}(\rho), \varepsilon, \sigma)$. Since $\sup_{0<\rho\leq\rho_c} \widetilde{F}(\lambda_0^*/(32\rho)) = \widetilde{F}(\lambda_0^*/(32\rho_c)) < +\infty$ by definition (Assumption 6) and

$$\sup_{0<\rho'\leq\rho_c} a\big(\underline{\lambda}_n(\rho'), \overline{\lambda}(\rho'), \varepsilon_0\rho', \varepsilon_0\sigma\big)$$

$$\leq \sup_{0<\rho'\leq\rho_c} a\left(\frac{1}{\rho'}\min\left(\frac{\lambda_0^*}{32}, \frac{\varepsilon_0\rho_{\mathrm{thres}}^2\rho_c^2}{4\overline{\mathrm{Var}}}\right), \max\left(\frac{\lambda_0^*}{2\rho'}, \frac{12\varepsilon_0\rho_c\log(2\times 6^{d/2})}{R^2}, \frac{e^{\frac{\sup_{f\in\mathcal{F}}\|f\|_\infty}{\varepsilon_0\rho_{\mathrm{thres}}}}\varepsilon_0}{\rho_{\mathrm{thres}}}\right), \varepsilon_0\rho', \sigma_0\rho'\right)$$

$$= \overline{a} < +\infty$$

where we used the monotonicity properties of $a$ (Corollary A.5) and (43).

In conclusion, along with (43), we obtain that

$$\overline{\rho}_n^2(\delta, \underline{\lambda}(\rho), \overline{\lambda}(\rho), \varepsilon, \sigma)$$

$$\leq \frac{117}{\sqrt{n}\underline{\lambda}(\rho)}\left(\mathcal{I}(\mathcal{F}, \|\cdot\|_\infty) + \max\left(\widetilde{F}\left(\frac{\lambda_0^*}{32\rho_c}\right), \overline{a}\right)\left(1 + \sqrt{\log\frac{1}{\delta}}\right)\right)$$

$$\leq \rho_n\rho,$$

by definition of $\rho_n$. The last part of the statement then follows from Lemma E.7. $\qquad\square$

# F Upper-bound on the empirical robust risk

In this section we prove Theorem 3.5 that complements the main results by providing both a lwoer and an upper bound on the empirical robus risk. In view of the previous section, the missing part is ther upper-bound, that we establish in this section.

The proof of the upper-bound is similar to the proof of our main results, yet simpler. Indeed, the bounds on the dual variable are required for the true distribution P, which is fixed, instead of the empirical distribution $\widehat{P}_n$. We slightly modify our main concentration result (Proposition B.1) in Proposition F.1. We simplify our bounds on the dual multiplier when the radius is close to the critical radius (Propositions D.1 and D.3) in Propositions F.2 and F.5.

## F.1 From empirical to true risk

**Proposition F.1.** *For $\rho > 0$, $\varepsilon \geq 0$, $\sigma > 0$ and $\delta \in (0,1)$, assume that there is some $0 < \underline{\lambda} \leq \overline{\lambda} < +\infty$ such that,*

$$\forall f \in \mathcal{F}, \quad \mathcal{R}^\varepsilon_{\rho^2}(f) = \inf_{\underline{\lambda} \leq \lambda \leq \overline{\lambda}} \lambda \rho^2 + \mathbb{E}_{\xi \sim P}\left[\phi(f, \xi, \lambda, \varepsilon, \sigma)\right].$$

*Then, when $\rho^2 \geq \overline{\rho}_n^2(\delta, \underline{\lambda}, \overline{\lambda}, \varepsilon, \sigma)$, with probability $1 - \delta$,*

$$\forall f \in \mathcal{F}, \quad \widehat{\mathcal{R}}^\varepsilon_{\rho^2 - \overline{\rho}_n^2(\delta, \underline{\lambda}, \overline{\lambda}, \varepsilon, \sigma)}(f) \leq \mathcal{R}^\varepsilon_{\rho^2}(f).$$

*Proof.* This proof closely mimics the one of Proposition B.1 but switches the roles of P and $\widehat{P}_n$. First, note that by following the proof of Lemma B.3 with and replacing P by $\widehat{P}_n$ and *vice versa* (and using statement $(a)$ of Lemma G.2 instead of $(b)$) yields the following

$$\sup_{(f,\lambda) \in \mathcal{F} \times [\underline{\lambda}, \overline{\lambda}]} \left\{ \frac{\mathbb{E}_{\xi \sim \widehat{P}_n}[\phi(f, \xi, \lambda, \varepsilon, \sigma)] - \mathbb{E}_{\xi \sim P}[\phi(f, \xi, \lambda, \varepsilon, \sigma)]}{\lambda} \right\} \leq \overline{\rho}_n^2(\delta, \underline{\lambda}, \overline{\lambda}, \varepsilon, \sigma).$$

We can now follow the last part of the proof of Proposition B.1. On the event above, for any $f \in \mathcal{F}$,

$$\mathcal{R}^\varepsilon_{\rho^2}(f) = \inf_{\underline{\lambda} \leq \lambda \leq \overline{\lambda}} \left\{ \lambda \rho^2 + \mathbb{E}_{\xi \sim P}\left[\phi(f, \xi, \lambda, \varepsilon, \sigma)\right] \right\}$$

$$= \inf_{\underline{\lambda} \leq \lambda \leq \overline{\lambda}} \left\{ \lambda \rho^2 + \mathbb{E}_{\xi \sim \widehat{P}_n}\left[\phi(f, \xi, \lambda, \varepsilon, \sigma)\right] - \lambda \frac{\mathbb{E}_{\xi \sim \widehat{P}_n}[\phi(f, \xi, \lambda, \varepsilon, \sigma)] - \mathbb{E}_{\xi \sim P}[\phi(f, \xi, \lambda, \varepsilon, \sigma)]}{\lambda} \right\}$$

$$\geq \inf_{\underline{\lambda} \leq \lambda \leq \overline{\lambda}} \left\{ \lambda \rho^2 + \mathbb{E}_{\xi \sim \widehat{P}_n}\left[\phi(f, \xi, \lambda, \varepsilon, \sigma)\right] - \lambda \sup_{\underline{\lambda} \leq \lambda' \leq \overline{\lambda}} \frac{\mathbb{E}_{\xi \sim \widehat{P}_n}[\phi(f, \xi, \lambda', \varepsilon, \sigma)] - \mathbb{E}_{\xi \sim P}[\phi(f, \xi, \lambda', \varepsilon, \sigma)]}{\lambda'} \right\}$$

$$\geq \inf_{\underline{\lambda} \leq \lambda \leq \overline{\lambda}} \left\{ \lambda \rho^2 + \mathbb{E}_{\xi \sim \widehat{P}_n}\left[\phi(f, \xi, \lambda, \varepsilon, \sigma)\right] - \lambda \overline{\rho}_n^2(\delta, \underline{\lambda}, \overline{\lambda}, \varepsilon, \sigma) \right\}$$

$$\geq \widehat{\mathcal{R}}^\varepsilon_{\rho^2 - \overline{\rho}_n^2(\delta, \underline{\lambda}, \overline{\lambda}, \varepsilon, \sigma)}(f).$$

$\square$

## F.2 Standard WDRO case

**Proposition F.2.** *Let Assumption 5 hold and fix a threshold $\delta \in (0,1)$. Assume that $\rho^2 \leq \rho_c^2(0,0)$. Then, we have,*

$$\forall f \in \mathcal{F}, \quad \mathcal{R}_{\rho^2}(f) = \inf_{\underline{\lambda} \leq \lambda} \lambda \rho^2 + \mathbb{E}_{\xi \sim P}\left[\phi(f, \xi, \lambda, 0)\right]$$

*where the dual bound $\underline{\lambda}$ is defined as*

$$\underline{\lambda} := \min\left(\lambda_2, \frac{\rho_c^2(0,0) - \rho^2}{L}\right),$$

*and $L$ is defined in Proposition D.1.*

*Proof.* Let $0 \leq \lambda \leq \underline{\lambda}$. By Lemma D.2 and the dominated convergence theorem, one has that,

$$\partial_\lambda \psi_\rho(\lambda) = \rho^2 - \mathbb{E}_{\xi \sim P}\left[\min\left\{\frac{1}{2}\|\xi - \zeta\|^2 : \zeta \in \arg\max_\Xi f - \frac{\lambda}{2}\|\xi - \cdot\|^2\right\}\right]$$

$$\leq \rho^2 - \left(1 - \frac{16\lambda}{\mu}\right)\mathbb{E}_{\xi \sim P}\left[\min\left\{\frac{1}{2}\|\xi - \zeta\|^2 : \zeta \in \arg\max_\Xi f\right\}\right]$$

$$\leq \rho^2 - \rho_c^2(0,0) + L\lambda,$$

which is non-negative by definition of $\underline{\lambda}$ and thus concludes the proof. $\square$

We can now state analogues of Theorems E.1 and E.3. Note that the bounds $\underline{\lambda}(\rho)$ that we obtained in this section are better than the ones we got in the main proof. For the sake of simplicity, we give up this additional precision and use the same bounds as in Theorems E.1 and E.3.

**Corollary F.3.** *In the same setting as Theorem E.1, with probability $1 - \delta$, it holds,*

$$\forall f \in \mathcal{F}, \quad \mathcal{R}_{\rho^2}(f) \geq \widehat{\mathcal{R}}_{\rho(\rho-\rho_n)}(f).$$

*Proof.* This result is obtained as a combination of Propositions F.1–F.2, which gives the desired result with probability at least $1 - \frac{\delta}{2}$ and *a fortiori* $1 - \delta$. $\square$

**Corollary F.4.** *In the same setting as Theorem E.3, with probability $1 - \delta$, it holds,*

$$\forall f \in \mathcal{F}, \quad \mathcal{R}_{\rho^2}(f) \geq \widehat{\mathcal{R}}_{\rho(\rho-\rho_n)}(f).$$

*Proof.* This result follows by combining Propositions F.1 and C.1, which gives the desired result with probability at least $1 - \frac{\delta}{2}$ and *a fortiori* $1 - \delta$. $\square$

To conclude, in the context of Theorem E.1 (resp. Theorem E.3), Corollary F.3 (resp. Corollary F.4) with $\rho \leftarrow \rho + \rho_n$ yields, with probability at least $1 - \delta$,

$$\forall f \in \mathcal{F}, \quad \widehat{\mathcal{R}}_{\rho(\rho+\rho_n)}(f) \leq \mathcal{R}_{(\rho+\rho_n)^2}(f),$$

so that, since $\rho \geq \rho_n$,

$$\forall f \in \mathcal{F}, \quad \widehat{\mathcal{R}}_{\rho^2}(f) \leq \mathcal{R}_{\rho(\rho+3\rho_n)}(f),$$

Combining this bound with Theorem E.1 (resp. Theorem E.3) completes the bound of Theorem 3.5.

### F.3 Regularized case

In the regularized case, the bound simplifies as well compared to Proposition D.3.

**Proposition F.5.** *Fix a threshold $\delta \in (0,1)$. When $\rho^2 \leq \rho_c^2(\varepsilon, \sigma)$, we have,*

$$\forall f \in \mathcal{F}, \quad \mathcal{R}_{\rho^2}^\varepsilon(f) = \inf_{\underline{\lambda}_n \leq \lambda \leq \overline{\lambda}} \lambda\rho^2 + \mathbb{E}_{\xi \sim P}\left[\phi(f, \xi, \lambda, \varepsilon, \sigma)\right]$$

*where the dual bounds are defined by*

$$\underline{\lambda} := \frac{\varepsilon}{\text{Var}(\varepsilon, \sigma)}\left(\rho_c^2(\varepsilon, \sigma) - \rho^2\right)$$

$$\text{and} \quad \overline{\lambda} := \max\left(\frac{12\varepsilon}{R^2}\log(2 \times 6^{d/2}), e^{\frac{\sup_{f \in \mathcal{F}} \|f\|_\infty}{\varepsilon}}\frac{\varepsilon_0}{\rho}\right),$$

*and $\lambda_0^*$, $\mu^*$ were defined in Assumption 8.*

*Proof.* The proof of the upper-bound is exactly the same as in Proposition D.3 so we focus on the lower-bound. Following the same reasoning as the one to get (34) in Proposition D.3 but with P instead of $\widehat{P}_n$ we get that

$$\partial_\lambda\left\{\lambda\rho^2 + \mathbb{E}_{\xi \sim P}\left[\phi(f, \xi, \lambda, \varepsilon, \sigma)\right]\right\} \leq \rho^2 - \mathbb{E}_{\xi \sim P}\left[\mathbb{E}_{\zeta \sim \pi_\sigma^{f/\varepsilon}(\cdot|\xi)}\left[\frac{1}{2}\|\xi - \zeta\|^2\right]\right] + \frac{\lambda}{\varepsilon}\text{Var}(\varepsilon, \sigma)$$

$$= \rho^2 - \rho_c^2(\varepsilon, \sigma) + \frac{\lambda}{\varepsilon}\text{Var}(\varepsilon, \sigma),$$

which is non-positive when $\lambda \leq \underline{\lambda}$. $\square$

**Corollary F.6.** *In the same setting as [Theorem E.5](#), with probability $1 - \delta$, it holds,*

$$\forall f \in \mathcal{F}, \quad \mathcal{R}_{\rho^2}^{\varepsilon}(f) \geq \widehat{\mathcal{R}}_{\rho(\rho - \rho_n)}^{\varepsilon}(f).$$

*Proof.* This result follows by combining [Propositions F.1](#) and [C.1](#) and [Proposition F.5](#), which gives the desired result with probability at least $1 - \frac{\delta}{2}$ and *a fortiori* $1 - \delta$. □

To conclude, in the context of [Theorem E.5](#), [Corollary F.6](#) with $\rho \leftarrow \rho + \rho_n$ and $\varepsilon_0 \leftarrow \frac{\varepsilon_0 \rho}{\rho + \rho_n}$, i.e., $\varepsilon \leftarrow \frac{\varepsilon_0 \rho}{\rho + \rho_n} \times (\rho + \rho_n)$, yields,[3] with probability at least $1 - \delta$,

$$\forall f \in \mathcal{F}, \quad \widehat{\mathcal{R}}_{\rho(\rho + \rho_n)}^{\varepsilon_0 \rho}(f) \leq \mathcal{R}_{(\rho + \rho_n)^2}^{\varepsilon_0 \rho}(f), \quad \text{and, in particular,} \quad \widehat{\mathcal{R}}_{\rho^2}^{\varepsilon_0 \rho}(f) \leq \mathcal{R}_{\rho(\rho + 3\rho_n)}^{\varepsilon_0 \rho}(f).$$

Combining this bound with [Theorem E.5](#) completes the bound of [Theorem 3.5](#).

# G  Technical lemmas

In this section, we recall and adapt known results, as well as establish technical facts, all useful in our developments. They are presented in self-contained lemmas and are arranged in four thematic subsections.

## G.1  Laplace approximation

**Lemma G.1** (Restriction to $\Xi$). *Consider $\Xi \subset \mathbb{R}^d$, $\varepsilon_1, \tau_1 > 0$ and a map $\zeta^\star : [0, \tau_1] \to \Xi$ defined by $\zeta^\star(\tau) = \xi + \tau g$ with $\xi \in \Xi$, $g \in \mathbb{R}^d$ and assume that there is a positive radius $R$ such that,*

    *1. The closed ball $\overline{\mathbb{B}}(\zeta^\star(0), R)$ is included in $\Xi$.*

    *2. $R, \tau_1$ and $\|g\|$ satisfy $\frac{R^2}{6} \geq \tau_1^2 \|g\|^2$.*

*Then, for $(\varepsilon, \tau) \in [0, \varepsilon_1] \times [0, \tau_1]$,*

$$\left| (2\pi\varepsilon\tau)^{-\frac{d}{2}} \int_\Xi \exp\left( -\frac{\|\zeta - \zeta^\star(\tau)\|_2^2}{2\varepsilon\tau} \right) d\zeta - 1 \right| \leq 6^{d/2} e^{-\frac{R^2}{12\varepsilon\tau}},$$

*Proof.* The quantity to bound rewrites

$$\left| (2\pi\varepsilon\tau)^{-\frac{d}{2}} \int_\Xi \exp\left( -\frac{\|\zeta - \zeta^\star(\tau)\|_2^2}{2\varepsilon\tau} \right) d\zeta - 1 \right| = (2\pi\varepsilon\tau)^{-\frac{d}{2}} \int_{\mathbb{R}^d \setminus \Xi} \exp\left( -\frac{\|\zeta - \zeta^\star(\tau)\|_2^2}{2\varepsilon\tau} \right) d\zeta,$$

so let us bound this integral. Since $\overline{\mathbb{B}}(\zeta^\star(0), R)$ is inside $\Xi$, this means that, for any $\zeta \notin \Xi$, $\|\zeta - \xi\|$ is at least equal to $R$. Hence, for any $\zeta \notin \Xi$, one has that

$$\begin{aligned}
\|\zeta - \zeta^\star(\tau)\|^2 &\geq \frac{1}{2}\|\zeta - \xi\|^2 - \tau^2 \|g\|^2 \\
&\geq \frac{1}{6}\|\zeta - \xi\|^2 + \frac{1}{6}R^2 + \frac{1}{6}R^2 - \tau^2 \|g\|^2 \\
&\geq \frac{1}{6}\|\zeta - \xi\|^2 + \frac{1}{6}R^2,
\end{aligned}$$

so that we get the bound

$$\begin{aligned}
(2\pi\varepsilon\tau)^{-\frac{d}{2}} \int_{\mathbb{R}^d \setminus \Xi} \exp\left( -\frac{\|\zeta - \zeta^\star(\tau)\|_2^2}{2\varepsilon\tau} \right) d\zeta &\leq e^{-\frac{R^2}{12\varepsilon\tau}} \times (2\pi\varepsilon\tau)^{-\frac{d}{2}} \int_{\mathbb{R}^d \setminus \Xi} \exp\left( -\frac{\|\zeta - \zeta\|_2^2}{12\varepsilon\tau} \right) d\zeta \\
&= 6^{d/2} e^{-\frac{R^2}{12\varepsilon\tau}}.
\end{aligned}$$

□

---

[3]Though $\varepsilon_0$ now formally depends on $\rho'$, the same bounds still hold and do not become degenerate since $\varepsilon_0$ lies $[\varepsilon_0/2, \varepsilon_0]$ that avoids zero.

## G.2 Concentration

We rely on standard concentration tools that we encapsulate in the following lemma for convenience.

**Lemma G.2.** *Let* $(\mathcal{X}, \mathrm{dist})$ *be a (totally bounded) separable metric space,* $\mathrm{P}$ *a probability distribution on a probability space* $\Xi$ *and* $\widehat{\mathrm{P}}_{\mathrm{n}} = \frac{1}{n} \sum_{i=1}^{n} \delta_{\xi_i}$ *with* $\xi_1, \dots, \xi_n \sim \mathrm{P}$ *i.i.d.. Consider a mapping* $X : \mathcal{X} \times \Xi \to \mathbb{R}$ *and assume that,*

1. *For each* $x \in \mathcal{X}$, $\xi \mapsto X(x, \xi)$ *is measurable;*

2. *There is a constant* $L > 0$ *such that, for each* $\xi \in \Xi$, $x \mapsto X(x, \xi)$ *is L-Lipschitz;*

3. $X$ *almost surely belongs to* $[a, b]$.

*Then, for any* $\delta \in (0, 1)$,

(a) *With probability at least* $1 - \delta$,

$$\forall x \in \mathcal{X}, \quad \mathbb{E}_{\xi \sim \widehat{\mathrm{P}}_{\mathrm{n}}}[X(x, \xi)] - \mathbb{E}_{\xi \sim \mathrm{P}}[X(x, \xi)] \le \frac{48 L \mathcal{I}(\mathcal{X}, \mathrm{dist})}{\sqrt{n}} + 2(b - a) \sqrt{\frac{\log \frac{1}{\delta}}{2n}} \,.$$

(b) *With probability at least* $1 - \delta$,

$$\forall x \in \mathcal{X}, \quad \mathbb{E}_{\xi \sim \mathrm{P}}[X(x, \xi)] - \mathbb{E}_{\xi \sim \widehat{\mathrm{P}}_{\mathrm{n}}}[X(x, \xi)] \le \frac{48 L \mathcal{I}(\mathcal{X}, \mathrm{dist})}{\sqrt{n}} + 2(b - a) \sqrt{\frac{\log \frac{1}{\delta}}{2n}} \,.$$

*Proof.* First, let us note that we can assume that $\mathbb{E}_{\xi \sim \mathrm{P}}[X(x, \xi)] = 0$ provided that we prove the bound above with the left-hand side divided by a factor two. Indeed, considering the random variables $Y(x, \xi) := X(x, \xi) - \mathbb{E}_{\zeta \sim \mathrm{P}}[X(x, \zeta)]$, we see that $Y$ satisfy the assumptions of the lemma, albeit with the constants $L \leftarrow 2L$, $a \leftarrow a - b$ and $b \leftarrow b - a$. Moreover, we only prove the assertion $(a)$ since the $(b)$ follows from $(a)$ with $X \leftarrow -X$.

Step 1: Bound on the expectation. First, we focus on bounding the expectation of the quantity

$$\sup_{x \in \mathcal{X}} \{ \mathbb{E}_{\xi \sim \widehat{\mathrm{P}}_{\mathrm{n}}} X(x, \xi) \} \,.$$

By the symmetrization principle (e.g., (Boucheron et al., 2013, Lem. 11.4)), with $\mathrm{s}_1, \dots, \mathrm{s}_n$ i.i.d. Rademacher random variables,

$$\mathbb{E} \left[ \sup_{x \in \mathcal{X}} \left\{ \mathbb{E}_{\xi \sim \widehat{\mathrm{P}}_{\mathrm{n}}} X(x, \xi) \right\} \right] \le 2 \mathbb{E} \left[ \sup_{x \in \mathcal{X}} \frac{1}{n} \sum_{i=1}^{n} \mathrm{s}_i \, X(x, \xi_i) \right] \,.$$

Take $x, x' \in \mathcal{X}$. By the Lipschitz property of $X$, with $\xi \sim \mathrm{P}$, for any $i = 1, \dots, n$, the random variable

$$\frac{\mathrm{s}_i (X(x, \xi) - X(x', \xi))}{\sqrt{n} L} \tag{44}$$

is bounded, in absolute value, by $\frac{\mathrm{dist}(x, x')}{\sqrt{n}}$ and as such it is sub-Gaussian with parameter $\frac{\mathrm{dist}(x, x')^2}{n}$ by Hoeffding's lemma (e.g., (Boucheron et al., 2013, Lem. 2.2)). As a consequence, by independence, the random variable

$$\sum_{i=1}^{n} \frac{\mathrm{s}_i (X(x, \xi) - X(x', \xi))}{\sqrt{n} L}$$

is sub-Gaussian with parameter $\mathrm{dist}(x, x')^2$. Since, in addition, it is zero-mean, we can invoke Dudley's bound (e.g., (Boucheron et al., 2013, Cor. 13.2)) to get that,

$$\mathbb{E} \left[ \sup_{x \in \mathcal{X}} \frac{1}{\sqrt{n} L} \sum_{i=1}^{n} \mathrm{s}_i \, X(x, \xi_i) \right] \le 12 \mathcal{I}(\mathcal{X}, \mathrm{dist}) \,,$$

or, in other words, by (44), that

$$\mathbb{E}\left[\sup_{x\in\mathcal{X}}\{\mathbb{E}_{\xi\sim\widehat{\mathrm{P}}_{\mathrm{n}}}X(x,\xi)\}\right]\leq\frac{24L\mathcal{I}(\mathcal{X},\mathrm{dist})}{\sqrt{n}}. \tag{45}$$

Step 2: Concentration inequality. Since the functions $X$ are uniformly bounded, $\sup_{x\in\mathcal{X}}\left\{\mathbb{E}_{\xi\sim\widehat{\mathrm{P}}_{\mathrm{n}}}X(x,\xi)\right\}$, seen as a function of $(\xi_1,\ldots,\xi_n)$, satisfies the bounded difference property with constant $b-a$. Therefore, the bounded difference inequality (e.g., (Boucheron et al., 2013, Thm. 6.2)) readily yields that, with probability at least $1-\delta$,

$$\sup_{x\in\mathcal{X}}\left\{\mathbb{E}_{\xi\sim\widehat{\mathrm{P}}_{\mathrm{n}}}X(x,\xi)\right\}\leq\mathbb{E}\sup_{x\in\mathcal{X}}\left\{\mathbb{E}_{\xi\sim\widehat{\mathrm{P}}_{\mathrm{n}}}X(x,\xi)\right\}+(b-a)\sqrt{\frac{\log\frac{1}{\delta}}{2n}}$$

$$\leq\frac{24L\mathcal{I}(\mathcal{X},\mathrm{dist})}{\sqrt{n}}+(b-a)\sqrt{\frac{\log\frac{1}{\delta}}{2n}}$$

where we plugged in (45), the bound on the expectation from the first step. □

## G.3 Dudley's integral bounds

**Lemma G.3.** *Let $(\mathcal{X}_1,\mathrm{dist}_1)$ and $(\mathcal{X}_2,\mathrm{dist}_2)$ be two metric spaces, and consider $\mathcal{X}:=\mathcal{X}_1\times\mathcal{X}_2$ equipped with the distance $\mathrm{dist}:=c_1\,\mathrm{dist}_1+c_2\,\mathrm{dist}_2$ with $c_1,c_2>0$. Then*

$$\mathcal{I}(\mathcal{X},\mathrm{dist})\leq c_1\mathcal{I}(\mathcal{X}_1,\mathrm{dist}_1)+c_2\mathcal{I}(\mathcal{X}_2,\mathrm{dist}_2).$$

*Proof.* Note that, for any $t>0$, the inequality $N(t,\mathcal{X},\mathrm{dist})\leq N(t,\mathcal{X}_1,c_1\,\mathrm{dist}_1)\times N(t,\mathcal{X}_2,c_2\,\mathrm{dist}_2)$ holds, so that, by subdadditivity of the square root,

$$\begin{aligned}\mathcal{I}(\mathcal{X},\mathrm{dist})&=\int_0^{+\infty}\sqrt{\log N(t,\mathcal{X},\mathrm{dist})}\mathrm{d}t\\&\leq\int_0^{+\infty}\sqrt{\log N(t,\mathcal{X}_1,c_1\,\mathrm{dist}_1)}\mathrm{d}t+\int_0^{+\infty}\sqrt{\log N(t,\mathcal{X}_2,c_2\,\mathrm{dist}_2)}\mathrm{d}t\\&=\int_0^{+\infty}\sqrt{\log N(t/c_1,\mathcal{X}_1,\mathrm{dist}_1)}\mathrm{d}t+\int_0^{+\infty}\sqrt{\log N(t/c_2,\mathcal{X}_2,\mathrm{dist}_2)}\mathrm{d}t\\&=c_1\mathcal{I}(\mathcal{X}_1,\mathrm{dist}_1)+c_2\mathcal{I}(\mathcal{X}_2,\mathrm{dist}_2),\end{aligned}$$

where we performed changes of variable in to obtain the last equality. □

**Lemma G.4.** *For $c>0$,*

$$\mathcal{I}([0,c],|\cdot|)\leq\frac{c}{2}(1+2\log 2).$$

*Proof.* Noticing that $N(t,[0,c],|\cdot|)=1$ whenever $t\geq c$, we get that

$$\begin{aligned}\mathcal{I}([0,c],|\cdot|)&=\int_0^c\sqrt{\log N(t,[0,c],|\cdot|)}\mathrm{d}t\\&\leq\int_0^c(1+\log N(t,[0,c],|\cdot|))\mathrm{d}t.\end{aligned}$$

Now, a rough bound on $N(t,[0,c],|\cdot|)$ is $1+\frac{c}{t}$ which fits our purpose and yields

$$\mathcal{I}([0,c],|\cdot|)\leq c+\int_0^c\log\left(1+\frac{c}{t}\right)\mathrm{d}t=c(1+2\log 2).$$

□

### G.4 Auxiliary results

We conclude these sections with auxiliary technical results.

The following lemma recalls basic inequalities with the logarithm function.

**Lemma G.5.** *For $0 \leq x \leq \frac{1}{2}$, the following inequalities hold,*

$$\log(1 - x) \geq -2x \qquad and \qquad \log(1 + x) \leq x \,.$$

**Lemma G.6.** *For $\alpha > 0$, the function $x \mapsto \frac{\log(\alpha + x)}{x}$ is non-increasing on $([e^{W(1)} - \alpha]_+, +\infty)$.*

*Proof.* Denote by $f : x \mapsto \frac{\log(\alpha + x)}{x}$ this function, defined on $(0, +\infty)$. Its derivative is $f' : x \mapsto \frac{1}{x}\left(\frac{1}{x+\alpha} - \log(x + \alpha)\right)$. But the function $x \mapsto \frac{1}{x+\alpha} - \log(x + \alpha)$ is non-increasing, goes to $-\infty$ at infinity and its only potential zero is $e^{W(1)} - \alpha$ if it is positive,[4] which yields the result. $\square$

**Lemma G.7.** *For $\Xi \subset \mathbb{R}^d$ a compact set, $g \in \mathcal{C}(\Xi)$ and, $Q \in \mathcal{P}(\Xi)$,*

$$\log \mathbb{E}_{\xi \sim Q}\left[e^{g(\xi)}\right] \leq \frac{\mathbb{E}_{\xi \sim Q}\left[g(\xi)e^{g(\xi)}\right]}{\mathbb{E}_{\xi \sim Q}\left[e^{g(\xi)}\right]} \,.$$

*Proof.* Define $\phi : t \mapsto \log \mathbb{E}_{\xi \sim Q}\left[e^{tg(\xi)}\right]$ which is convex and differentiable, since $g$ is continuous on the compact set $\Xi$. Hence,

$$0 = \phi(0) \geq \phi(1) + \phi'(1)(0 - 1) \,,$$

so that $\phi'(1) \geq \phi(1)$ which is the desired inequality. $\square$

**Lemma G.8.** *For $a, b, c, r > 0$ fixed, consider the function defined on $\mathbb{R}_+$ by*

$$\phi(\lambda) = a\lambda + \frac{b}{\lambda + r} - c\log(\lambda + r) \,.$$

*Then, for any $\overline{\lambda} > 0$, $\phi$ is strongly convex on $[0, \overline{\lambda}]$ with strong convexity constant*

$$\mu^* := \frac{2b}{(\overline{\lambda} + r)^3} + \frac{c}{(\overline{\lambda} + r)^2} \,.$$

*and the unique solution to the minimization problem*

$$\min_{\lambda \geq 0} \phi(\lambda)$$

*is given by,*

$$\lambda^\star = \left[\frac{c + \sqrt{c^2 + 4ab}}{2a} - r\right]_+ \,.$$

*Proof.* $\phi$ is twice differentiable and its derivatives are, for $\lambda \geq 0$,

$$\phi'(\lambda) = a - \frac{b}{(\lambda + r)^2} - \frac{c}{\lambda + r}$$

$$\phi''(\lambda) = \frac{2b}{(\lambda + r)^3} + \frac{c}{(\lambda + r)^2} \,,$$

which shows that $\phi$ is strictly convex on $\mathbb{R}_+$ and yields its strong convexity on compact intervals. Then, the first order optimality condition $\phi'(\lambda) = 0$ gives us that

$$a(\lambda + r)^2 - c(\lambda + r) - b = 0 \,, \tag{46}$$

which has an unique solution satisfying $\lambda + r \geq 0$ which is given by,

$$\lambda^\star = \frac{c + \sqrt{c^2 + 4ab}}{2a} - r \,.$$

If $\lambda^\star \geq 0$, then this is the solution we are looking for. If is not, this means that both roots of (46) are non-positive and therefore $\phi'(0) \geq 0$ which means that $0$ is the solution to the minimization problem. $\square$

---

[4]$W$ denotes the Lambert function, i.e., the inverse of the map $x \mapsto xe^x$.

**Lemma G.9.** *For $c > 0$, $\overline{\rho} > 0$ such that $\overline{\rho} \geq (4c)^{1/3}$, the inequality*

$$\overline{\rho}^2 - \rho^2 - \frac{c}{\rho} \geq 0$$

*holds in particular when*

$$\frac{2c}{\overline{\rho}^2} \leq \rho \leq \overline{\rho} - \frac{2c}{\overline{\rho}^2}$$

*Proof.* When $0 < \rho \leq \overline{\rho}$, the inequation $\overline{\rho}^2 - \rho^2 - \frac{c}{\rho} \geq 0$ is implied by

$$\overline{\rho}^2 \rho - \overline{\rho}\rho^2 - c \geq 0 \,.$$

Solving the latter yields the interval

$$\left[ \frac{\overline{\rho}^2 - \sqrt{\overline{\rho}(\overline{\rho}^3 - 4c)}}{2\overline{\rho}} , \; \frac{\overline{\rho}^2 + \sqrt{\overline{\rho}(\overline{\rho}^3 - 4c)}}{2\overline{\rho}} \right]$$

and the inequality $1 - u \leq \sqrt{1 - u}$ for $u \in [0, 1]$ yields the result. $\qquad\square$

# H   Numerical illustrations

We present numerical experiments supporting our theoretical results. On logistic and linear regression models, we illustrate that, provided the radius is large enough, the robust loss on the training distribution is indeed an upper-bound on the true loss. For $f(\theta, \xi)$ as defined in Examples 4.1 and 4.2, we estimate the following probability, as in Esfahani and Kuhn (2018, §7.2.A),

$$\mathrm{P}^{\otimes^n} \left( \widehat{\mathcal{R}}^\varepsilon_{\rho^2}(f(\widehat{\theta}_n, \cdot)) \geq \mathbb{E}_{\xi \sim \mathrm{P}}\left[ f(\widehat{\theta}_n, \xi) \right] \right) \quad \text{where} \quad \widehat{\theta}_n = \arg\min_{\Theta} \widehat{\mathcal{R}}^\varepsilon_{\rho^2}(f(\theta, \cdot)) \,,$$

and $\mathrm{P}^{\otimes^n}$ denotes the distribution of the training set $(\xi_i)_{1 \leq i \leq n}$ with $\xi_i \sim \mathrm{P}$ i.i.d..

We observe on the plots that, for $\rho$ large enough, the above probability is close to 1, for both models and for both standard and regularized cases (as guaranteed by Theorems 3.1 and 3.4).

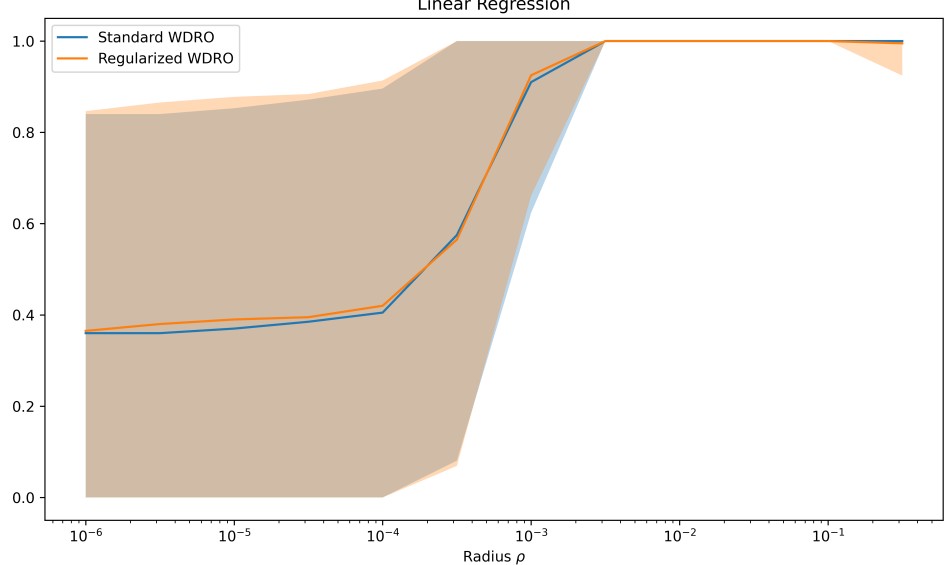

Estimates of the probability $P^{\otimes^n}\left(\widehat{\mathcal{R}}^{\epsilon}_{\rho^2}(f(\hat{\theta}_n,\cdot)) \geq \mathbb{E}_P[f(\hat{\theta}_n,\xi)]\right)$ where $\hat{\theta}_n$ is the robust model with radius $\rho$ for the linear regression model (Example 3.7). $\theta$ has dimension $d = 10$, $n = 1000$ synthetic training samples are used, $\sigma$ and $\epsilon$ are chosen proportional to $\rho$ following Theorem 3.4. For each value of $\rho$, we sample 200 training datasets and solve the WDRO problem on each of them, to obtain an estimate of the probability above. The solid line is the average over these 200 results, while the shaded area represents the standard deviation. As predicted by Theorems 3.1 and 3.4, we observe that for $\rho$ large enough, the probability that the robust loss on the training set upper bounds the true risk is almost 1. We also observe that standard and regularized WDRO have almost identical generalization behaviours. The WDRO problems are solved by LBFGS-B combined with the formulas of Example 2 of Wang et al. (2023).

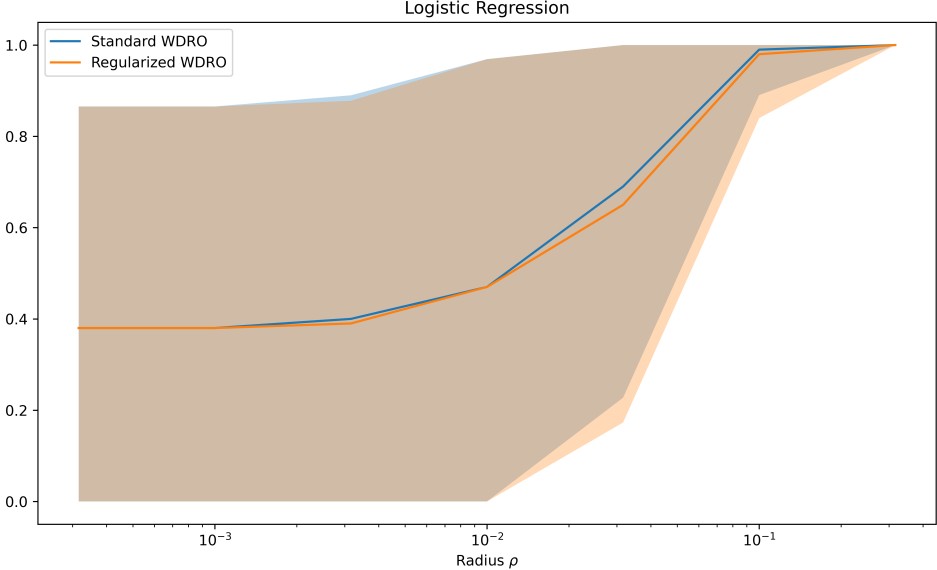

Estimates of the probability $P^{\otimes^n}\left(\widehat{\mathcal{R}}^{\epsilon}_{\rho^2}(f(\hat{\theta}_n,\cdot)) \geq \mathbb{E}_P[f(\hat{\theta}_n,\xi)]\right)$ where $\hat{\theta}_n$ is the robust model with radius $\rho$ for the logistic regression model (Example 3.6). $\theta$ has dimension $d = 5$, $n = 500$ synthetic training samples are used, $\sigma$ and $\epsilon$ are chosen proportional to $\rho$ following Theorem 3.4. For each value of $\rho$, we sample 100 training datasets and solve the WDRO problem on each of them, to obtain an estimate of the probability above. The solid line is the average over these 100 results, while the shaded area represents the standard deviation. As for the previous plot and as predicted by Theorems 3.1 and 3.4, we observe that for $\rho$ large enough, the probability that the robust loss on the training set upper bounds the true risk is almost 1. We also observe that standard and regularized WDRO have almost identical generalization behaviours. The standard WDRO problem is solved using the algorithm of Blanchet et al. (2022b) while the regularized problem is solved using LBFGS-B combined with an explicit expression of the inner integral in the robust loss.