# OpenReview forum: "Exact Generalization Guarantees for (Regularized) Wasserstein Distributionally Robust Models"
_NeurIPS.cc/2023/Conference — NeurIPS 2023 poster_

### Official Review · Reviewer_6D6k · 2023-07-01

**Soundness:** 3 good
**Presentation:** 3 good
**Contribution:** 2 fair
**Rating:** 5
**Confidence:** 3

**Summary:**

The paper provides theoretical results characterizing the generalization capabilities of methods based on Wasserstein distributionally robust approaches. In particular, the results presented extend the conditions under which the performance guarantees are not affected by the curse of dimensionality and are applicable for general classes of models.

**Strengths:**

The paper shows that the usage of Wasserstein radius of the order 1/sqrt(n) can provide generalization bounds in situations more general than those considered in existing works (linear models). In addition, the results presented also cover regularized versions of WDRO. The more general results are obtained using a novel type of proof based on a concentration bound for the dual problem, which is of independent interest.

**Weaknesses:**

The paper contribution with respect to the state of the art needs to be better described. In particular, the extension to non-linear models of the scaling 1/sqrt(n).

The problematic dimension-dependent scaling arises in Wasserstein methods while other techniques based on robust risk minimization have been shown to provide performance guarantees with the scaling 1/sqrt(n). It would be good if the authors describe this fact and the related work.

If I am not mistaken, examples 3.6 and 3.7 correspond to cases for which the right scaling was already proven in previous works. In order to better assess the paper's contribution, it would be good if the authors discuss interesting examples for which the paper provides the right scaling while existing results cannot.


**Questions:**

Would it be possible to include numerical results describing the theoretical results presented?. The choice of the radius in practice is often problematic in Wasserstein methods. The theoretical results provide the scaling of such radius but not a concrete recipe to choose it. It would be useful to explore choices of such radius with the right scaling that result in small error and provide performance guarantees.

**Limitations:**

The paper adequately describes the limitations of the methods proposed, mostly in terms of the specific assumptions needed for the results to hold.

---

> ### Author Rebuttal · Authors · 2023-08-09
>
> We thank the reviewer for the helpful comments and suggestions. Here is a detailed response to all remarks and questions.
>
> - *"The paper contribution with respect to the state of the art needs to be better described. In particular, the extension to non-linear models of the scaling 1/sqrt(n)."* Thank you for this important suggestion; we will clarify this point in the introduction. There are two points to make.
>     - To the best of our knowledge, the only existing *exact* generalization guarantees with the right $1 / \sqrt n$ scaling are in the work Shafieezadeh-Abadeh et al. (2019), Thm. 39. These guarantees are shown in the restricted context of linear models with Lipschitz loss (eg robust linear regression, support vector machines, logistic regression) and leverage a closed form of the robust risk, to get an *exact* upper bound, as in our results.
>     - The work of An and Gao (2021), which is the closest to ours, covers non-linear models, but provides generalization guarantees with additional error terms.
>
>
> - *"other techniques based on robust risk minimization have been shown to provide performance guarantees with the scaling 1/sqrt(n). It would be good if the authors describe this fact and the related work."* Other DRO neighborhoods indeed also provide similar generalization guarantees with the right scaling. In the revision, we will refer to, in particular, MMD DRO [1, 3]. A result very similar to ours -- an exact upper-bound on the true loss by the empirical robust -- indeed exist for MMD DRO [3, Cor. 3.1]. As other generalization guarantees for MMD DRO or its variants [2, 4], the radius of the MMD is indeed only required to scale as $1 / \sqrt n$. Adding this discussion in the introduction will help us underline that, in this work, we focus on advancing the theoretical understanding of WDRO.
>
> - *"it would be good if the authors discuss interesting examples for which the paper provides the right scaling while existing results cannot."* Thank you for this suggestion. We will improve the example section with  additional examples, namely kernel methods and neural networks, see the main rebuttal. For these models, our work is the first to give *exact* generalization guarantees in WDRO.
>
> - *"Would it be possible to include numerical results describing the theoretical results presented?"* We provide some numerical simulations on linear and logistic regression in the main rebuttal: we show that, both for standard and regularized WDRO, the robust risk with respect to the empirical training distribution is an upper bound on the test loss with probability almost one provided the radius $\rho$ is large enough.
>
> - *" The choice of the radius in practice is often problematic in Wasserstein methods [...]"* We fully agree that choosing the Wasserstein radius remains a major challenge in practice. Though some works have started addressing this question (Esfahani and Kuhn, 2018; Blanchet et al. 2021), this issue is part of our future work. We will mention this in the conclusion.
>
> References for Kernel DRO:
> [1] Zhu JJ, Jitkrittum W, Diehl M, Schölkopf B. Kernel distributionally robust optimization: Generalized duality theorem and stochastic approximation. AISTATS 2021
>
> [2] Jia-Jie Zhu, Christina Kouridi, Yassine Nemmour, Bernhard Schölkopf:
> Adversarially Robust Kernel Smoothing. AISTATS 2022:
>
> [3] Staib M, Jegelka S. Distributionally robust optimization and generalization in kernel methods. Advances in Neural Information Processing Systems. 2019
>
> [4] Zeng Y, Lam H. Generalization bounds with minimal dependency on hypothesis class via distributionally robust optimization. Advances in Neural Information Processing Systems. 2022

---

> > ### Comment · Reviewer_6D6k · 2023-08-15
> >
> > I thank the authors for their responses that mostly address my comments/questions. I believe the paper deserves to be published and it will be improved in the camera ready version

---

> > > ### Author Response · Authors · 2023-08-15
> > >
> > > Thank you for your kind words, and once again, we are grateful for your detailed comments and suggestions. If there are still specific points you would like us to expand upon, please feel free to let us know.

---

### Official Review · Reviewer_A8VY · 2023-07-05

**Soundness:** 3 good
**Presentation:** 3 good
**Contribution:** 3 good
**Rating:** 7
**Confidence:** 5

**Summary:**

This paper presents generalization bound for Wasserstein DRO and entropic regularized Wasserstein DRO (or called Sinkhorn DRO in Wang et al.) formulations. Those generalization bounds do not suffer from the curse of dimensionality. The theoreical analysis is also supported by two examples in Section 3.4.

**Strengths:**

- The theoretical analysis is interesting from two aspects. First, the authors reveal that the radius selection of WDRO to make the empirical robust loss  dominate the true loss does not suffer from the curse of dimensionality. The analysis follows different techniques from existing literature such as Gao et al, Blanchet et. al, etc. Second, the technique is general enough so that it also applies to entropic regularized Wasserstein DRO (or called Sinkhorn DRO in Wang et al.) formulations. This is the first work that investigates the statistical properties of such formulations.
- The authors also present two examples in machine learning to demonstrate the technique assumption holds and the proposed theoretical analysis applies.

**Weaknesses:**

- The writing of this paper could be potentially improved:
 1. There should be a comma in Eq.(1), or equation between line 83-84, or Eq. (4), or equation between line 194-195, or equation between line 301-302.
 2. There should be a period in Eq. (10).
 3. The contribution and related work part in the introduction section should be separated.
 4. It would be a little bit confusing to first introduce KL-divergence regularized WDRO risk in Eq.(5-6) and then introduce it corresponds to the Sinkhorn ambiguity set in line 194-195. The authors should put them together in Section 2.2
 5. The notation could be potentially improved. For example, in Eq. (7) the authors use $\hat{\mathcal{R}}$ to refer to the risk based on empirical distribution $P_n$. I would suggest replace the notation $P_n$ with $\hat{P}_n$ for consistency. Further, in Eq. (7) I think the authors are meaning $\rho$ should at least scale in the order of $\sqrt{(1+\log(1/\delta))/n}$, then why not write $\Omega(\sqrt{(1+\log(1/\delta))/n})\le \rho$ instead of $O(\sqrt{(1+\log(1/\delta))/n})\le \rho$? The same applies for equation between line 199-200.

- It is great that the authors present statistical analysis for entropic regularized Wasserstein DRO. I would suggest the authors add some explanation or numerical example to demonstrate the benefit of introducing entropic regularization. Will it bring extra benefits than standard WDRO?

- The analysis is limited to quadratic cost function, which could be restrictive. From my own trial and reading, I think the major difficulty for generalization is that, it is difficult to apply Laplace approximation technique for general p-th power of norm function. In other words, it is difficult to obtain the p-th power of norm counterpart of Lemma A.3 and Lemma G.1.If so, I suggest the authors add explanation for the difficulty of extension.

- Some literature is missing. For example, readers may wonder why consider adding entropic regularization to WDRO problem and what is the applications? I suggest the authors make the following revisions:
   1. update reference [J. Wang, R. Gao, and Y. Xie. Sinkhorn distributionally robust optimization. arXiv preprint arXiv:2109.11926, 2021] as [J. Wang, R. Gao, and Y. Xie. Sinkhorn distributionally robust optimization. arXiv preprint arXiv:2109.11926, 2023]. In the updated version, the authors demonstrate that people can find $\delta$-optimal solution to general entropic regularization WDRO problem with complexity $\tilde{O}(1/\delta^2)$. So one major benefit of adding entropic regularization is the computational tractability;
  2. add several application papers brought by entropic regularization WDRO in literature review:
     (i) Dapogny, Charles, et al. "Entropy-regularized Wasserstein distributionally robust shape and topology optimization." Structural and Multidisciplinary Optimization 66.3 (2023): 42.
     (ii) Song, Jun, et al. "Provably Convergent Policy Optimization via Metric-aware Trust Region Methods." arXiv preprint arXiv:2306.14133 (2023).
     (iii) Wang, Jie, and Yao Xie. "A data-driven approach to robust hypothesis testing using sinkhorn uncertainty sets." 2022 IEEE International Symposium on Information Theory (ISIT). IEEE, 2022.
     (iv) Wang, Jie, et al. "Improving sepsis prediction model generalization with optimal transport." Machine Learning for Health. PMLR, 2022.

**Questions:**

N/A

---

> ### Author Rebuttal · Authors · 2023-08-09
>
> We heartwarmingly thank the reviewer for the numerous suggestions, comments and references. It is a pleasure for us to read that "The theoretical analysis is interesting from two aspects [...]"  and that "it is great that the authors present statistical analysis for entropic regularized Wasserstein DRO [...]".  Here is a point-by-point response.
>
>
> - About the writing: thank you for the suggestions that we will implement in the revision.
>
> - *"I would suggest the authors add some explanation or numerical example to demonstrate the benefit of introducing entropic regularization."*
> Our work theoretically shows that regularized WDRO enjoys similar generalization guarantees as standard WDRO, with less restrictive assumptions: Thm. 3.4 for regularized WDRO requires less assumptions than both Thm. 3.1 and Thm. 3.3 on standard WDRO to obtain similar generalization results. On the numerical aspect, we will provide, in the revision, a discussion about the interest of entropic regularization. First, we will underline that J. Wang, R. Gao, and Y. Xie., (2023) illustrates that the out-of-sample performance of regularized WDRO is on par, if not better in some cases, than standard WDRO. We confirm this with some additional numerical simulations on linear and logistic regression in the main rebuttal, to be added to the revision.
>
> - *"The analysis is limited to quadratic cost function, [...] I suggest the authors add explanation for the difficulty of extension."* The reviewer is indeed right: it is not clear how to apply the Laplace approximation when the squared norm is replaced by the norm to the power $p$, and so, in particular, how to extend appendix A.3. Moreover, the analysis of appendix D.1 also relies heavily on $p$ being equal to 2 and more work would needed to determine whether it can be extended or not. On the other hand, the appendix D.2 would seem to extend to general exponents, as would appendix C when $\epsilon = 0$ (similarly to An and Gao (2021)). We will add this remark to the revision in conclusion.
>
> - About the additional references: we will gladly add them to the revision and update the reference  [J. Wang, R. Gao, and Y. Xie. 2023].

---

> > ### Comment · Reviewer_A8VY · 2023-08-10
> > **After reading the rebuttal**
> >
> > I have read the rebuttal and I am happy to raise my score to 7.

---

> > > ### Author Response · Authors · 2023-08-15
> > >
> > > Thank you again for your suggestions and comments, that will help improve our work!

---

### Official Review · Reviewer_wf5D · 2023-07-05

**Soundness:** 3 good
**Presentation:** 3 good
**Contribution:** 3 good
**Rating:** 7
**Confidence:** 3

**Summary:**

This work proves generalization guarantees for Wasserstein DRO models that only require the radius of order $O(n^{-1/2})$ under mild assumptions for general classes of models. This provides concentration results that do not suffer from the curse of dimensionality.

**Strengths:**

The theoretical contribution is the main strength. The empirical concentration of Wasserstein distance suffers from the curse of dimensionality, and this paper is able to prove the results (under some assumptions) that do not have this curse of dimensionality issue and provide statistical guarantees on the performance of WDRO solutions.

**Weaknesses:**

There is no significant weakness in this paper. Nevertheless, I think adding some discussion or examples for which the assumptions and thus the results in this paper do not hold can be beneficial; it can show failure cases and may also motivate future directions for the extension.

**Questions:**

It might be better to split Section 3 into two shorter sections for better readability. And the discussion may also be extended by adding examples of failure cases with potential methods of relaxation.

**Limitations:**

yes

---

> ### Author Rebuttal · Authors · 2023-08-09
>
> We thank the reviewer for their reading, comments and suggestions.
>
> - *"adding some discussion or examples for which the assumptions and thus the results in this paper do not hold can be beneficial;"* We agree that such examples were missing in the submission. In the main rebuttal, we provide more examples of parametric models, in particular kernel methods and neural networks. We also point out which cases our framework fails to cover in this context, see e.g. the discussion at the end of the kernel example. We will use all this material to enrich the example section in the revision.
>
> - *"It might be better to split Section 3 into two shorter sections for better readability."* Thank you for this suggestion which is well aligned with the addition of new examples. In the revision, we will create a section 4 dedicated to examples.

---

### Official Review · Reviewer_ZTJQ · 2023-07-05

**Soundness:** 3 good
**Presentation:** 2 fair
**Contribution:** 3 good
**Rating:** 7
**Confidence:** 2

**Summary:**

This paper provides generalization guarantees of Wasserstein DRO for a general class of functions, in which the radius scales as $1/\sqrt{n}$ and does not suffer from the curse of dimensionality. Moreover, these guarantees hold for any distribution in the neigbourhood of the true distribution, so that they still apply when the distribution shifts at testing time. The results in this paper hold for both constrained and regularized version of Wasserstein DRO. The authors also provide a proof sketch that explains the main ideas and techniques used in the proof, and apply their results to logistic and linear regression.

**Strengths:**

1. This paper provides novel generalization guarantees such that the robustness radius does not suffer from the curse of dimensionality. To the best of my knowledge, the results in this paper are novel and make a non-trivial contribution to the DRO community. Moreover, the authors consider the regularized version of Wasserstein DRO and provide similar guarantees as well.

2. Most parts of the paper are well-written. The necessary backgrounds are clearly explained, and theorems are accompanied with detailed explanations of related definitions and concepts.

**Weaknesses:**

In Section 3.4, the authors considers logistic and linear regression as applications of their theorems. It would be better if more complicated and popular parametric models can be included in this section to justify the main assumptions.

**Questions:**

1. How does the setting considered in this paper compared with other works? Could you give a brief and high-level discussion of why the generalization guarantee is dimension-independent in your setting?

2. Is it possible to obtain similar generalization guarantee for DRO with $\phi$-divergence?

**Limitations:**

This paper does not have potential negative societal impact.

---

> ### Author Rebuttal · Authors · 2023-08-09
>
> We thank the reviewer for their reading, comments and questions. We very much appreciate the comment "the results in this paper are novel and make a non-trivial contribution to the DRO community." Here are point-by-point answers to your questions.
>
> - *"It would be better if more complicated and popular parametric models can be included in this section to justify the main assumptions."*: In the main rebuttal, we discuss additional examples of parametric models: in particular, kernel regression and neural networks. We will add them to the revision of the paper.
>
> - *"How does the setting considered in this paper compared with other works?"* We will provide a more detailled comparison of our setting with the closest previous works:
>     - Blanchet et al. (2022), Blanchet and Shapiro (2023): these works consider the parametric setting, where $f(\theta, \xi)$ is twice differentiable with Lipschitz gradient, satisfies a uniform quadratic growth condition and study the asymptotic concentration properties of the WDRO solution in the neighborhood of a minimizer of the true risk.
>     - Gao (2022), An and Gao (2021): these works are the closest to ours. They consider piecewise differentiable loss functions on a compact set with Hölder gradient (however, note that, for their bounds to have vanishing errors, the points of non-differentiability have to be sufficiently far from the data distribution). Note however that they obtain generalization guarantees with additional error terms compared to our bounds, see the discussion in our paper line 181.
> - *"Could you give a brief and high-level discussion of why the generalization guarantee is dimension-independent in your setting?"* Our bound indeed does not suffer form the curse of dimensionality: the minimal radius to obtain generalization bounds scales as $1 / \sqrt n$ instead of $1 / n^{1/d}$ (Esfahani and Kuhn, 2018). The main difference with (Esfahani and Kuhn, 2018) is that we consider the WDRO objective as a whole, instead of proceeding in two steps: 1) considering the Wasserstein distance independently and invoking concentration results on the Wasserstein distance and 2) plugging this result in the WDRO problem.
> - *"Is it possible to obtain similar generalization guarantee for DRO with $\phi$-divergence?"* Though some of their generalization properties have been studied (eg Blanchet and Shapiro (2023)), to the best of our knowledge, this is not possible to obtain exact upper-bounds like ours for $\phi$-divergences. A possible explanation may be the following: unlike Wasserstein uncertainty sets, uncertainty sets defined with $\phi$-divergences only contain distribution whose support is included in the one of $P_n$.

---

### Author Rebuttal · Authors · 2023-08-09

We thank the reviewers for suggesting adding non-linear examples to the paper. We discuss three examples (kernel models, neural networks and family of invertible mappings) that we will add to the revision of the paper.
We also discuss a numerical illustration of our main theorems that we will also add to the revision.
### Kernel ridge regression
We present the example of kernel ridge regression and show that both Thm. 3.3 and 3.4 apply. Our work is the first to provide *exact* generalization bounds that do not suffer from the curse of dimensionality for these non-linear models in WDRO.

Take a kernel $k : \mathcal{X} \times \mathcal{X} \to \mathbb R$ with $\mathcal{X}$ compact and $k$ smooth (eg Gaussian, polynomial...). We consider the following class of loss functions:
$$\left\\{(x, y) \in \mathcal X \times \mathbb R \mapsto \frac{1}{2} \left(\sum_{i = 1}^m \alpha_i k(x, x_i) - y\right)^2 + \frac{\mu}{2} ||\alpha||^2_2: (\alpha_1,\dots,\alpha_m) \in A_m,\, (x_1,\dots,x_m) \in \mathcal{X}_m\right\\}$$
 where $m$ is a fixed integer, $A$ is a compact subset of $\mathbb R^m$, $\mathcal X_m$ can be any closed subset of $\mathcal X^m$ and $\mu \geq 0$ is the regularization parameter. A typical choice for $X_m$ would be the data points of the training set.

This class fits into our framework of parametric models of $\S3.4$ by setting $\xi = (x, y)$, $\Xi$ to some compact subset of $\mathcal X \times \mathbb R$,  $\theta=(\alpha_1,\dots,\alpha_m,x_1,\dots,x_m)$, $\Theta = A_m \times \mathcal X_m$ and
$$f(\theta, \xi) = \frac{1}{2} \left(\sum_{i = 1}^m \alpha_i k(x, x_i) - y\right)^2 + \frac{\mu}{2} ||\alpha||^2_2$$
- With that setting, Assumption 2 is readily satisfied so that Thm 3.4 applies.
- To apply Thm 3.3, we further need to assume that Assumption 4 holds. This non-degeneracy assumption is common in the WDRO literature (eg Blanchet et al. (2022); Blanchet and Shapiro (2023); Gao (2022); An and Gao (2021)).
- However, Thm. 3.1 cannot be applied directly as it is not yet clear what conditions on $k$ could ensure that Assumption 5 is satisfied. Moreover, as in other related works on WDRO, it is also not obvious how to extend this framework to cover non-smooth kernels (eg Laplace).
Finally, note that kernel logistic regression is also covered by our framework by combining the arguments above with the the logistic regression example Example 3.6.

## Smooth neural networks
We present the example of neural networks: as in the case of kernels, we show that both Thm. 3.3 and 3.4 apply to smooth neural networks. Again, our work is the first to provide *exact* generalization bounds that do not suffer from the curse of dimensionality.

Denote by $\mathcal{NN}(x, \theta, \sigma)$ a multi-linear perceptron that takes $x$ as input, has weights and biases $\theta$ and a smooth activation function $\sigma$ (eg GELU, tanh, ...). Choose $\ell(\hat y, y)$ a smooth loss function.
Then, we consider the family of losses $\left\\{ (x, y) \mapsto \ell(\mathcal{NN}(x, \theta, \sigma), y): \theta \in \Theta \right\\}$
with $\Theta$ some compact set.
Provided that the inputs $(x, y)$ lie in a compact set $\Xi$, the situation is the same as for kernels.
- Thm. 3.4 applies since Assumption 2 is readily satisfied.
- Thm. 3.3 applies provided the non-degeneracy assumption Assumption 4 is satisfied.
- We do not know how to ensure that Assumption 5 of Thm. 3.1 is satisfied for general neural networks nor how to extend this framework to cover non-smooth activation functions (eg ReLU).

## Family of diffeomorphisms and generative modelling
In this example, we show how the three theorems Thm. 3.1, 3.3 and 3.4 apply.
Consider a parametric function of the form $f(\theta, \xi) = h(g(\theta, \xi))$ where $g : \Theta \times \Xi \to \Xi$ is such that for any $\theta$, $g(\theta, \cdot) : \Xi \to \Xi$ is a diffeomorphism and $h : \Xi \to \mathbb R$ satisfies a mild technical assumption (the standard Morse-Bott condition, see eg (Arbel and Mairal, 2022, A.1)). Normalizing flows, widely used in generative modeling and sampling, are diffeomorphisms and thus lead to loss functions of this form.

For these functions, we readily see that Thm. 3.4 applies since Assumption 2 is readily satisfied and Thm. 3.3 applies provided the non-degeneracy assumption Assumption 4 is satisfied.
To apply Thm. 3.1, we show that Assumption 5 holds, as follows.
1. The function $h$ alone satisfies the first item of Assumption 5 since it is continuous and $\Xi$ is compact. $f$ then also satisfies it since $g$ is Lipschitz in $\xi$ uniformly in $\theta$.
2. We now show that the second item of Assumption 5 is also satisfied. In $\S A.5$, we showed that the second item of Assumption 5 is implied by the so-called parametric Morse-Bott assumption of Arbel and Mairal, 2022. But lemma 1 in $\S A.2$ in their paper shows that this family does satisfy this assumption in Lemma 1 in $\S A.2$.
Hence, Assumption 5 is satisfied and Thm. 3.1 also applies.
## Numerical illustration
We present numerical experiments supporting our theoretical results. On linear and logistic regression models, we illustrate that, provided the radius $\rho$ is large enough, the robust loss on the training distribution is indeed an upper-bound on the true loss. For $f(\theta, \xi)$ as defined in examples 3.6 and 3.7, we estimate the following probability, as in (Esfahani and Kuhn, 2018, $\S 7.2.A$),

$P\left(\hat{\mathcal R}^\varepsilon_{\rho^2} (f(\hat\theta_n, \cdot))\geq E_{P}[f(\hat\theta_n, \xi)]\right)\quad \text{where} \quad \hat\theta_n = argmin_{\Theta}\hat{\mathcal{R}}^\varepsilon_{\rho^2}(f(\theta, \cdot)))$


We observe on the plots (cf pdf) that, for $\rho$ large enough, the above probability is close to 1, for both models and for both standard and regularized cases (as guaranteed by theorems 3.1 and 3.4).

---

### Decision · Program_Chairs · 2023-09-21

**Decision:**

Accept (poster)

**Comment:**

The paper presents an interesting approach for robust generalization. Robust optimization turns to be instrumental in explaining the generalization of deep learning, and this work adds to this line of work. The work itself is very technical and better work could have been done by the authors to make the work accessible to the broader theoretical community of NeurIPS.